Registered report

# Testing the convergent validity, domain generality, and temporal stability of selected measures of people's tendency to explore

Farid Anvari [1,2,3,4] ✉, Stephan Billinger [2], Pantelis P. Analytis[2], Vithor Rosa Franco [5] & Davide Marchiori [2] ✉

Given the ubiquity of exploration in everyday life, researchers from many disciplines have developed methods to measure exploratory behaviour. There are therefore many ways to quantify and measure exploration. However, it remains unclear whether the different measures (i) have convergent validity relative to one another, (ii) capture a domain general tendency, and (iii) capture a tendency that is stable across time. In a sample of 678 participants, we found very little evidence of convergent validity for the behavioural measures (Hypothesis 1); most of the behavioural measures lacked sufficient convergent validity with one another or with the self-reports. In psychometric modelling analyses, we could not identify a good fitting model with an assumed general tendency to explore (Hypothesis 2); the best fitting model suggested that the different behavioural measures capture behaviours that are specific to the tasks. In a subsample of 254 participants who completed the study a second time, we found that the measures had stability across an 1 month timespan (Hypothesis 3). Therefore, although there were stable individual differences in how people approached each task across time, there was no generalizability across tasks, and drawing broad conclusions about exploratory behaviour from studies using these tasks may be problematic. The Stage 1 protocol for this Registered Report was accepted in principle on 2nd December 2022 https://doi.org/10.6084/m9.figshare.21717407.v1. The protocol, as accepted by the journal, can be found at https://doi.org/10.17605/OSF.IO/64QJU.

Exploration is ubiquitous in life. We explore when we try something new[1,2], test a promising but still unfamiliar option[3], search for causal relationships within a hypothesis space[4], choose among different strategies to solve problems[5,6], forage for information[7,8], and even when deciding on who to date or marry[9]. Many exploratory behaviours involve an inherent trade-off between exploration and exploitation. The fundamental trade-off in exploration-exploitation dilemmas can be expressed concisely: should you try an option that is (somewhat) unknown but that holds promise for a better outcome; or should you reap the rewards from the option that is best based on what is currently known? Still, the exact conditions of exploration-exploitation dilemmas in real life vary depending on the environmental properties of the specific decision-making problems. Such properties include how uncertainty is experienced, the relevant costs of exploring or not, the number of options to choose from, etc. Reflecting this diversity in real life decision-making problems as well as the methodological

[1]Social Cognition Center Cologne, University of Cologne, Cologne, Germany. [2]Strategic Organization Design group, University of Southern Denmark, Odense, Denmark. [3]Department of Psychology, Dresden University of Technology, Dresden, Germany. [4]Institute of Psychology, University of Bern, Bern, Switzerland. [5]Postgraduate Program of Psychology, São Francisco University, Campinas, Brazil. ✉e-mail: faridanvari.fa@gmail.com; mrcdvd77@gmail.com

preferences of each discipline, different tasks and self-report scales have been used to study people's tendency to explore across scientific fields (including but not limited to psychology, neuroscience, economics, biology, and management).

Consequently, there are many ways to quantify and measure exploratory behaviour[9]. However, there is little work that examines the validity of the different (behavioural and self-report) measures. As a result, three key questions remain unanswered: (i) Is there convergent validity between different measures of exploratory behaviour?; (ii) Does exploratory behaviour generalize across different tasks and situations such that different behavioural measures capture a domain-general tendency?; and (iii) Do the different measures of exploratory behaviour capture a tendency that is stable across time? We conducted a study that addresses these questions in the context of exploration-exploitation trade-offs. The study is one step towards bringing the interdisciplinary literature to bear on the claims and assumptions it contains.

The most common methods used to measure exploration are behavioural tasks and self-report scales. Behavioural tasks engage people in actual explore-exploit dilemmas in which they search and choose among different options or courses of action. Behavioural measures of exploration have face-validity and are intuitively appealing because they involve incentivized choices and observed behaviour in tasks that capture the fundamental trade-off in exploration-exploitation dilemmas. Nonetheless, their construct validity—whether they actually measure what they are supposed to measure—remains largely understudied (for exceptions, see refs. [10,11]). Self-report scales, by contrast, require people to respond to statements about their general search and exploration tendencies. These statements can be either broad and abstract (e.g., "I actively seek as much information as I can in new situations"), or specific and situational (e.g., "I take time to read the whole menu when dining out")[12–14]. Self-report measures are less costly than behavioural tasks, easier to administer, and potentially more general in their outlook. However, they rely heavily on people's introspective abilities and have often been criticised for this[15,16]. Recently, self-report scales of exploratory behaviour were criticised for their inability to predict exploration in behavioural tasks[17], though this leaves open the question about whether it is the self-reports that are the problem or the tasks. (See ref. [18] for a review of criticisms of self-reports and when and why self-reports may actually be superior to behavioural measures.)

Even within the behavioural task and self-report traditions, researchers have used a wide array of different measures to study exploratory behaviour. In the behavioural tradition, exploration has been examined in various paradigms including, among others, multi-armed bandit tasks[3,19,20], tasks designed to mimic foraging[11,21–24], search in a complex multi-attribute information space such as in the alien game[25–27], the sampling paradigm[28–33], the secretary task[11,34], the optional stopping task[35,36], the observe-or-bet task[37–39], and other minimal exploration-exploitation tasks[40,41]. Table S6 provides a brief description of each of these tasks. In the self-report tradition, researchers have used a variety of different questionnaires to uncover people's tendency to search or explore. Social and personality psychologists tend to use the Curiosity and Exploration Inventory[13,42,43], whereas judgment and decision making researchers mostly use one of the several existing maximization scales, part of which is intended to capture the tendency to exhaustively search for the best alternative[14,44,45].

Researchers studying exploration-exploitation dilemmas sometimes use a single behavioural paradigm or only self-report scales to study and draw conclusions about people's exploratory behaviour and tendencies more generally (e.g. refs. [3,21,22,30,37,46,47]). Such conclusions assume that empirical findings will generalize beyond the single measure used to other tasks and to other domains in life; or that responses on self-report scales reflect behaviour in general. Therefore,

it may appear as if researchers working on exploration-exploitation problems collectively assume that the different tasks and scales measure the same construct—that is, a general tendency to explore—or at least some aspects of it. In other words, it is, perhaps implicitly, assumed that due to the common fundamental nature of the exploration-exploitation trade-off (i) the different measures of exploration capture the same construct, and (ii) the tendency to explore is a domain-general (rather than task-specific) construct that is captured by the different measures. However, these assumptions have not been effectively tested. Probing these interrelated assumptions will advance our understanding of people's exploratory behaviour.

Several streams of research provide theoretical and empirical grounds to believe that the tendency to search or explore (in humans and animals) is domain-general, or that it is a trait-like tendency that is reflected by individual differences similar to, and sometimes embedded within, the personality traits[7,12–14,21,22,31,42,44,45,48–60]. For example, researchers have hypothesized a domain-general cognitive search process that calibrates people's tendencies towards exploration and exploitation[21,22,48,57,61]. A logical consequence of this hypothesis is that there is a domain-general tendency to explore, such that how and how much people explore and exploit will be similar across different contexts.

Some researchers have even argued that there may even be a genetic component that modulates uncertainty driven exploration in humans[62], and that differences in learning rates in bandit tasks can be linked to genetic variability[63]. Moreover, the cognitive models designed to capture people's behaviour in explore-exploit tasks are governed by the same fundamental principles, such as uncertainty reduction, and surprise-based learning. These cognitive mechanisms often have clear-cut signatures at the neural level as certain brain regions seem to participate consistently in exploration-exploitation decisions[46,64], suggesting that the same low level neural mechanisms are at play as well, providing a proximate explanation for a domain-general tendency to explore. Having domain-general mechanisms for mediating the exploration-exploitation trade-off is also supported from an evolutionary perspective[22,48], suggesting that proximate and ultimate explanations for a domain-general tendency are in agreement. If domain-general mechanisms mediate how people approach exploration-exploitation dilemmas, and the tendency to explore is trait-like, then exploratory behaviour should generalize across different task situations.

If different measures of exploratory behaviour, either obtained from behavioural tasks or self-reports, tap into the same construct then we should observe positive correlations between them—i.e., different measures of the same construct should show convergent validity[65–67]. However, there are only a few studies that have investigated (i) the convergent validity of multiple behavioural measures of exploration[10,11,68], or (ii) convergent validity between self-report measures and behavioural measures[17,44]; and these have together produced inconsistent or inconclusive results.

In terms of multiple behavioural measures, one study found no evidence for convergent validity between exploratory behaviour in three tasks (a multi-armed bandit, a secretary problem, and a foraging task), with the authors suggesting that structural differences between the tasks may have caused the lack of convergence[11]. Another paper reports several studies with mixed findings: a positive but moderate correlation between exploration in the sampling paradigm and the secretary task; but negative correlations between exploration in the sampling paradigm, observe or bet task, and a card search task, which are structurally very similar[10]. And yet a third paper reports two studies finding positive correlations between exploratory behaviour in two different tasks that are also structurally similar: the chain task and the grid task[68]. (In the chain task, participants could either press a button with some probability of moving forward to the next option for a higher reward or choose to stay on the current option for a small

reward. In the grid task, participants searched a 10 by 5 grid of options for rewards.) These contradictory findings may have a simple explanation. The sample sizes in these papers for the relevant correlations between each pair of tasks range from 52 to 261, many of them being under 100. The problem is that when the true correlation is small, studies with small samples can produce statistically significant results that not only have great variability around the true value, but a high likelihood that the significant results are in the wrong direction[69,70].

Indeed, in our pilot data reported below, we found a statistically significant positive correlation between exploration in the sampling paradigm and the observe or bet task ($r = 0.44$). This is in direct contrast to the negative correlation found between exploration in these same two tasks by Meyers and Kohler ($r = -0.33$)[10], as noted in the preceding paragraph. Their analyses for this correlation involved 56 participants, our pilot data consisted of 38 participants. These contradictory results thus provide further empirical evidence in support of the notion that statistically significant results may be in the wrong direction, given sample sizes too small to reliably detect small but theoretically relevant true correlations between the measures.

Regarding self-reports, the few studies that have investigated convergent validity between self-report and behavioural measures have used only a single behavioural task (the sampling paradigm), with correlations ranging between $r = -0.18$ and $0.32$ depending on the self-report scale and research group[17,44]. Hence it is not clear whether the problem is with the behavioural measure of exploration or the self-report measure, or possibly even both. More extensive research is therefore needed to systematically examine the convergent validity of different measures of exploratory behaviour, with a particular focus on widely used behavioural measures that share structural similarities.

Taken together, if two measures of exploratory behaviour tap into the same general tendency to explore, then they will be positively correlated. Hence, for each of the eleven measures of exploration in the present study, we hypothesized that it would have some convergent validity such that it would be positively correlated (i.e., $r > 0$) with the other measures (Hypothesis 1; see Table S11). There is no strict, widely accepted threshold for convergent validity. Convergent validity correlations found in the literature have been reported to be as low as $r = 0.2$, though it has been argued that this should be considered a very low, and perhaps unacceptable, standard[71,72]. We used $r = 0.2$ as a threshold. Therefore, to make all data patterns interpretable for each pair of measures tested in Hypothesis 1, such as if two measures are not statistically significantly correlated, we used equivalence testing to examine whether the correlation between them is smaller than $r = 0.2$. Hypotheses 1, combined with the corresponding equivalence tests, thus allowed us to identify the pairs of measures that have some convergent validity (statistically significant positive correlations) and those that do not have sufficient convergent validity (statistically significant equivalence tests)—statistically significant negative correlations would indicate a lack of convergent validity.

Uncovering the correlation structure of different measures of exploration, as in the present study, makes it possible to formulate a general index of exploration, $E$, which captures people's tendency to explore across different tasks and contexts. This would provide a strong test of the domain generality hypothesis of exploratory behaviour. However, to demonstrate the domain-generality of a theoretical construct, researchers should use psychometric modelling techniques, where the data are modelled such that there is an assumed unobserved or latent construct (e.g., a domain-general tendency to explore) that causes covariation in the observed variables (e.g., exploratory behaviour in the tasks). This methodology has been successfully used in other fields addressing similar research questions, such as for general cognitive abilities[73], risk preferences[74], psychopathology[75], cooperation[76,77], and the patterns of behaviours in behavioural economic games more generally[78].

We found only two studies that have used psychometric modelling to examine the domain-generality of exploratory behaviour. One study found evidence that exploratory behaviour was task-specific in three structurally dissimilar tasks[11]. Another study found evidence for domain-generality in two structurally similar tasks[68]. Although psychometric modelling has been used for self-report measures of exploration, showing that the different questions on the scales measure the same tendency, these have not included any behavioural measures[13,14,42,44,45,53]. Therefore, evidence for whether many commonly used paradigms of exploratory behaviour capture a domain-general or task-specific tendency is rather limited and inconclusive.

We tested the domain-generality hypothesis, with a focus on the behavioural measures of exploration, using psychometric modelling techniques including exploratory and confirmatory factor analyses[79]. Specifically, we randomly divided the data into two roughly equal parts. With one half of the data, we first derived three exploratory data-driven models. With the other half of the data, we then compared the three exploratory models and three baseline confirmatory models against each other with confirmatory factor analysis (CFA). If the tendency to explore is a domain-general construct that is captured by (at least some of) the different measures of exploratory behaviour in the present study, then the best fitting model from the CFA should be one in which a common latent factor describes variance across multiple behavioural measures, or a combination of behavioural and self-report measures (Hypothesis 2; see Table S11). This would provide some preliminary evidence for the domain-generality of the tendency to explore and the validity of the behavioural measures in capturing it. If a common factor could not be extracted, such that the data are better explained by modelling exploration in each task separately, then this would bring into question either the idea that the tendency to explore is a domain-general construct or the validity of the behavioural measures in capturing such a construct.

Finally, to be considered a domain-general or trait-like tendency, as argued earlier, an attribute must maintain some stability not only across situations but also across time—i.e., temporal stability[80]. Therefore, if there is a trait-like tendency to explore, how much a person explores now should be related to how much they explore later. Moreover, any measure that is intended to capture such a trait-like tendency must show this stability, such that exploratory behaviour in one task should positively correlate with exploratory behaviour in the same task across multiple points in time—that is, the measure should have test-retest reliability. For the behavioural measures, only two studies have examined test-retest reliability: One study showed that exploratory behaviour in a 2-armed restless bandit was positively correlated across two time points separated by another task[81]; another showed that exploratory behaviour in a 5-armed bandit was positively correlated when the task was completed twice back-to-back[11]. We found no studies that have systematically investigated the test-retest reliability of any other behavioural measures of the tendency to explore, or reliability over longer time periods. Some self-report scales have been found to have good test-retest reliability[12,42].

To examine the test-retest reliability of the different measures of exploration in the present study, we invited a random sample of participants to complete the same tasks and scales 1 month after their first participation. For each measure, we expected a positive correlation between exploration at Time 1 and Time 2 (Hypothesis 3; see Table S11)—the strength of the correlation gives an estimate of the measure's test-retest reliability. (Although stated as a hypothesis test, we were interested in estimating the reliability of each measure under the assumption that the tendency to explore is a stable trait-like construct, with the strength of the correlation between Time 1 and Time 2 being the estimate of the measure's reliability.)

In sum, there are theoretical reasons to expect a domain-general tendency to explore and that this tendency is captured by different behavioural and self-report measures of exploratory behaviour. This

**Table 1 | Descriptive statistics and reliabilities for each measure at Time 1 and the estimates of temporal stability from Time 1 to Time 2**

| Measure | Time 1 | | | Temporal stability |
|---|---|---|---|---|
| | M (SD) | Range | α [CI95%] | r [CI95%] |
| Bandit Switch | 0.36 (0.20) | 0, 1 | 0.90 [88, 0.92] | 0.51 [0.41, 0.59] |
| Bandit Exploit Comp. | 0.51 (0.23) | 0, 1 | 0.86 [84, 0.88] | 0.49 [0.39, 0.58] |
| Alien Hamming | 1.72 (1.17) | 0, 5.93 | 0.87 [84, 0.89] | 0.64 [0.56, 0.71] |
| Alien Active Search | 6.81 (2.53) | 0, 10 | 0.86 [84, 0.88] | 0.69 [0.62, 0.75] |
| Optional Stopping | 5.45 (3.05) | 1, 20 | 0.81 [76, 0.84] | 0.44 [0.34, 0.54] |
| Sampling Paradigm | 17.82 (20.30) | 2, 100 | 0.90 [87, 0.92] | 0.47 [0.37, 0.56] |
| Observe or Bet | 6.94 (6.32) | 0, 47.5 | 0.72 [63, 0.79] | 0.54 [.44, 0.62] |
| Alternative Search | 4.44 (0.90) | 1, 6 | 0.92 [91, 0.93] | 0.74 [0.68, 0.79] |
| Maximization Tendency | 3.46 (0.66) | 1.67, 5 | 0.86 [84, 0.87] | 0.71 [0.64, .76] |
| Exploration Scale | 3.44 (0.87) | 1.2, 5 | 0.88 [87, 0.90] | 0.77 [0.72, 0.82] |
| General Explore-Exploit | 5.83 (1.98) | 1, 10 | n/a | 0.59 [0.51, 0.67] |

Note. Bandit Switch = switch rate in multi-armed bandit task. Bandit Exploit Comp. = the complement of the best reply rate in multi-armed bandit task. Alien Hamming = Hamming distance in alien game. Alien Active Search = active search measure in alien game. Optional Stopping = exploration in the optional stopping task. Sampling Paradigm = average number of samples per block in the sampling paradigm. Observe or Bet = average number of observe trials in the observe or bet task. Alternative Search = mean ratings on the items of the alternative search scale. Maximization Tendency = mean ratings on the items of the maximization tendency scale. Exploration Scale = mean ratings on the items of the exploration scale. General Explore-Exploit = ratings on the single-item measure of the general tendency to explore-exploit.

idea has received some attention in recent times[10,11]. Although the studies addressing the issue have made significant contributions, the results have been inconsistent or inconclusive. For example, in one study, there was a negative correlation between exploration in two structurally very similar tasks (i.e., sampling paradigm and observe or bet task)[10], whereas another study found a positive correlation between a different pair of tasks that are also structurally very similar (i.e., grid search task and chain task)[68]. Likewise, exploration in the secretary task was positively correlated with exploration in the sampling paradigm in one study[10], but in another study it was negatively correlated with exploration in the multi-armed bandit (structurally very similar to the sampling paradigm)[11]. As noted earlier, these inconsistencies may be explained by sample sizes too small to reliably detect a correlation that is small, but relevant. In addition, stable individual differences in the tendency to explore may present themselves within measures across time (i.e., test-retest reliability), even if not across measures. Further, it is necessary to conduct a systematic examination of the convergent validity between different behavioural measures and different self-report scales.

We aimed to address the inconsistencies and gaps in the literature by conducting a study examining eleven different measures of exploration—seven measures from five behavioural tasks and four self-report measures—drawn from multiple disciplines: social and personality psychology, behavioural economics, cognitive science, and management. We used a large representative sample (more than double any previous study) and highly powered statistical tests suitable for not only detecting small effect sizes, but also for identifying the absence of theoretically important effect sizes. Our study would determine (i) the convergent validity of the eleven measures of exploration relative to one another, (ii) whether the various measures capture a domain-general tendency to explore, and (iii) the test-retest reliability of the different measures.

## Results

Table 1 presents the descriptive statistics and reliability coefficients for each of the 11 measures of exploration (means, standard deviations, range, and Cronbach's alpha).

### Hypothesis 1

Figure 1 presents the correlations and p-values for the one-sided correlation tests of Hypothesis 1 (H1-1 to H1-55; see Table S11). In the correlations testing Hypothesis 1, the 2 measures of exploration in the

multi-armed bandit (H1-1) were positively correlated with one another, as were the 2 measures of exploration in the alien game (H1-20), indicating that these 2 pairs of measures had some convergent validity. In addition, there were statistically significant, small correlations of the Hamming distance in the alien game with the switch-rate (H1-2) and the exploit-complement (H1-11) in the multi-armed bandit. Therefore, the Hamming distance in the alien game had some convergent validity with the two measures in the multi-armed bandit. None of the other behavioural measures of exploration were statistically significantly correlated with any other measure of exploration. The self-reports mostly showed some evidence of convergent validity with one another, except that the general explore-exploit question and the alternative search subscale were not statistically significantly correlated at the corrected alpha level. We tested the robustness of the results for Hypothesis 1 after removing extreme responses for the multi-armed bandit and alien game measures (i.e., people who had scores of 0 or 1 in bandit measures or 0 or 10 in alien game's active search). There were hardly any substantive changes to the results (reported in Fig. S3 in Supplemental).

For each pair of measures for which we found no evidence of convergent validity (i.e., the correlations were not statistically significant at the corrected alpha level), we conducted equivalence tests to examine whether there was evidence that the measures lacked sufficient convergent validity (i.e., whether the correlation between them was smaller than $r = 0.2$). The results of the equivalence tests are presented in Table 2. We found evidence for a lack of sufficient convergent validity for all except 7 (cells with bold text in Table 2) of the pairwise correlations that had statistically nonsignificant correlations. Specifically, the equivalence tests were nonsignificant for the relationships of: the switch rate in the bandit task with active search in the alien game; the exploit-complement in bandit task with number of boxes opened in the optional stopping task; the exploit-complement in bandit task with ratings on the Maximization Tendency Scale; the Hamming distance in the alien game with the number of boxes opened in the optional stopping task; the number of boxes opened in the optional stopping task with the number of samples in the sampling paradigm or with ratings on the Maximization Tendency Scale; and ratings on the Alternative Search Scale with the General Explore-Exploit Question. For these pairs of measures, therefore, we can only conclude that we found neither evidence for convergent validity nor evidence for a sufficient lack of convergent validity.

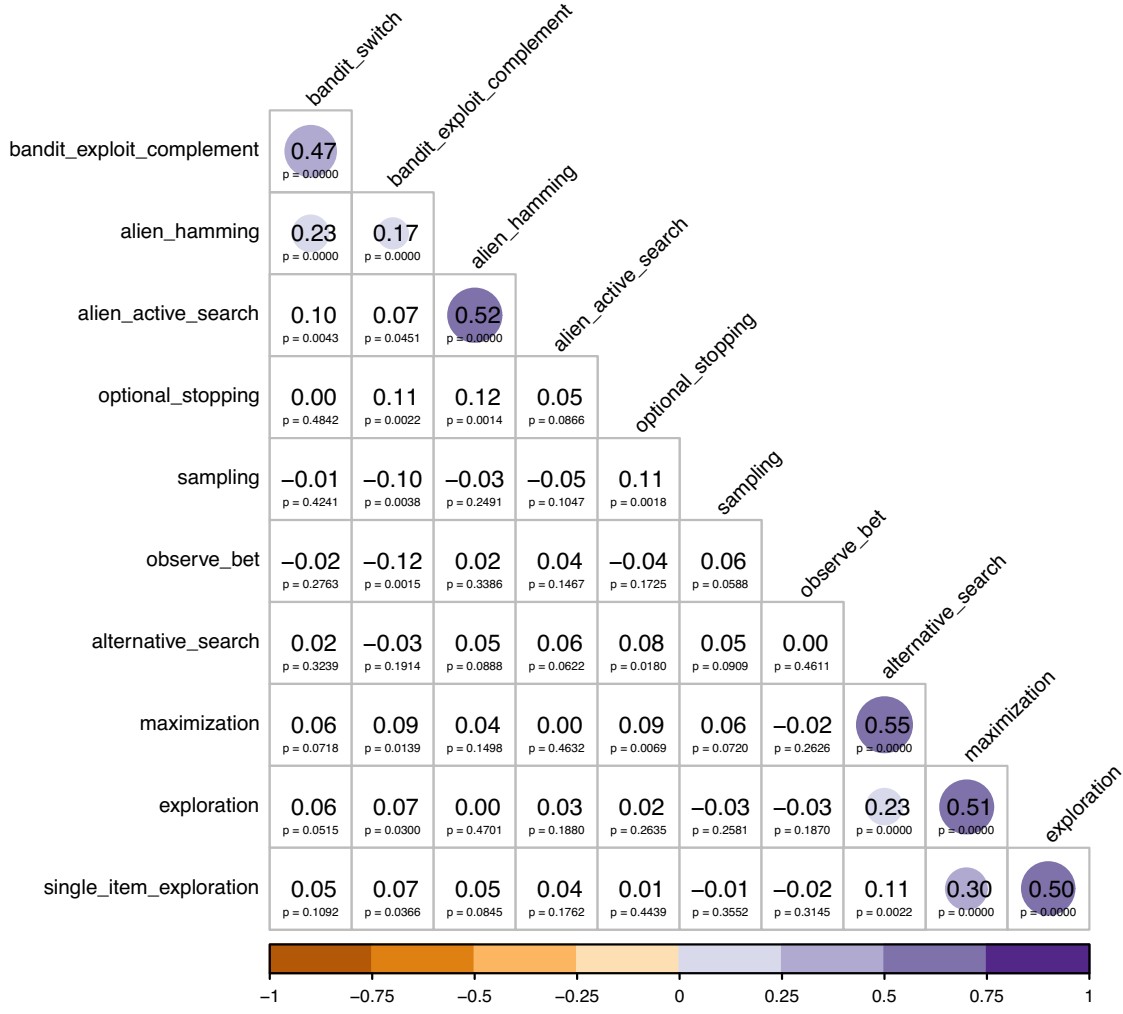

**Fig. 1 | Correlation coefficients (*r*, and one-sided *p*-values) testing Hypothesis 1.** One-sided Pearson correlation tests; the *p*-values are unadjusted for multiple comparisons; the Bonferroni corrected alpha is 0.0009. The circles indicate that the correlation was statistically significant at the corrected alpha level of 0.0009 (larger and more opaque means stronger correlation). bandit_switch = switch rate in multi-armed bandit task. bandit_exploit_complement = the complement of the best reply rate in multi-armed bandit task. alien_hamming = Hamming distance in alien game. alien_active_search = active search measure in alien game.

optional_stopping = exploration in the optional stopping task. sampling = average number of samples per block in the sampling paradigm. observe_bet = average number of observe trials in the observe or bet task. alternative_search = mean ratings on the items of the alternative search scale. maximization = mean ratings on the items of the maximization tendency scale. exploration = mean ratings on the items of the exploration scale. single_item_exploration = ratings on the single-item measure of the general tendency to explore-exploit.

In sum, the only behavioural measures of exploration that had evidence of some convergent validity were the Hamming distance in the alien game with the 2 measures in the bandit task, and the 2 pairs of measures derived from the same task (i.e., switch rate and exploit-complement, both in bandit task, and Hamming distance and active search, both in the alien game). Nonetheless, the correlations between the measures across different tasks were rather small (i.e., *r* = 0.17 and 0.23). For almost all of the other behavioural measures, we found evidence for a lack of sufficient convergent validity across tasks. In contrast, the self-report measures of exploration mostly had evidence for some convergent validity with one another, though this could of course be due to common methods variance[82]. There was evidence for a lack of sufficient convergent validity of the self-reports with the behavioural measures, except for the relationship of the Maximization Tendency Scale with the exploit-complement in the bandit task and with exploration in the optional stopping task. For these relationships, there was neither evidence of convergent validity nor evidence for a lack of sufficient convergent validity. Taken together, the behavioural measures of exploration tended to lack sufficient convergent validity

with one another or, when they showed evidence of some convergent validity, the correlations were very weak. Therefore, the behavioural measures should not be used interchangeably in studies of exploration, as they would likely produce different results and lead to different conclusions.

The results of the correlation tests for Hypothesis 1 suggest that, other than the Hamming distance in the alien game and the 2 measures of exploration in the bandit task, it is extremely unlikely that the behavioural measures are capturing a general tendency that generalizes across tasks or that the self-reports are capturing a tendency that predicts behaviour in such behavioural tasks.

**Hypothesis 2**

To test the domain-generality hypothesis (Hypothesis 2), we conducted preregistered psychometric analyses. We randomly split the data into halves, setting one half aside for the exploratory factor analyses (training sample) and the other half for confirmatory factor analyses and model comparisons (test sample). We used the WLSMV estimator because the normality assumption did not hold for

**Table 2 | Results (p-values) from the equivalence tests of Hypothesis 1**

| | 1. | 2. | 3. | 4. | 5. | 6. | 7. | 8. | 9. | 10. | 11. |
|---|---|---|---|---|---|---|---|---|---|---|---|
| 1. Bandit Switch | – | | | | | | | | | | |
| 2. Bandit Exploit Complement | n/a | – | | | | | | | | | |
| 3. Alien Hamming | n/a | n/a | – | | | | | | | | |
| 4. Alien Active Search | **p = 0.0050** | p = 0.0002 | n/a | – | | | | | | | |
| 5. Optional Stopping | p < 0.0001 | **p = 0.0085** | **p = 0.0125** | p < 0.0001 | – | | | | | | |
| 6. Sampling Paradigm | p < 0.0001 | p < 0.0001 | p < 0.0001 | p < 0.0001 | **p = 0.0105** | – | | | | | |
| 7. Observe or Bet | p < 0.0001 | p < 0.0001 | p < 0.0001 | p < 0.0001 | p < 0.0001 | p = 0.0001 | – | | | | |
| 8. Alternative Search Scale | p < 0.0001 | p < 0.0001 | p < 0.0001 | p = 0.0001 | p < 0.0001 | p < 0.0001 | p < 0.0001 | – | | | |
| 9. Maximization Tendency Scale | p < 0.0001 | **p = 0.0012** | p < 0.0001 | p < 0.0001 | **p = 0.0026** | p < 0.0001 | p < 0.0001 | p < 0.0001 | n/a | | |
| 10. Exploration Scale | p = 0.0002 | p = 0.0002 | p < 0.0001 | p < 0.0001 | p = 0.0004 | p < 0.0001 | p < 0.0001 | p < 0.0001 | n/a | – | |
| 11. General Explore-Exploit Question | p < 0.0001 | p = 0.0003 | p < 0.0001 | p < 0.0001 | p < 0.0001 | p < 0.0001 | p < 0.0001 | **p = 0.0078** | n/a | n/a | – |

*Note.* Equivalence tests consist of two one-sided tests, examining if the effect size is smaller and/or bigger than the lower and upper bounds of the smallest effect size of interest, respectively. The p-values are not adjusted for multiple comparisons; the corrected alpha level for the multiple comparisons is 0.0009. Cells with bold text indicate statistically nonsignificant equivalence tests (i.e., we found no evidence that the correlation was statistically significantly smaller than $r = 0.2$). n/a = the correlation test in Fig. 1 was statistically significant and so no equivalence test was required. Bandit Switch = switch rate in multi-armed bandit task. Bandit Exploit Complement = the complement of the best reply in multi-armed bandit task. Alien Hamming = Hamming distance in alien game. Alien Active Search = active search measure in alien game. Optional Stopping = exploration in the optional stopping task. Sampling Paradigm = average number of samples per block in the sampling paradigm. Observe or Bet = average number of observe trials in the observe or bet task. Alternative Search Scale = mean ratings on the items of the alternative search scale. Maximization Tendency Scale = mean ratings on the items of the maximization tendency scale. Exploration Scale = mean ratings on the items of the exploration scale. General Explore-Exploit Question = ratings on the single-item measure of the general tendency to explore-exploit.

our data (see Supplemental for details of the analyses examining the normality assumption).

Following our preregistered plan for testing Hypothesis 2, we first did a preliminary check of how well the confirmatory baseline models fit the data in the training sample (Baseline1, Baseline2 and Baseline3 are represented in Figs. 2, 3, and 4, respectively). Table 3 shows the factor loadings and Table 4 shows the fit statistics. The factor loadings can be interpreted as the correlation between the measure and the underlying common factor. From Table 3, it is possible to see that, overall, the behavioural measures seem to be less consistent than the self-report measures. More specifically, the measures derived from the alien game and from the multi-armed bandit share some variance that is not shared much with the other behavioural tasks (see factor loadings in Table 3 on the Behaviour factors of Baseline2 and Baseline3). It is also noticeable that the measures do not all load very well on what should be the general factor of exploration (in Baseline1 and Baseline3). None of the behavioural measures load at the threshold of 0.3 on the Exploration factor in Baseline1, and only the two measures in the bandit task load well onto the exploration factor in Baseline3 with the other behavioural measures loading weakly and/or negatively onto that factor. In Baseline2, the measures from the bandit and alien tasks load well onto the behaviour factor but the remaining 3 behavioural measures load very poorly, 2 of which have negative loadings. Not shown in the table, the correlation between the factors in Baseline2 was not statistically significant ($z = 1.499$, $p = 0.1338$, $r = 0.120$). Thus, according to the factor loadings, and from a theoretical perspective, none of the Baseline models fit the data very well in the training sample. According to the preregistered fit statistics (CFI, TLI, and RMSEA), Baseline1 and Baseline2 fit the training data poorly, whereas Baseline3 was acceptable (see Table 4).

We added one set of additional analyses for Hypothesis 2 that we did not preregister but which help with interpreting the data for interested readers. Table 3 includes, in the bottom row, results of analyses for composite reliability (CR)[83]. The CR is a measure of the internal consistency of a score, similar to Cronbach's alpha, but that takes into account that each indicator is differently related to the underlying common factor. From the CR, it is possible to see that, overall, the behavioural measures seem to be less consistent than the self-report measures.

The next preregistered step was to use the training sample to derive the three exploratory models. Regarding the first exploratory model (Exploratory1), we used the exploratory graph analysis (EGA) procedure to derive an estimate of the most likely factor model (Fig. S6 shows this model, with the results interpreted such that nodes of the same colour should form a common factor). These results indicated that the measures from the multi-armed bandit and the observe or bet tasks should form a single factor, that the measures from the alien game should form a single factor, the sampling paradigm and the optional stopping task would form a single factor, and the self-reports would form a final fourth factor. Complementary to the EGA, we conducted exploratory factor analysis where the number of extracted factors was estimated by the parallel analysis to be equal to four (Table S9 shows the results). The measures from the optional stopping, sampling, and observed bet tasks did not load well onto any factor (i.e., all their absolute factor loadings are below 0.300 and/or the loadings were negative). These results suggest that there are four different factors, each with only two measures: 1 factor with the bandit measures, 1 factor with the alien measures, 1 factor with the two self-reports from the judgement and decision making literature, and 1 factor with the exploration scale and the general explore-exploit question.

Taking into account the results of the EGA and complementary exploratory factor analysis together (shown in Fig. S6 and Table S9), the Exploratory1 model was proposed as follows. First, only three factors were kept: one with all of the self-report measures; the second with both measures from the bandit task; and the third factor with

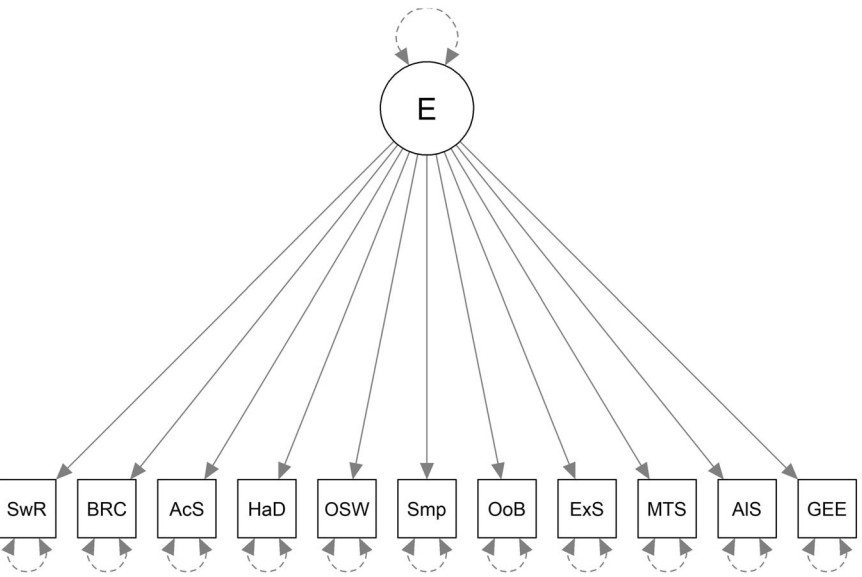

**Fig. 2 | Baseline1 model.** E general factor. SwR multi-armed bandit switch-rate. BRC multi-armed bandit best-reply complement. AcS alien game active search. HaD alien game Hamming distance. OSW optional stopping with recall. Smp sampling paradigm. OoB observe or bet. ExS exploration scale. MTS maximization tendency scale. AIS alternative search. GEE general explore-exploit question.

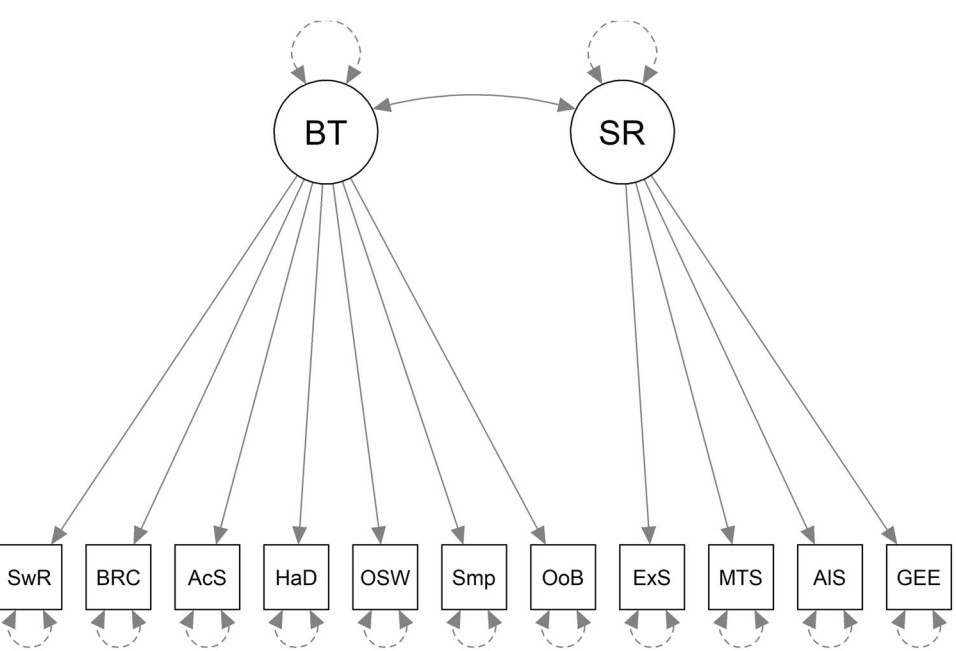

**Fig. 3 | Baseline2 model.** BT behavioral-tasks specific factor. SR self-reports specific factor. SwR multi-armed bandit switch-rate. BRC multi-armed bandit best-reply complement. AcS alien game active search. HaD alien game Hamming distance. OSW optional stopping with recall. Smp sampling paradigm. OoB observe or bet. ExS exploration scale. MTS maximization tendency scale. AIS alternative search. GEE general explore-exploit question.

both measures from the alien game. These factors can be considered methods factors, given that their respective measures are derived from the same method (i.e., self-reports) or task. The measures from the optional stopping, sampling, and observe or bet tasks were considered independent of all the other measures, as their absolute loadings were below 0.300 (see Table S9). We also included error covariances between (i) the exploration scale and general explore-exploit question and (ii) the alternative search and maximization self-report measures. These were included because of their loadings in Table S9, indicating that they share additional variance beyond the variance shared with the other self-report measures (i.e., Fig. S6 shows that the self-reports

should form a single factor together, but Table S9 shows 2 factors each with 2 self-reports). Error covariances were used instead of an independent common factor for the 2 pairs of measures because this latter model was unidentified. The Exploratory1 model is represented in Fig. 5.

For Exploratory2, we preregistered using the factors identified for Exploratory1 and adding a general factor of exploration so that Exploratory2 would be a bifactor model. However, as evidenced first with Baseline3 and in Hypothesis 1, because the measures are poorly correlated, the bifactor produces null and negative loadings for many of the measures, which do not allow the factor to be interpreted as a

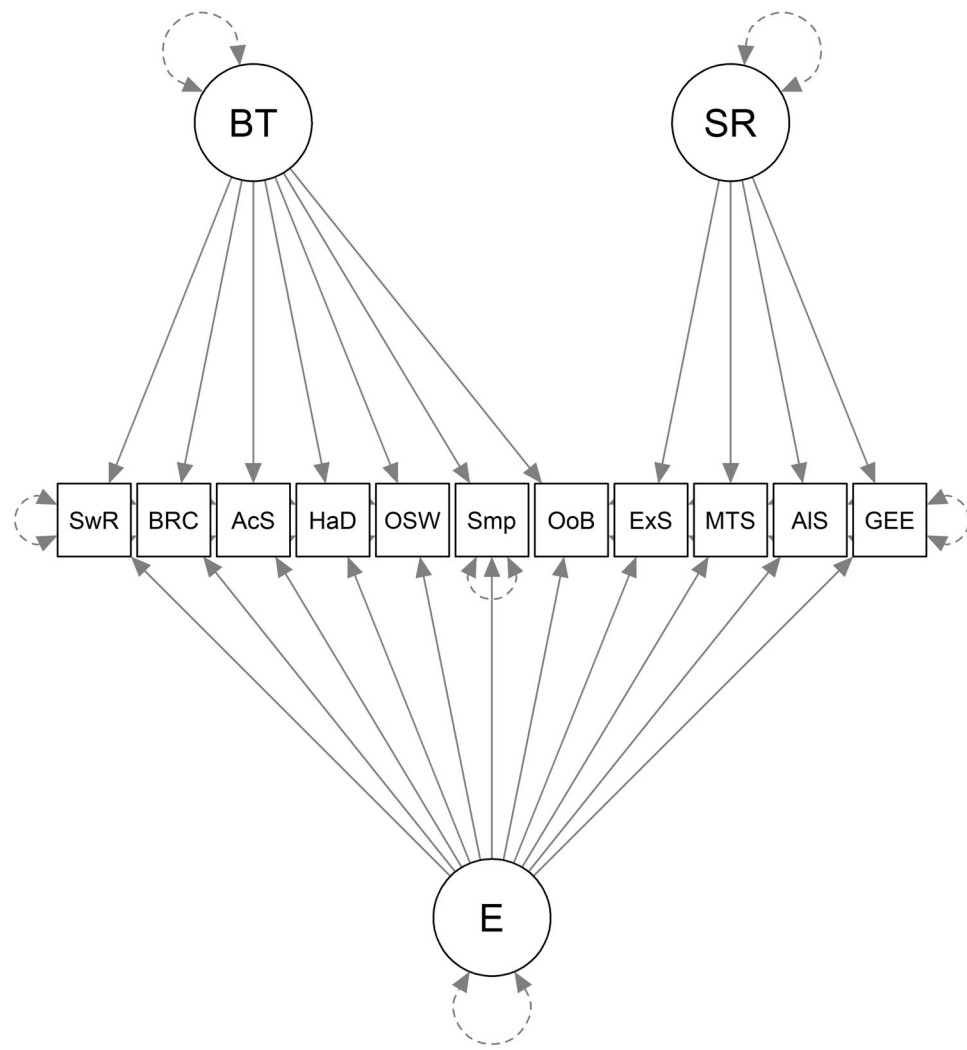

**Fig. 4 | Baseline3 model.** BT behavioral-tasks common method factor. SR self-report measures' common method factor. E general factor. SwR multi-armed bandit switch-rate. BRC multi-armed bandit best-reply complement. AcS alien game active search. HaD alien game Hamming distance. OSW optional stopping with recall. Smp sampling paradigm. OoB observe or bet. ExS exploration scale. MTS maximization tendency scale. AlS alternative search. GEE general explore-exploit question.

**Table 3 | Factor loadings and composite reliabilities (CR) of the baseline models and the Exploratory1 model in the training sample**

| Indicators | Baseline1 | Baseline2 | | Baseline3 | | | Exploratory1 | | |
|---|---|---|---|---|---|---|---|---|---|
| | Exploration | Behaviour | SelfReport | Behaviour | SelfReport | Exploration | SelfReport | Alien | Bandit |
| bandit_switch | 0.144 | 0.631 | | 0.310 | | 0.462 | | | 0.787 |
| bandit_exploit_complement | 0.160 | 0.526 | | 0.231 | | 0.752 | | | 0.551 |
| alien_hamming | 0.056 | 0.499 | | 0.870 | | −0.049 | | 0.975 | |
| alien_active_search | 0.050 | 0.379 | | 0.613 | | −0.075 | | 0.548 | |
| optional_stopping | 0.102 | 0.109 | | 0.115 | | 0.030 | | | |
| sampling | −0.026 | −0.151 | | 0.034 | | −0.241 | | | |
| observe_bet | −0.090 | −0.123 | | 0.072 | | −0.272 | | | |
| alternative_search | 0.516 | | 0.527 | | 0.543 | −0.024 | 0.327 | | |
| maximization | 0.877 | | 0.899 | | 0.903 | 0.113 | 0.752 | | |
| exploration | 0.627 | | 0.630 | | 0.606 | 0.151 | 0.701 | | |
| single_item_exploration | 0.364 | | 0.362 | | 0.342 | 0.111 | 0.323 | | |
| CR | 0.636 | 0.499 | 0.710 | 0.469 | 0.704 | 0.342 | 0.618 | 0.755 | 0.624 |

general factor of exploration. Moreover, Exploratory1 with the addition of a general factor produced a nonidentifiable model. Therefore, the Exploratory2 model was both computationally infeasible and theoretically nonsensical and was not examined further.

For Exploratory3, we planned to use an exploratory bifactor analysis. However, the feasible bifactor model derived with this method was very similar to the Baseline3 model. In fact, the only difference was that the factor loadings of the optional stopping, sampling, and observe or bet measures were all too low to be included in the model (i.e., absolute value of the largest factor loading was below 0.300). Therefore, we decided to not include Exploratory3 in the model comparisons.

The factor loadings of the Exploratory1 model in the training sample are shown in Table 3. It can be seen that, in comparison to

### Table 4 | Comparison of fit indices for the baseline and exploratory models fitted to the training and test datasets

| Model | Sample | $\chi^2$ | df | p | CFI | TLI | RMSEA |
|---|---|---|---|---|---|---|---|
| Baseline1 | Training | 231.680 | 44 | 0.000 | 0.390 | 0.238 | 0.115 |
| | Test | 204.479 | 44 | 0.000 | 0.472 | 0.340 | 0.106 |
| | Difference | −27.201 | — | — | 0.082 | 0.102 | −0.009 |
| Baseline2 | Training | 120.993 | 43 | 0.000 | 0.747 | 0.676 | 0.075 |
| | Test | 100.094 | 43 | 0.000 | 0.812 | 0.760 | 0.064 |
| | Difference | −20.898 | — | — | 0.066 | 0.084 | −0.011 |
| Baseline3 | Training | 59.587 | 33 | 0.003 | 0.914 | 0.856 | 0.050 |
| | Test | 52.443 | 33 | 0.017 | 0.936 | 0.893 | 0.043 |
| | Difference | −7.144 | — | — | 0.022 | 0.037 | −0.007 |
| Exploratory1 | Training | 72.405 | 42 | 0.002 | 0.901 | 0.871 | 0.047 |
| | Test | 45.567 | 42 | 0.326 | 0.988 | 0.985 | 0.016 |
| | Difference | −26.838 | — | — | 0.087 | 0.114 | −0.031 |

*Note.* Chi-square tests, CFI, TLI, and RMSEA from Confirmatory Factor Analyses using the WLSMV estimator (because normality assumption did not hold) for each model in both the training and test samples.

Baseline2, the loadings of the Self-Report factor are lower, more obviously reflected in a smaller CR. The Alien and Bandit factors are considerably more consistent, with all loadings above 0.500. For this model, the Self-Report factor is not statistically correlated with the Alien factor ($z = 0.039$, $p = 0.9688$, $r = 0.003$) nor with the Bandit factor ($z = 1.711$, $p = 0.0871$, $r = 0.154$). But the Alien factor and the Bandit factor are statistically correlated ($z = 3.320$, $p = 0.0009$, $r = 0.320$).

The next step in the preregistered analysis plan for Hypothesis 2 was to do model comparisons using confirmatory factor analyses in the test sample. Table 4 shows the fit indices from the confirmatory factor analyses for each of the selected models in the different samples (even though we noted already that several of the models made no sense from a theoretical perspective). The fit indices on the test sample are relevant here, though we also report the fit indices for each model on the training sample and the difference between the training and test samples to show whether the model fit was better or worse between the training and test samples (positive difference in CFI and TLI and negative difference in RMSEA indicate improvement). We note here that all models had better fit statistics in the test sample as compared to the training sample; but non-preregistered analyses showed that this was due to chance (see Supplemental, section titled "Main Study: Sensitivity Analysis of Differences Between Training and Testing Samples").

The Baseline1 model (the one that considers that all the measures share a common variance) has the worst fit of all the models. Regarding the best fitting models, Baseline3 and Exploratory1 are similar, though Exploratory1 has better fit statistics in the test sample. Ordinarily, therefore, we could select both Baseline3 and Exploratory1.

However, Exploratory1 makes more sense from a theoretical perspective as compared to Baseline3. In Baseline3, for the training sample, 5 of the 11 measures had negative loadings on the exploration factor, and only the two bandit measures had loadings above 0.300. For the test sample, as shown in Table 5, two of the 11 measures had negative loadings on the exploration factor, and only the two bandit measures and the hamming distance measure of the alien task had

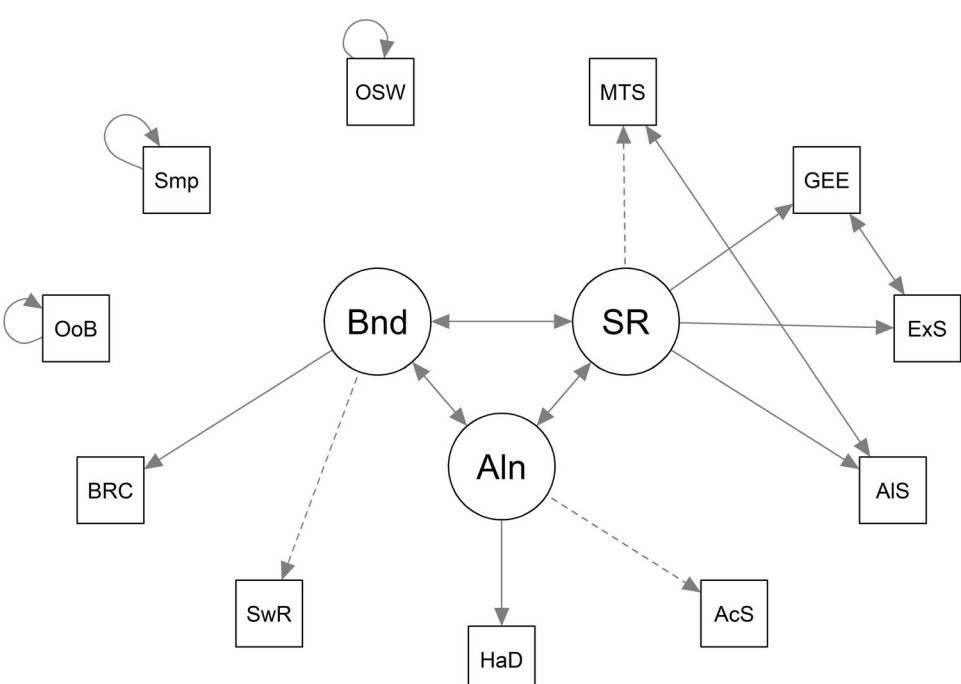

**Fig. 5 | Exploratory1 model.** SR: self-report measures' common factor. ExS exploration scale. MTS maximization tendency scale. AIS alternative search. GEE general explore-exploit question. BnD bandit task common factor. SwR multi-armed bandit switch-rate. BRC multi-armed bandit best-reply complement. Aln alien game common factor. AcS alien game active search. HaD alien game Hamming distance. OSW optional stopping with recall. Smp sampling paradigm. OoB observe or bet. The dashed arrows represent items with loadings fixed to 1.

**Table 5 | Factor loadings for all the models and indicators for the test sample**

| Indicators | Baseline1 Exploration | Baseline2 Behavior | SelfReport | Baseline3 BehaviorBias | SelfReportBias | Exploration | Exploratory1 Bandit | Alien | SelfReport |
|---|---|---|---|---|---|---|---|---|---|
| bandit_switch | 0.146 | 0.589 | | 0.174 | | 0.560 | 0.698 | | |
| bandit_exploit_complement | 0.157 | 0.603 | | 0.282 | | 0.761 | 0.689 | | |
| alien_hamming | 0.115 | 0.482 | | −0.620 | | 0.502 | | 0.421 | |
| alien_active_search | 0.103 | 0.316 | | −0.552 | | 0.278 | | 1.144 | |
| optional_stopping | 0.120 | 0.206 | | 0.050 | | 0.205 | | | |
| sampling | 0.047 | −0.042 | | 0.132 | | −0.017 | | | |
| observe_bet | −0.016 | −0.022 | | 0.059 | | −0.015 | | | |
| alternative_search | 0.473 | | 0.477 | | 0.492 | 0.033 | | | 0.263 |
| maximization | 0.768 | | 0.785 | | 0.807 | 0.089 | | | 0.559 |
| exploration | 0.668 | | 0.687 | | 0.665 | 0.101 | | | 0.855 |
| single_item_exploration | 0.570 | | 0.569 | | 0.529 | 0.157 | | | 0.664 |
| CR | 0.521 | 0.463 | 0.729 | 0.361 | 0.723 | 0.433 | 0.650 | 0.826 | 0.691 |

loadings above 0.300. Therefore, although Baseline3 had somewhat satisfactory fit indices, suggesting that the measures capture some common latent cause, we believe that the latent cause is uninterpretable in relation to a general tendency to explore.

The basic assumption in latent variable modelling (or psychometrics in general) is that correlations indicate that observed variables can be explained by the same latent common cause. In our case, higher values on the observed variables indicate that people explore more. Therefore, under the critical assumption that the latent variable is the general tendency to explore, we would expect the observed variables to be positively correlated with one another and to thus have positive loadings on the latent variable. Negative loadings are contradictory to the critical assumption. For example, the sampling paradigm has a negative loading on the latent variable in Baseline3. This means that in this task people who took more samples before choosing an option (i.e., those who could be said to have "explored more") would have a lower general tendency to explore, which is at odds with the definition of the measure itself. From this perspective, therefore, the obtained negative loadings mean that the latent variable in Baseline3 makes little sense with respect to an assumed general tendency to explore.

Regarding Exploratory1, only 1 measure loads slightly less than 0.300 on its respective factor and there are no measures with negative loadings. Moreover, Exploratory1's structure better reflects the correlation patterns in Fig. 1 as compared to Baseline3. Importantly, in non-preregistered analyses, we did 1,000 random splits into training and test samples and calculated the mean for each of the fit statistics as well as their 95% confidence intervals. These are reported in the Supplemental (Table S5). Exploratory1 had significantly better fit statistics than Baseline3 (i.e., higher CFI and TLI, and lower RMSEA) as indicated by the fact that the confidence intervals did not overlap with the point estimates. Thus, Exploratory1 is the best fitting model based on model comparisons as well as on theoretical grounds.

Therefore, we were not able to identify a model with acceptable fit indices in which multiple behavioural measures (from different tasks), or a combination of behavioural and self-report measures, could be explained by a common factor representing an assumed general tendency to explore. Indeed, the 2 factors with behavioural measures in Exploratory1 can best be thought of as methods factors, since the 2 measures in each of these factors are derived from the same behavioural tasks (i.e., alien game and multi-armed bandit). In short, we could not find support for Hypothesis 2. Instead, the best fitting model that made most sense from a theoretical perspective was Exploratory1, which assumes no general tendency to explore that explains covariation between the behavioural measures. In the Supplemental we also report the results of exploratory, non-preregistered analyses in which we modeled the behavioural measures at the level of blocks, rather

than aggregating across blocks, and including a general methods factor for the self-reports, finding a good fitting model that also did not include an assumed general tendency to explore. See "Non-Preregistered Exploratory Analyses for Hypothesis 2" in Supplemental.

### Preregistered assessment of measurement invariance

We preregistered testing the selected model from Hypothesis 2 for measurement invariance across time. Therefore, we did the invariance analysis for Exploratory1. To achieve this, we fit four different models: configural, weak factorial, strong factorial, and strict variance models. As decision criteria, a ΔCFI larger than −0.01, a ΔGamma hat larger than −0.001, and ΔMcDonald's NCI larger than −0.02 indicate that the null hypothesis of invariance should be rejected and that the measurement model is thus not invariant across time. The results are shown in Table S10. It is possible to see that all the indices indicate that the Exploratory1 model is strictly invariant; i.e., the factor loadings, the latent intercepts, and the residuals can be considered to be constant across Time 1 and Time 2, in these data.

### Hypothesis 3

The right-hand column of Table 1 (labelled, "Temporal Stability") presents the correlations for the tests of Hypothesis 3 (H3-1 to H3-11). All of these correlations were statistically significant (all $p$s < 0.0001). Every measure displayed some temporal stability. For the self-reports, the temporal stability estimates ranged from $r = 0.59$ to 0.74. For the behavioural measures, the estimates ranged from $r = 0.44$ to 0.69. These results show support for Hypothesis 3.

## Discussion

Many of our everyday behaviours involve striking a balance between exploration and exploitation, a balance between testing unknown or uncertain options and choosing the best option based on what is currently known. Given the ubiquity of exploration in everyday life, researchers across many disciplines have developed different methods to measure and study exploratory behaviour. And several lines of theoretical and empirical research suggest the existence of a trait-like, domain-general tendency to explore[7,12–14,21,22,31,42,44,45,48–60]. However, the extent to which the different measures of exploration have convergent validity with one another and capture a tendency that is domain-general was in need of systematic investigation. Using a registered report, we set out to examine, (i) whether different measures of exploratory behaviour that have been used in the social and behavioural sciences have convergent validity (Hypothesis 1), (ii) whether we could identify a latent factor model with an assumed trait-like tendency to explore that generalizes across different behavioural and self-report measures (Hypothesis 2), and (iii) whether the measures of

exploratory behaviour are stable across time (Hypothesis 3). To create a fair test we focused on broadly studied and conceptually similar behavioural measures, and correlated them also to commonly used self-report measures.

We found very little evidence for convergent validity for the behavioural measures (Hypothesis 1); most of the correlations involving behavioural measures were statistically smaller than $r = 0.2$, suggesting that they lacked sufficient convergent validity with one another or with the self-reports. Although there were two statistically significant positive correlations between behavioural measures across different tasks, these correlations are too weak for the measures to be used interchangeably ($r$s = 0.23 and 0.17). In psychometric modelling analyses, we could not identify a good fitting model with an assumed general tendency to explore (Hypothesis 2); the best fitting model suggested that the different behavioural measures capture behaviours that are specific to the tasks. Finally, our findings showed that the measures we examined had temporal stability, or test-retest reliability, across a 1 month timespan (Hypothesis 3); each measure also had good internal reliability, in so far as the measure within each block of a task correlated strongly with the same measure in the other blocks of the same task.

With a larger sample size, a wider selection of conceptually similar tasks, and the inclusion of self-reports, our results consolidate findings from previous studies and align with those that found weak and non-significant correlations between different behavioural measures of exploration [10,11,68]. In addition, our results testing Hypotheses 1 and 2 echo recent findings in several other domains in the behavioural sciences. For example, regarding convergent validity, researchers have found either no evidence for positive correlations, or very weak positive correlations, between different behavioural measures or between behavioural and self-report measures of risk preferences [74,84], empathy [85], creativity [86], self-control [87], and listening effort [88]. Similar issues have also been found for measures of physical properties, such as there being no correlations between different analysis methods of electroencephalogram (EEG) data obtained from the same group of participants [89,90]. Of the above studies, those that used factor analyses also could not identify good fitting models with a common factor (i.e., an assumed general tendency) that explained covariation across multiple behavioural measures or a combination of behavioural and self-report measures [74,84]. This is, however, not necessarily an inherent problem with behavioural tasks in general. In some domains, such as self-regulation [91], impulsivity [92], and pro-sociality [76,77], there does seem to be convergent validity and domain-generality across different behavioural measures, although the behavioural to self-report correlations may still be quite weak.

Our results on test-retest reliability for Hypothesis 3 are similarly in line with results from other domains. With a 1-month interval, the average test-retest correlation for the behavioural measures in our study was $r = 0.54$ (lowest = 0.44, highest = 0.69), and for the self-reports $r = 0.70$ (lowest = 0.59, highest = 0.77). In the domain of risk-preferences with a 6-month interval, the average test-retest correlation for behavioural measures was $r = 0.46$, and for self-reports $r$s = 0.68 and 0.65 for propensity and frequency measures, respectively [74]. For self-regulation with a time-interval of 2–8 months, the average estimate of test-retest reliability for behavioural measures was 0.45 and for self-reports it was 0.80 [91,93]. Therefore, although we found little to no evidence for convergent validity and domain-generality for the behavioural measures of exploration, individual differences in the way that people approach the explore-exploit dilemmas in the tasks that we selected have about the same level of stability across time as for behavioural measures from other domains.

There are several potential interpretations of our findings and we deal with each in turn. A simple interpretation is that there is no such thing as a domain-general tendency to explore in real life, despite the abundance of theoretical and intuitive reasons to expect such a

tendency. Instead, in each setting, people might behave in unique, task-specific ways that are adapted to or influenced by the unique contexts in which people find themselves [94,95]. This would imply that people in everyday life might be more or less exploratory than others, but only in specific domains. For example, some people may date more before committing to one person, others would try new foods more readily, or visit new cities and countries, or start new friendships more often, and so on. But someone who is more exploratory in one of these domains may not necessarily be more exploratory in the others. This could be due to fundamental differences across domains in people's preferences for exploration and/or the different heuristics and cognitive strategies involved in different contexts and tasks.

Although the above explanation about domain-specificity in exploration is parsimonious, it would be too soon to conclude that there is no general tendency to explore. There are alternative explanations that we will examine more closely. First, it could be that the tasks that we selected are not sufficiently similar. Second, the tasks' reliability may be too low for capturing individual differences. Third, the abstract tasks used by behavioural scientists may not be appropriate for measuring a general tendency that may be manifested in everyday life.

Regarding the first alternative explanation, in our study we selected tasks from different academic fields, but which had conceptual similarities with one another. That is, all tasks entail an inherent trade-off between exploration and exploitation, with some structural variations (see Table S8). We believe that this maximized the chances for identifying, with our psychometric analyses, a general factor across tasks. Yet, we found no single pair of behavioural tasks where the measures had substantially strong correlations, and thus no common factors that explained covariance across tasks. This is despite the clear conceptual connections across the selected tasks, and even with the inclusion of pairs of tasks that have particularly strong similarities. For example, until a final choice is made, every click on a button in the sampling paradigm is considered exploration; in the observe or bet task people click to observe the outcome on each trial without earning any points, and when they want to choose an option, they can bet. These two tasks and measures of exploration are very similar. And yet they lacked sufficient convergent validity.

Nonetheless, strong and stable correlations may possibly be found between behavioural measures in tasks that are conceptually and structurally even more similar than the ones we selected. For example, in the introduction we reported how Dale et al. [68], in relatively small samples of participants, found moderate-strong positive correlations ($r$s = 0.48 and 0.64) between exploration in two very similar tasks: a grid search task and a chain decision task. In tasks with extreme structural similarities, behaviour may be generalizable from one task to another, but this would not imply the existence of a general tendency. A general tendency to explore would also require generalizing across structurally dissimilar tasks. Our results suggest that it is unlikely that behavioural tasks (at least the ones we selected) capture a general tendency to explore.

To further test whether tasks used in the behavioural sciences can in some other way capture a general tendency to explore, future research could look for tasks where the same cognitive model can be applied to explain individual-level data patterns. Such research could examine (i) whether the same cognitive model explains data patterns in the different tasks, and (ii) the correlations between the parameters retrieved from the models across tasks. There is already some work demonstrating this approach. In the case of risk-taking behaviour, Pedroni et al. [96] used cumulative prospect theory [97], a broadly adopted and highly predictive model of risky choice, and retrieved the parameters of the model for the three, out of the six, risk-taking tasks that were more similar to each other. The authors found no evidence for sufficient convergent validity between model parameters across tasks (rank correlations were below 0.12). Closer to our topic, Wu et al. [59],

investigated how people deal with exploration-exploitation dilemmas in two tasks which were identical in terms of the reward structure and temporal horizon, but which differed in terms of the context or domain. The first task was framed as a spatial exploration task in which people searched for good options in a two dimensional grid. In the second task people searched in an abstract feature space (i.e., patterns in a similar sized two dimensional square). The authors found that the same Gaussian process model could explain people's behaviour in the two tasks. Yet only the parameter for undirected/random exploration was substantially correlated across tasks (Kendall rank correlation = 0.43). The parameter representing directed exploration was only weakly correlated (Kendall rank correlation = 0.18). Thus, even when the structural qualities of the environment are identical, the context might lead to considerable differences in the expressed behaviour, in which case generalizability between tasks seems modest.

In case people's exploratory preferences are context-dependent or domain-specific, an important task for future research and theorizing would be to define the boundaries of the domains. For example, is it a single domain of exploratory behaviour when people try out new foods, regardless of whether it is at new restaurants or new recipes to cook at home, or are these domains separate from one another? The boundaries of each domain should thus be outlined and conclusions from any given study should be clear about the contexts to which the results can be generalized. Theories and the parameters thereof would have predictive power only within a domain's boundaries and new empirical studies would be needed for different domains. Furthermore, the parameters of cognitive models used to explain people's behaviour in the exploration-exploitation dilemma would need to be retrieved in each specific context separately (see ref. 59).

A second alternative explanation of our results is that the behavioural measures of exploration may lack sufficient reliability. Dang et al.[98] and Hedge et al.[99] have argued that many tasks are designed to maximize within-person variance in behaviour at the cost of reducing between-person variance—which would reduce the measures' ability to capture individual differences and thus have convergent validity with other measures. However, in our study, the reliability coefficients (Cronbach's alpha) for each measure from the behavioural tasks were as high as for the self-reports. This suggests that the behavioural measures in our study did have sufficient between-person variability to capture individual differences. Our results are therefore unlikely to be due to insufficient between-person variability in the behavioural measures.

A third alternative interpretation of our results is that the types of tasks developed to make exploratory behaviour observable for scientific investigation may be too simplistic to capture a domain general tendency that manifests in real life. In everyday life, there may be some domain-generality in people's tendency to explore. For example, a person who is more exploratory in trying out new foods might also be, on average, more likely to try new destinations when travelling, form new friendships, and so on. Such a general tendency is unlikely to be expressed in the behavioural tasks designed to study exploration. This is because behavioural tasks use uncommon stimuli in specific, highly structured, task-dependent contexts[98] that have little relevance to people's daily lives. The problem is amplified when we consider the extremely different time scales in which exploration-exploitation dilemmas play out in behavioural tasks as compared to everyday life, evoking different psychological mechanisms[100]. Many real life explore-exploit contexts might be inherently linked with emotional experiences such as fear and anxiety, or joy and pleasure, which could create behavioural consistency across domains—e.g., the fear of trying something new and having a bad experience, or the joy and anticipation of finding new pleasurable experiences. The behavioural tasks designed for the study of exploration-exploitation dilemmas, by contrast, may evoke mostly cognitive processes rather than strong emotional responses. Further, our learning histories and habits which have

formed over a timespan of years and which may penetrate many aspects of our lives could create exploratory tendencies that are consistent across many domains but which are not captured in behavioural experiments. Yet another possibility is that the ecologies of exploration are fundamentally correlated in real life, because one set of activities might make others more possible and even more necessary. For example, people who are mobile and travel a lot may also be more likely to try more exotic cuisines and start new friendships more often. Similarly, people living in large cities will have many more opportunities to explore than those living in small towns.

In contrast to behavioural tasks, self-reports may be better equipped to capture a general tendency for exploration that manifests in real life contexts, at least to some extent. This is because people can respond to the items on self-reports by directly introspecting on how they behave across various situations where they encounter explore-exploit dilemmas. This is supported by some evidence using real-world data. Alessandretti et al.[101] found that self-reports on extraversion (from a personality inventory) were positively correlated with exploration in people's everyday lives, measured using mobile phone interactions and movement trajectories. In exploratory analyses in our study, we also found that the self-report measures of exploration were positively correlated with self-reports of extraversion, conscientiousness, and openness, and negatively with neuroticism (Fig. S4 in Supplemental). Self-reports, however, suffer from their own limitations. For example, some self-report questions that ask about general tendencies or cognitive processes may require a level of insight that participants may not always have[18]. Moreover, correlations between different self-report scales may be inflated by various forms of common methods bias[82]. This is why our focus in this paper was mainly on the behavioural measures, in which actual incentivized decisions are directly observed. Nonetheless, we found no correlations between self-reports and the behavioural measures in the various tasks.

Considering the evidence we have so far we believe that a fruitful avenue for future research would be to examine potential correlations of different behavioural tasks, and self-reports of exploration, with exploratory behaviours in real life settings. This could shed light on whether any of the measures that researchers use can statistically predict how much people explore in their everyday lives, and also whether there is a general tendency to explore across different real-world contexts (as compared to exploratory behaviour being domain specific). One approach to pursue such research is using digitally recorded behavioural traces, as in the study by Alessandretti et al.[101] where mobile phone tracking was used to study people's messaging and movement patterns, or in consumer choice contexts, as illustrated by Riefer et al.[102] in a study on supermarket choices and Schulz et al.'s[103] study on online food orders. In addition, researchers could examine exploratory behaviour in environments that are more inherently motivating such as in videogames[104,105] and virtual reality environments[106] to see whether behaviour in such contexts correlates with behaviour in the other contexts mentioned already. A major challenge down this path will be in quantifying exploration in real-world contexts, and connecting these to already established measures of exploratory behaviour in the lab or with self-reports.

We started with the premise that the fundamental trade-off in exploration-exploitation dilemmas can be expressed concisely and that a domain-general tendency to explore should be captured by the different behavioural and self-report measures we used in this study. We did not find domain generality across behavioural measures, although we found robust temporal stability for each of the measures. On the one hand, this suggests that researchers can well use the behavioural measures they've designed to examine people's strategies for dealing with the tasks' specific decision problems, and even individual differences in these strategies, which are, based on our findings, reliable and stable across time. On the other hand, drawing general conclusions about people's exploratory behaviours based on the

results from such studies may be problematic. Indeed, the results of these studies may not generalize outside of the task used. Therefore, the conclusions would be unjustified until further work reveals whether any of the behavioural measures captures a tendency that manifests in real world exploratory behaviours.

## Methods

### Ethics
The research complies with all necessary ethical requirements. Ethical approval was granted by the Research Ethics Committee of the University of Southern Denmark, case number 19/78419. Informed consent was obtained from all participants prior to participation. Including the anticipated bonus for performance on the behavioural tasks, participants who complete the study were expected to earn, on average, approximately £ 11.00 in total (see sampling plan below for details) for ~60 min of their time.

### Pilot data and simulations
We conducted four pilot studies using participants from Prolific.co. The purposes of these were to remove software glitches, ensure useability of the tasks, and to perform data quality checks. For the data quality checks, we combined the data from all pilot studies. These data included 20 participants with complete responses for the Alien Game, due to a software coding issue, and 38 participants with complete responses on the other behavioural tasks and self-report measures. As can be seen by the descriptive statistics in Table S7, there were no floor or ceiling effects for any of the measures of exploration. Moreover, we examined the internal consistency of each measure of exploration in the behavioural tasks. We did this by correlating the measure of exploration in each task across different blocks. For example, we calculated the switch rate for the bandit task using only the odd numbered blocks (i.e., blocks 1 and 3) and correlated this with the switch rate in the even numbered blocks (i.e., blocks 2 and 4). Apart from one combination of blocks for active search in the alien game, for which the small sample size produced large uncertainty around the estimates, all measures showed good internal consistency (see Supplemental Information).

A better measure of internal consistency is the Cronbach's alpha. We therefore calculated the Cronbach's alpha for each measure of the behavioural tasks, using each block as an observation. All measures showed satisfactory internal consistency as measured by Cronbach's alpha, though some had wide (95%) confidence intervals due to the small sample size: switch rate = 0.91 [0.80, 0.95]; best-reply-complement = 0.56 [0.14, 0.75]; Hamming = 0.87 [0.60, 0.95]; active search = 0.67 [0.25, 0.85]; observe or bet = 0.75 [0.27, 0.92]; optional stopping = 0.80 [0.48, 0.90]; samples = 0.74 [0.56, 0.83]. Importantly, as detailed in the "Internal consistency" section of the "Analysis Plan", we expect behavioural tasks to show lower internal consistency than self-reports, particularly because of the stochasticity underlying some tasks, and so shouldn't be held to the same standards on this measure of reliability. Therefore, we believe that even the best-reply-complement shows promise for satisfactory internal consistency.

In preparing the R code for the analyses for the main study we used the pilot data to run the code to make sure everything works correctly. In running the analyses on the pilot data, we found that, unsurprisingly, most of the one-sided correlation tests with and between the behavioural measures were not statistically significant at the corrected alpha of 0.005 (the corrected alpha for the pilot study was 0.005; after peer-review, the corrected alpha was changed to what is presented in the Analysis Plan below). The Supplemental Information contains plots with the correlations and uncorrected 95% confidence intervals between all the 11 measures (see Fig. S1 and Fig. S2, under the section "Results from Pilot Data" in the Supplemental Information). Surprisingly, there was a positive and statistically significant correlation between two behavioural measures of exploration:

the sampling paradigm and the observe or bet task, $r_{(36)} = 0.44$, $CI_{95\%}$ [0.19, 1], $p = 0.003$ (one-tailed). What's interesting about this is that it contradicts the negative correlation found between the same two measures in a previously published study[10], as we noted earlier in the introduction. This finding lends further support to the argument that small samples can produce unreliable correlation estimates, with errors of direction as well as of magnitude. The importance of our proposed study, with a very large sample, is thus inadvertently demonstrated by the results of our pilot data.

In addition, we ran simulations to test the potential test-retest reliability of the multi-armed bandit and the optional stopping with recall tasks. We focused on these two tasks because there are cognitive models available for them and it was feasible to model behavioural variability in these tasks. Thus, we simulated artificial agents in the tasks with the same parameters and design features that we'll be using for the main study (e.g., the payoff probabilities, and the number of blocks and trials), and we did this twice for each study with the same agents. Details of these simulations and the results are provided in the Supplemental Information. In sum, simulation results show that, assuming agents that behave consistently (i.e., with stable parameterizations of behaviour across task repetitions), both tasks have potential for producing good test-retest reliability correlations: $r = 0.95$ and $r = 0.96$ for the two measures from the multi-armed bandit; $r = 0.63$ for the optional stopping task.

### Design
Quality checks were implemented as per the exclusion criteria (as described below). This included requiring participants to correctly answer 3 comprehension check questions prior to beginning each behavioural task. Participants who didn't answer all of the questions correctly after 6 attempts for any one task were automatically moved onto the next task. One of the comprehension checks was an open text response, which helped to reduce the likelihood that bots or inattentive respondents would pass the check.

**Procedure.** Participants first read an information sheet describing an overview that outlined details of the study, including the number of tasks, the existence of comprehension checks and practice rounds, and the possibility of a bonus payment that participants can maximize by maximizing the points they earn. Following the information sheet, participants gave informed consent and then began the study. Participants were randomly allocated to either complete the behavioural tasks first or the self-report measures first. The behavioural tasks were presented in random order. All behavioural tasks started with a practice block that allowed participants to familiarize themselves with the tasks. Performance on each task was summarized and presented to participants after each block and their accumulated performance for each task was presented prior to beginning the next task. The four self-report measures of exploration that were tested in the confirmatory analyses were presented in random order (the general explore-exploit question was presented on the same page as another single-item question about near-far exploration that was included only for exploratory purposes). Finally, for exploratory purposes, participants completed some personality measures and responded to basic demographic questions (age and gender) before being debriefed, told their total bonus earnings, and paid.

**Behavioural tasks.** For each behavioural task, participants first read the instructions and then responded to three comprehension check questions (see Supplemental Information), to ensure all task details were attended to and understood, followed by a practice block with a few rounds. All details of the parameters for each task (e.g., payoff distributions), the instructions given to participants in full, which follow conventions and practices in published work, as well as screenshots of each task are provided in the descriptions below and/or

Supplemental Information. Each task consisted of a certain number of incentivized blocks which we used to measure exploratory behaviour, with each block consisting of a certain number of trials. In choosing the number of incentivized blocks and trials for each task, we aimed for a balance between the time spent in each task and the number of blocks and trials required in each task to avoid random events influencing the measure of exploratory behaviour. For example, in the optional stopping with recall task, someone could, by luck, select a high paying box on the first trial. If this task consisted of only two blocks, the measure of exploratory behaviour for that lucky person would be biased downwards, whereas for the observe or bet task, in which the number of trials is fixed, this is not an issue.

**Multi-armed bandit.** In our implementation of the multi-armed bandit task[3,19,20], participants were presented with five buttons on the screen, similar to the task in Von Helversen et al. (2018)[11]. On each trial, clicking on a button revealed a non-negative payout. There were 4 incentivized blocks each with 40 trials (the practice block had 20 trials). The trial number, how many times each button had been selected, the average points per click earned from each button, and the total points earned so far were displayed on the screen. For every 100 points earned, participants received a bonus payment of £ 0.01. We considered two possible measures of the tendency to explore in this task. We included both measures because our simulations showed that the correlation between the measures can be negligible or negative, and the pilot data similarly showed a negative correlation though the 95% confidence interval crosses zero (see Supplemental Information).

The first measure, the one most often used in behavioural studies, is the proportion of switches between options, which intuitively captures parallel exploration of the available options[11,107,108]. This is calculated by taking the number of trials in which the selected option is different from the one selected on the previous trial and dividing this by the total number of trials in the block. Proportion switches were averaged across all incentivized blocks.

The second measure is suggested by one of the most studied choice algorithms in multi-armed bandit problems—the ε-greedy algorithm. According to the basic formulation of this algorithm, decision makers are supposed to select the choice with the largest obtained average reward, and, with a given probability, randomly sample the other options regardless of the obtained rewards[109]. Within this framework, exploitation is defined as the frequency with which an agent selects the option with the largest known average reward—that is, the proportion of trials in which the selected option is the one that has so far paid out the largest reward, on average. Therefore, a plausible measure of exploration is the complement to one of this frequency (i.e., 1 minus the proportion of trials in which the highest paying option has been selected; see ref. 110 for a similarly defined measure). This measure was also computed by block, and then averaged across the four incentivized blocks.

**Alien game.** In the present study, when participants played the alien game[25], which is the original variation of search tasks with NK landscapes[26,111,112], they were presented with ten buttons, each representing an attribute with a binary choice. In each trial, participants had to decide the binary choices for all ten attributes in order to receive a payoff. The ten attributes were displayed in one row and participants were told that each row of ten buttons/attributes represents a work of art that they are to sell to aliens. This original framing of the task aimed to reduce the role of priors that participants may have had when making their choices. The screen displayed the trial number, the payoff from each submitted row, and all prior combinations within the same block. A block consisted of ten trials (the practice block consisted of three trials) and the earnings for each block were the accumulated payoffs obtained from each trial of that block. Accumulated payoffs create opportunity costs for the participant, since additional new combinations resemble instances of exploration that may yield lower payoffs than prior combinations, thereby reducing the potential earnings for a block. This incentive scheme allowed for an optional stopping of search[27], since subjects settle on a configuration they perceive as good and therefore repeat until the end of the block. There were three incentivized blocks of the Alien Game and participants were told that different aliens would buy art in the different blocks, so that in any one block the same alien pays the same amount for the same combination, but across blocks another alien pays different amounts. For every 10 points gained in this game, participants earned a bonus of £ 0.02. The Alien game involved two measures of exploratory behaviour.

The first measure of exploration is defined by the Hamming distance[113], which is the measure commonly used in the different variations of the task[25–27,111,112]. The Hamming distance takes integer values between zero and ten, and is the number of attribute changes between the current combination and the best-performing prior combination—i.e. the minimum number of buttons that need to be toggled on or off to go from the best-performing prior combination to the current combination. The Hamming distance was averaged across all trials and blocks. The larger the Hamming distance (or search distance) the more exploratory the search. Thus, one measure of exploratory behaviour in the alien game was the average Hamming distance in the three incentivized blocks.

The second measure of exploration within each block can be called active search, a measure also used in the literature for this task[27]. Active search is given by the number of trials in which participants submit a new combination. In other words, when participants decide to repeat a combination (in the same block), they also stop exploration in this trial. Active search is thus the number of trials in each block in which the combination is different from all previous trials in that block (i.e., the number of unique combinations submitted in each block), averaged across the incentivized blocks. To calculate this, each submitted configuration was compared to all previously submitted configurations in the block; if there was a match then the submitted configuration was exploitation, if there was no match then the submitted configuration was exploration. And this was averaged across the incentivized blocks.

The two different measures are due to different conceptualizations of exploration[27]. Whereas the Hamming distance distinguishes between narrow and broad exploration (in other words, the Hamming distance looks at where people explore), active search, or the number of trials in which participants make changes compared to previous combinations, provides a measure of exploring vs. not-exploring (or, whether to search).

**Optional stopping with recall.** In the present adaptation of the optional stopping task[35,114], participants were presented with a 4 by 5 matrix consisting of 20 boxes on the screen each with a value drawn from a uniform distribution U(a, b), ranging from a = 0 to b = 10 points. Participants could search each box, one at a time, by clicking on it, revealing the value of that box. However, each time a participant opened a new box they paid a small cost ranging from 0.05 – 0.4 points (the cost was constant in each block but changed from one block to the next). The current best option, cost of opening each box, total costs for the block, and the trial number were all displayed on the screen. Participants could decide to stop at any time, at which point they received a payoff determined by the highest valued box that had been opened minus the total cost of the search in that block. Participants played one practice block and 8 incentivized blocks of the task. In this task, participants earned a bonus payment of £ 0.02 per point. Exploration was measured as the number of boxes that a participant has opened in each block, averaged across the 8 incentivized blocks[36].

**Sampling paradigm.** In the sampling paradigm[115], participants are presented with 2 buttons on the screen. In the present implementation, each button had an unknown probability of paying out a certain fixed number of non-negative points (or zero otherwise). Participants could choose to sample from each button as many times as they liked to observe its payoffs. When participants sampled from a button, a random draw was taken based on the payout probabilities of that button and they saw how much it would have paid. Participants could keep sampling from the two buttons, up to a maximum of 100 samples, before deciding which button they wanted to be paid out from. Participants had to sample from each button at least once before deciding on one. The screen displayed the trial number and the payoff from the sampled button. Once they decided on the button they wanted, participants could choose to be paid from that button. This ended one block of the task. Participants earned points determined by a random draw based on the payout probabilities of the button they chose. Participants played one practice block followed by 5 incentivized blocks. For each point earned, participants were paid a bonus of £ 0.05. Exploration is measured by the total number of times participants sampled from the two buttons in a block, averaged across the 5 incentivized blocks[17,29,116].

**Observe or bet.** In the observe or bet task[117], participants saw one lightbulb on their screen. On each trial, the light bulb turned on to be either red or blue (each block had set probabilities for how often the light turned on blue or red, and there was always a higher probability of the light turning one colour than the other) though participants did not necessarily see which colour it turned on. Prior to each trial, participants chose to either observe, guess red, or guess blue. Participants only saw what colour the light turned on if they chose to observe on that trial, but they could not win or lose any points when they observed. When participants chose to either guess red or guess blue in a trial, they were not shown what colour the light turned on in that trial. They could, nevertheless, earn a point if they guessed correctly or lose a point if they guessed incorrectly, and points were only displayed at the end of each block. During each block, the screen displayed the trial number, whether the participant observed or whether they guessed on the current trial, as well as the number of times each of the three options had been selected so far. Each incentivized block consisted of 50 trials (the practice block had 25 trials). Exploration is measured by the number of observe trials in each block averaged across the two incentivized blocks[39,54,117].

**Selection of behavioural tasks.** We identified several behavioural tasks from our respective fields of expertise and did a literature search to see whether these tasks and their variants are widely used across different subfields and whether researchers have used them to measure exploratory behaviour, sometimes drawing conclusions about human exploration tendencies in general. A main focus was to identify behavioural decision tasks that captured the core trade-off of exploration-exploitation dilemmas, but that also shared some structural similarities with one another. If there is a general tendency to explore, then it should best be captured by behavioural paradigms that have basic structural properties designed to capture such a tendency. Thus, we excluded tasks that did not satisfy this criterion.

Note that we have excluded tasks that have been used by previous studies but which are structurally substantially different. For example, we excluded foraging tasks because in these tasks people's choice of whether to explore or exploit is primarily driven by diminishing returns in terms of the resources people can accumulate from different foraging patches. In these tasks, the exploration-exploitation trade-off is thus inextricably intertwined with the effect of diminishing returns. Perhaps more importantly, Von Helversen et al.[11] used confirmatory factor analysis to examine the correlation between exploratory behaviour in a foraging task (the number of new patches visited by

participants) and in a 5-armed bandit task (the switch rate), finding a negative correlation after removal of outliers and a nonsignificant correlation otherwise. This finding suggests, as mentioned by Von Helversen et al., that structural differences between tasks may hide any domain general tendencies towards exploration. We chose to use the multi-armed bandit task because several of the other tasks that we also include are derivatives of the multi-armed bandit task (i.e., the sampling paradigm and observe or bet task), and because we are more familiar with these tasks, relative to foraging paradigms. Domain general tendencies should be more likely to emerge when comparing tasks that are more similar to one another, such as those mentioned. Therefore, although we use some of the theoretical reasoning developed to explain foraging behaviour, we've made design choices for the proposed study that we think are defensible and which offer a compelling overall research design.

We selected tasks that we thought might be more likely to be related to one another due to the similarities they have. This means that our test for domain-generality is somewhat more limited than it could otherwise be. Nonetheless, if there is a domain-general tendency to explore, and the measures typically used in various literatures capture that tendency, then we should observe the data patterns we've predicted among the measures that we've selected. If these data patterns aren't observed among our relatively homogeneous tasks, then we can be less confident in the ability of these tasks in capturing a domain-general tendency (or at least the summary measures typically used from them, such as the switch-rate in the bandit task). There may still be a domain-general tendency that is captured by foraging tasks and other exploration tasks, but that would be an empirical question for future research to examine. On the other hand, if we do observe the predicted data patterns, then we can be more confident in the idea that behavioural tasks and the summary measures used from them can capture a domain-general tendency to explore. Future research could then examine how far that general tendency stretches (e.g., whether it extends to foraging tasks). Moreover, researchers could examine whether the domain-general tendency extends to cognitive foraging tasks (e.g., information search, memory search, and/or anagram search). In addition, given that there is a diverse range of foraging tasks, it would be an interesting question for future research to examine the convergent validity of these relative to one another.

As another example, the secretary task has been often used to measure and draw conclusions about people's exploratory behaviour, but its objective is to find only the best option in a sequential search where options cannot be recalled. These properties of the task are different from all other exploration-exploitation tasks used in the literature. Instead, we opted to use the optional stopping with recall task, which has some similarities with the secretary task, because the goal is to find a good option in a sequential search process. In contrast to the secretary task, the optional stopping task is also structurally akin to other exploration-exploitation tasks as it allows for recall of previous options and has a reward structure in which the exact size of the reward makes a difference to people whereas in the secretary task this isn't the case. More broadly, we included tasks in our study that are representative of the broader literature. That is, the tasks we included share important structural properties with other tasks that we could not include but which have been used to study the exploration-exploitation trade-off either formally or empirically. For instance, we included the alien game in the present study because it involves exploring a multi-attribute space, a structural property that is common to other exploration-exploitation tasks used in the literature[2,3] but not shared by other tasks in our study.

The included tasks share some common underlying features: they all have multiple rounds, parameters that remain stationary across trials (though they all could be adapted to be dynamically changing), a finite known amount of options to explore in each trial, as well as a finite number of trials to create a trade-off between exploration and

exploitation. There are, however, other structural aspects that are similar between some tasks and different between others. Specifically, among the 5 behavioural tasks, the structures have similarities and dissimilarities in terms of whether: (i) there are only two or multiple options to explore; (ii) exploration and exploitation choices are clearly distinguishable; (iii) exploration involves direct or indirect costs that can result in stopping; (iv) they allow switching between exploration and exploitation multiple times; (v) exploring an option can completely resolve uncertainty about it; (vi) exploration involves search in a multi-attribute option space; (vii) there are rewards earned while people explore; (viii) there is a monetary explore-exploit trade-off; and (ix) they have an easily obtainable optimal solution. Table S8 displays which of the tasks have these structural qualities.

In an abstract sense, all tasks are to some degree isomorphic, or in some ways similar, to the multi-armed bandit task, with the optional stopping task and the alien game corresponding to boundary cases in which the variance of the different arms of the multi-armed bandit is zero, (all the uncertainty about the option is resolved when trying that option). Moreover, the alien game introduces multi-attribute exploration within a single trial, where participants can explore how different combinations of the attributes produce varying payouts, which no other task covers. The sampling paradigm and the observe-or-bet, also arguably variants of the multi-armed bandit, distinguish themselves from the latter by having clearly delineated phases of exploration and exploitation. That is, in the multi-armed bandit, clicking on the same button consecutively is considered as exploitation by the measure of exploration typically used (i.e., proportion of switch trials). However, these consecutive clicks may be exploration of the payoff distribution of that button. In contrast, in the sampling paradigm exploitation only occurs once a person makes a final choice, all previous clicks are part of the exploration phase. Similarly, in the observe-or-bet, when a person chooses to observe, they gain only information and so this is considered pure exploration, whereas when they choose to bet, they gain rewards without any information, and it is thus considered pure exploitation.

In the optional stopping task, costs of exploration are direct, whereas in the other tasks there are indirect time costs (e.g., sampling paradigm) and/or opportunity costs (e.g., multi-armed bandit). In the sampling paradigm and the optional stopping task, once a decision to exploit is made, the task ends (for that block) truncating the exploration phase, and thereby preventing multiple switching between exploration and exploitation phases. Exploration in the alien game and the optional stopping task completely resolves uncertainty, such that the value of the options becomes known after exploring them, a feature not shared with the other tasks. In the multi-armed bandit and alien game, people obtain rewards even while they explore the available options, whereas in the other tasks rewards are gained only after a decision is made to stop exploring. In the sampling paradigm, there is no monetary trade-off between exploration and exploitation, such that people can explore as much as they like without losing out on rewards; in the other tasks, choosing to explore means losing out on possible rewards that could otherwise be exploited. Last but not least, the optional stopping task has an easily obtainable forward-looking solution, which can be calculated by trading off the benefits from learning the exact value of an option and the cost of search. The optimal solution for the static version of the observe-or-bet task is computationally expensive to derive but nonetheless possible to obtain by solving the relevant Bellman equation[39]. By contrast, calculating the optimal strategy in the remaining tasks is either computationally extremely expensive or intractable altogether. When optimal strategies can be obtained, they can be also used to benchmark whether people's behavior is too exploratory, or not exploratory enough, in relation to the optimal strategy.

Our implementations were designed such that they covered the main instantiations of the tasks used in the literature. In addition, these implementations present various tasks with structural similarities but also differences (see Table S8), with some tasks clustered closer together (e.g., sampling paradigm and observe or bet task) than others (e.g., sampling paradigm and alien game). This provided a good test for (i) convergent validity, given the structural similarities between the tasks, and (ii) the domain-generality hypothesis, given also the structural differences (if all the tasks were too similar then it would be a weak test of domain-generality). Moreover, we made design choices to make the tasks aesthetically pleasing and engaging for participants.

Our measures of exploration, particularly for the behavioural tasks, are based on one approach used in past work and relies on summarizing people's responses across trials and blocks. Although the tasks in the present study were designed (e.g., the appearance of each task, number of trials and blocks, and scaling of payoffs) to reduce noise, the various summary measures may still, to some degree, deviate from each other because they could be influenced by factors other than a general tendency to explore. For example, behaviour in a task may be impacted by the task's optimal strategy (if there is one), the participants' personal goals, chance variation between participants in the observed payoffs leading to a decision point, and the sequence of the blocks of the task presented to each person. Importantly, behaviour in the present study's tasks have been, and can be, computationally modelled with parameters that represent a latent tendency to explore (e.g. refs. [118–120]). And such measures can be designed to account for the influence of the task specific factors on behaviour. Examining the convergent validity of measures from computational models is beyond the scope of the present study, but is definitely worthy of future research.

**Self-report scales.** For the self-report measures we focused on two independent literatures: (i) social and personality psychology and (ii) judgment and decision making. Questions in each self-report scale were presented together although the scales were presented in random order. For each scale measuring the tendency to explore, participants were instructed to, "Rate the statements below for how accurately they reflect the way you generally feel and behave. Do not rate what you think you should do, or wish you do, or things you no longer do. Please be as honest as possible". Each scale is presented in full in the Supplemental Information.

**Exploration scale.** The self-reported exploration scale is one facet of the Curiosity and Exploration Inventory II[13]. The exploration subscale has 5 items (e.g., "I actively seek as much information as I can in new situations"). These statements are rated on the following scale: 1 = very slightly or not at all, 2 = a little, 3 = moderately, 4 = quite a bit, 5 = extremely. The average rating across these items provides a measure of the tendency to explore, with higher scores representing a greater tendency.

**Maximization tendency scale.** The maximization tendency scale is designed to measure people's tendency to pursue the best option available[12]. The scale has 9 items (e.g., "I am uncomfortable making decisions before I know all of my options"). All items were rated on a scale from 1 = strongly disagree, to 5 = strongly agree; the other points were not labelled. The average rating across all items provides a measure of the tendency to explore, with higher scores representing a greater tendency.

**Alternative search.** The alternative search subscale of the maximization inventory is designed to measure the tendency to explore all possible options to increase the likelihood of finding the best one[14]. The alternative search scale has 12 items (e.g., "I can't come to a decision unless I have carefully considered all of my options", and "I take time to read the whole menu when dining out"). We adapted items related to shopping to also include internet shopping. All items were

rated from 1 = strongly disagree, to 6 = strongly agree; the other points were not labelled. The average rating across these items provides a measure of the tendency to explore, with higher scores representing a greater tendency.

**General explore-exploit question.** For the single-item measure of the tendency to explore, which we call the general explore-exploit question, participants were asked, "Are you generally a person who makes the most of what you know and have, or who explores new things and places, on a scale from 0 ("make most of what I know/have") to 10 ("explores new things")?". Answers range from 0–10 with higher scores reflecting a greater tendency to explore.

**Selection of self-report scales.** We included a measure of exploration from the social and personality psychology literature that's been widely used: the first version of the Curiosity and Exploration Inventory[42] has almost 1000 citations, according to Google Scholar, and the second version[13], which is the one that we used, has over 500 citations. We selected two measures from the judgment and decision-making literature, a literature which is largely focused on maximizing and satisficing, to obtain some representativeness regarding disagreements about how the tendency to maximize should be measured. One measure of maximizing and satisficing that we included is thus a unidimensional measure[12], and the other is one facet of a multi-dimensional measure[14]. We selected the latter because past research has found contradicting evidence regarding its convergent validity with exploration behaviour in the sampling paradigm[12,44]. In addition, we devised a single-item self-report measure aimed at capturing a general tendency to explore-exploit. We based the wording of this measure on the wording of the general risk preference item that past research has found to correlate with behaviour in some risk taking tasks[74,121].

## Sampling plan

**Participants.** Given that we needed a minimum of 632 participants (see Analysis Plan, below), we aimed for a total of 700 participants with complete data for all measures (after exclusions). We overshot the minimum sample size requirement to maximize statistical power for our tests within feasible limits. Anticipating dropouts and exclusions we therefore recruited 750 participants for the proposed study, using Prolific.ac (an online survey platform). We did not replace participants who were excluded based on the pre-registered exclusion criteria, meaning that our final sample of participants with complete data for all measures could have been more or less than 700. However, if the number of participants included in any of the analyses was below 632 (e.g., due to an unforeseen high number of exclusions), we intended to recruit another batch of participants: For every 30 participants below 632, we intended to recruit another batch of 50 participants (Prolific provides this service while maintaining the representativeness of the sample). For example, if the final sample from the original batch of participants was 580 (e.g., because many participants failed the comprehension checks), which is 52 participants fewer than the required minimum of 632, then we would recruit another batch of 100 participants. The aim was so that the final sample for the analyses would be at least 632. We used a representative sample matched on age, sex, and ethnic group from the UK. Participants were paid £6.00 for completing the full study (which we predicted would take ~60–70 min), plus a bonus payment that was contingent on their performance in the behavioural tasks and which was expected to amount to an average of £5.00 per participant.

For Hypothesis 3 (on test-retest reliability of the measures), we required some participants to take part in the study a second time, 1 month after their first participation. There is no strict threshold for test-retest reliability, and we intended to estimate the reliability of the measures rather than test them against an a priori criterion.

Importantly, to test the measurement model derived from the factor analyses in Hypothesis 2 for measurement invariance, we invited only those participants whose data were used in the confirmatory factor analysis at Time 1 (which we expected to be ~350; i.e., half of the sample) to do the study again at Time 2. Participants at Time 2 were paid the same base pay and contingent bonus. We anticipated that around 300 participants would complete the study on this second occasion and satisfy the inclusion criteria.

**Inclusion/Exclusion criteria.**

1. We had three main criteria for excluding/including participants from/in analyses.

   Only participants who completed the full study would be included in the final analyses because incomplete data may mean a lack of motivation resulting in dropout—we wanted to use data from participants motivated to complete the study. Specifically, participants only received performance-based bonuses if they completed the entire study. And monetary incentives are known to create differences in effort and to change behaviour in some judgment and decision tasks[122,123]. This is particularly important for tasks that participants can choose to end without putting any effort in exploring the different options (i.e., sampling paradigm, optional stopping task). Participants who drop out may, furthermore, have been less motivated to do well in the tasks they completed, particularly if they drop out intentionally due to having other demands on their time during the online experiment. Therefore, the difference in receiving vs not receiving incentives, in addition to likely differences in the motivation of participants to complete the study, may add noise to the data and thus reduce the statistical power of the analyses.

2. Participants had up to 6 attempts to correctly answer the comprehension check questions for each task before they could continue with that task. Each task had an elaborate set of instructions, and it was important that these instructions were understood so that the noise in our measures of exploration was minimized.

   a. Participants who answered the questions incorrectly on 6 attempts for any one task were excluded from analyses involving only that task for Hypotheses 1 and 3 (as they would be automatically moved past that task and onto the next task); they could continue with the study and observations from them on the other measures were included in all other analyses for Hypotheses 1 and 3. Allowing fewer attempts would have risked dropping participants who may have understood the task if given sufficient time, but allowing more attempts would have risked including participants who got the answers correct through trial and error.

   b. Participants with 6 incorrect attempts for any one task were excluded from all analyses for Hypothesis 2 (i.e., we will use listwise deletion of these participants' observations for Hypothesis 2), as this method of deletion is a simpler approach for factor analyses.

3. In case there were any instances of multiple responses from the same participant (i.e., the same Prolific ID), we intended to include only the first response in the analyses and exclude subsequent responses; if in the participant's first response the study was not completed until the end then all observations from that participant would be excluded.

## Analysis plan

**Positive controls and reliability analyses.** The maximization tendency scale and the alternative search subscale from the maximization inventory should be positively correlated. Indeed, this is shown in past research ($r = 0.38$)[14] and in our pilot data ($r = 0.8$). Therefore, as a

positive control in our main analyses, we expected that the maximization tendency scale and the alternative search scale would be positively correlated by at least $r = 0.3$. If the correlation between these two scales was too low then the data on the self-reports would be suspect. In that case we intended to conduct the correlational and factor analyses with and without the self-reports as a type of sensitivity check and draw conclusions accordingly. As can be seen in the results section, the maximization tendency scale and alternative search subscale were correlated at $r = 0.55$. Therefore, this positive control was satisfied.

**Internal consistency.** We used Cronbach's alpha to assess internal consistency. For the behavioural tasks, we calculated Cronbach's alpha using each block of a task as an observation point (like an item on a self-report scale). Kline (p. 92)[124] states that Cronbach's alpha values of about .70 may be considered as adequate but that these classifications are merely "some guidelines". Importantly, such guidelines are typically derived from researchers' experience with self-reports, and self-reports are prone to many types of common methods variance (e.g., consistent responding, social desirability, acquiescence responding, etc)[82] that spuriously inflate inter-item correlations and thus Cronbach's alpha[125]. In contrast, behavioural tasks have not been shown to be consistently prone to these types of method variance, and though they may be prone to other types of common method variance, we think it unlikely that the correlations between blocks of a task would be as spuriously inflated as the correlations between self-report items. It may be that some of the behavioural tasks will have lower internal consistency due to stochasticity (e.g., luck of the draw), though we made design choices to mitigate this issue. Moreover, the measures we used, including the aggregation of the behavioural measures across trials and blocks of a task, are as they are used in the literature. It is important to examine whether the measures, as used in the literature, have convergent validity with one another. Taken together, therefore, instead of using cut-offs to modify the measures or analysis plan, we intended to temper our conclusions to take account of any results that showed a measure to have dramatically low internal consistency relative to the other measures. Specifically, very poor internal consistency of a measure would put an upper bound on how strongly the measure could correlate with other measures. We intended to further conclude that a measure with poor internal consistency should be treated with caution as it may be indicative of a poor measure. As can be seen in Table 1 in the results section, the Cronbach's alphas suggest good reliabilities.

**Measurement invariance.** Because we had measurements at two different points in time for a large group of our participants (see Hypothesis 3), we could test for measurement invariance in regards to time. There are two ways to achieve that: either testing each task individually for measurement invariance, or by testing the structural equation model derived from the factor analyses. We opted for the second approach because it would provide important information about the measurement model selected from the factor analyses.

We intended to test the model selected from the factor analyses for Hypothesis 2 for different levels of invariance to time using the differences between fit indices of the configural, weak factorial, strong factorial, and strict variance models. Specifically, the fit indices we used were ΔCFI, ΔGamma hat, and ΔMcDonald's NCI[126]. As decision criteria, a ΔCFI larger than −0.01, a ΔGamma hat larger than −0.001, and ΔMcDonald's NCI larger than −.02 indicate that the null hypothesis of invariance should be rejected and that the measurement model is thus not invariant across time. If at any level we could not establish invariance for the entire construct, we intended to use the bias-corrected bootstrap confidence intervals approach to identify and remove any problematic tasks and try to establish partial measurement invariance for the remaining tasks at the same level. The bias-

corrected bootstrap confidence intervals approach to measurement invariance is centred on evaluating the confidence intervals for all the differences between parameters in the model. If a confidence interval does not include zero, it can be inferred that the specific parameter is not invariant.

For example, if we could establish weak factorial invariance for the entire model, but not strong factorial invariance, we could remove any tasks that do not satisfy strong factorial invariance in order to establish partial strong invariance for the model in the remaining tasks. If we could not establish even partial measurement invariance for the full measurement model, then we would assume that the model is unlikely to be temporally stable. We intended to, nonetheless, examine the test-retest reliability of each measure using the pairwise correlations for Hypothesis 3.

**Hypothesis 1.** Hypothesis 1 consisted of the main tests for which we powered our analyses and calculated the required sample size. There was a total of 55 pairwise correlations between each combination of the 11 measures tested in Hypothesis 1. The corrected alpha for each test was (0.05/55) 0.0009. We conducted statistical power analyses using G power[127] for the Pearson correlation tests against zero (Hypothesis 1) and the R TOSTER package[128] for the equivalence testing. Prior to running any analyses, we followed the preregistered inclusion/exclusion criteria (see Sampling Plan), and then calculated the measure of exploration for each task and scale as described above (see Design).

To test the hypothesis that each measure of exploratory behaviour would have some convergent validity such that it would be positively correlated (i.e., $r > 0$) with the other measures (Hypothesis 1; see Table S11), we conducted one-sided correlation tests against a null hypothesis of zero. For the reasons provided earlier, we took a correlation of $r = 0.2$ as the minimum necessary correlation for convergent validity for which we powered our analyses.

For each test to have 95% power to detect a correlation of at least $r = 0.2$, with the corrected alpha of 0.0009, we required a minimum sample size of 556. To test whether the correlation between any two measures of exploration is too weak for convergent validity (i.e., $r < 0.2$), particularly if the test for Hypothesis 1 is nonsignificant, we used equivalence testing with the upper bound of $r = 0.2$ and lower bound of $r = −1$ (given that we were interested in a one-sided test). The equivalence tests would thus determine whether the observed correlation is statistically smaller than $r = 0.2$ and larger than $r = −1$. At the corrected alpha level, for 95% power to reject effect sizes $r \geq 0.2$, we required a sample size of 632 for each equivalence test which defined the minimum sample size for the study. We conducted the equivalence tests only if the corresponding correlation test was statistically nonsignificant. (Negative correlations that are statistically significant at the corrected alpha level indicate convergent invalidity between the two relevant measures.)

The anticipated final sample size of 700 (see Sampling Plan) provided the correlation tests with 98.7% power to detect effect sizes at least as large as $r = 0.2$ (95% power to detect $r = 0.18$) and the equivalence tests with 97.4% power to reject effect sizes $r \geq 0.2$.

**Hypothesis 2.** Our analyses for testing Hypothesis 2 included three preregistered baseline confirmatory models and three exploratory data-driven models. For these analyses, we randomly divided the data into two roughly equal parts (expected to be ~$N = 350$ each). With one half of the data, we first derived three exploratory data-driven models and did a preliminary check of how well the confirmatory models fit the data. With the other half of the data, we then compared all six models (the three baseline confirmatory models and the three data-driven models) against each other with confirmatory factor analysis (CFA). For these analyses, we used the aggregated scores of each measure as described in the "Design" section (e.g., the switch-rate

aggregated across the different blocks of the multi-armed bandit task, and the mean ratings on the items of the exploration scale).

The baseline models represent three basic theoretical assumptions and are depicted in Figs. 2–4. In the first baseline model, Baseline1, the explained variance is accounted for by only one common underlying factor (i.e., the tendency to explore). In Baseline2, the explained variance is accounted for by only two specific oblique factors reflecting the methodologies of measurement (i.e., behavioral and self-report measures). In Baseline3, the explained variance is accounted for by two orthogonal factors reflecting common method variance for the behavioral tasks and the self-report measures, and one common underlying factor representing the tendency to explore. In the first exploratory model, Exploratory1, we used both exploratory graph analysis and parallel analysis combined with exploratory factor analysis to derive a simple structure model. In Exploratory2, we intended to have a model in which the explained variance accounted for one common underlying factor representing the tendency to explore in addition to the factors derived for Exploratory1. In Exploratory3, we expected to select between bifactor exploratory models with a single bifactor according to the interpretability of the largest loadings. Now we describe the models in more detail, including how the data-driven models were expected to be derived, and then we detail how we would do the model comparisons in the CFA.

**Baseline1.** The first baseline model assumes a general factor representing the tendency to explore, thus excluding any methodological effect of the elicitation methods of explorative behavior. Figure 2 visualizes Baseline1.

**Baseline2.** The second baseline model, visualized in Fig. 3, assumes the existence of two specific factors, one associated with the self-report measures and the other associated with the behavioral measures. Specifically, the explained variance is accounted for by only two specific oblique factors reflecting the methodologies of measurement (i.e., behavioral and self-report measures). This model thus assumes that the tendency to explore is expressed differently in the two broad categories of measures, best captured by two separate factors instead of one. The two factors are allowed (and expected) to be correlated with each other.

**Baseline3.** The third baseline model, visualized in Fig. 4, relies on the assumption that the different measures are affected by a single common factor reflecting the general tendency to explore, but part of the observed variances are also explained by common method variance. Specifically, the explained variance is accounted for by two orthogonal factors reflecting common method variance for the behavioral tasks and the self-report measures, and one common underlying factor representing the tendency to explore.

**Exploratory1.** The first exploratory model would be derived using two methods for assessing the dimensionality of an instrument (i.e., the inferred number of latent common causes): (i) exploratory graph analysis[129] and (ii) parallel analysis combined with exploratory factor analysis. Exploratory graph analysis would tell us the number of factors that could be extracted and also the measures that pertain to each factor. In contrast, parallel analysis would only tell us the number of factors that could be extracted, but not which measures pertain to which factors. Therefore, in a second step after the parallel analysis, we would use exploratory factor analysis with the number of factors predetermined by the parallel analysis. In the exploratory factor analysis, we would consider a measure to be derived from a factor if its absolute loading on the factor was at least 0.30−this threshold is within the bounds of what has been recommended[130] and done[74] in the literature. The model thus derived from the combination of parallel analysis and exploratory factor analysis would be compared with the

model derived from the exploratory graph analysis. In this comparison, the most interpretable between these two models would be selected as Exploratory1. Interpretability here refers to what makes sense from a theoretical perspective. Essentially, we would choose the model with the most parsimonious theoretical justification. The theoretical justification would be based on, for example, the structural similarities and differences between the tasks noted in Table S8. The selected model would be Exploratory1 and the factors would also be used as one component of the Exploratory2 model.

**Exploratory2.** The second exploratory model assumes that there is a single general factor that captures people's tendency to explore, as with Baseline1. However, we intended to also keep the factors identified in Exploratory1. Exploratory2 would therefore be a bifactor model with a general tendency to explore that loads to all measures but that also includes the factors identified in Exploratory1 loading to their respective measures.

**Exploratory3.** The third exploratory model would be derived using exploratory bifactor analysis[131,132]. Because there is no dimensionality approach for bifactor models, we intended to select the "best" model from the exploratory bifactor analysis based on its interpretability regarding the loadings of the tasks to each extracted factor. As described for Exploratory1, our model selection based on interpretability depended on what made the most sense from a parsimonious theoretical perspective. The selected model would be Exploratory3.

**Confirmatory factor analysis: model comparisons.** By now, we had used a random half of our data to derive the above exploratory models and preliminarily examined the confirmatory models. We then aimed to use the remaining random half of our data to fit the six models described in the preceding paragraphs (i.e., Baseline1, Baseline2, Baseline3, Exploratory1, Exploratory2, and Exploratory3) and compare them using CFA. For the EFA-derived models (i.e., Exploratory1-3), absolute loadings below 0.30 would be fixed as 0 (which is equivalent to not including them in the factor). If the data from the tasks were normally distributed, parameter estimation would be done with robust Maximum Likelihood estimation (MLR). If the normality assumption did not hold, parameter estimation would be done with the Weighted Least Square Mean and Variance Adjusted Estimators (WLSMV)[133].

The models would be compared with regard to fit indices commonly used in the context of model comparison in factor analysis: CFI; TLI; and RMSEA. The best fitting model would therefore be the one with CFI and TLI closest to 1 and RMSEA closest to 0. If two or more models provided very similar fit to the data, the model we selected would be the one that made the most sense from a parsimonious theoretical perspective. For instance, Baseline3 may have had better fit indices than Baseline1, even if Baseline1 showed good enough fit indices. In this case, the latter would be the most parsimonious from a theoretical perspective. That is, it would be reasonable to assume that the unidimensional model (i.e., Baseline1) is a better description of the data. The selected model would also be evaluated for measurement invariance using the analysis plan described earlier.

If the best fitting models in the CFA were those in which no two behavioural measures, or no combination of behavioural and self-report measures, fit well onto the same factor, or if we failed to identify any such models that fit the data well, then we would conclude that we did not find evidence for a latent variable that captures a domain general behavioural tendency across tasks. There would be evidence for a domain-general tendency to explore, $E$, to the extent that multiple behavioural measures (or a combination of behavioural and self-report measures) fit well onto a common factor together. The strongest evidence for $E$, and the validity of the measures in capturing it,

would be if a single factor could be extracted that explained variance in all of the 11 measures of exploration.

**Hypothesis 3**. To test the hypothesis that each measure of exploratory behaviour has test-retest reliability (Hypothesis 3; see Table S11), we used correlation tests for each measure separately to examine the relationship between exploration at Time 1 (T1) and exploration at Time 2 (T2), one month later. The strength of these correlations provides an estimate of each measure's test-retest reliability. We used one-sided correlation tests against a null hypothesis of zero. As described in the sampling plan, we estimated an approximate sample size of 300 for these tests. Given that each measure is a family of tests unto itself, there were no corrections for multiple comparisons, with the anticipated sample size giving the one-sided tests 95% power to detect a correlation of $r = 0.188$.

### Participants

Given the above sampling plan, from April 20th to 28th, 2023, we recruited 750 participants of whom 580 completed the full study (this was partly due to a server problem in the early stage of data collection which increased the number of participants who dropped out of the study prior to completing it). We therefore recruited another 100 participants. The final sample of participants who completed the full study was 678. These had a mean age of 44 years ($SD = 15$) ranging from 18–82 years, and reported their gender as either female ($n = 350$), male ($n = 323$), or other ($n = 5$). Participants earned an average bonus of £4.47 across the behavioural tasks in addition to the £6.00 base payment. The median completion time was 49 min (range, 18–1051 min)— 1051 min was the largest observed value, with the next largest being 198 min. The results remain unchanged even after excluding the extreme outlier who took a very long time to complete the study. In line with the inclusion/exclusion criteria, of those who completed the full study, we excluded 27 participants for the analyses of Hypothesis 2 because they failed the comprehension check questions on the 6th attempt and thus did not complete at least one of the behavioural tasks (6 in the Alien task, 7 in the Bandit task, 6 in the Observe or Bet task, 0 in the Optional Stopping task, and 10 in the Sampling task; only one participant failed multiple comprehension checks—specifically, in the Alien, Observe or Bet, and Sampling tasks). For Hypothesis 1, we included these participants in the analyses for the behavioural tasks that they did complete.

For Hypothesis 3, we invited the participants in the confirmatory half of the factor analyses testing Hypothesis 2 to do the study again ~1 month later at T2 (from May 29th to June 2nd). Of the 339 participants we invited, 254 participants completed the study at T2, with mean age of 46 years ($SD = 15$), ranging from 18 to 82 years, of whom 129 were female, 123 male, and 2 reporting gender as other. Participants earned an average bonus of £4.52 across the behavioural tasks in addition to the £6.00 base payment. The median completion time was 34 min (range, 14–122 min).

### Reporting summary

Further information on research design is available in the Nature Portfolio Reporting Summary linked to this article.

## Data availability

The anonymized data (both for the pilot and main studies) have been deposited in the Open Science Framework database [https://doi.org/10.17605/OSF.IO/F62MY]. The pages have been made publicly available. The materials are available in the Supplemental Information.

## Code availability

The R code for analyses (both for the pilot and main studies) have been deposited in the Open Science Framework database https://doi.org/10.17605/OSF.IO/F62MY. The pages have been made publicly available.

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

## Acknowledgements

This work has been mainly funded by a grant from the Danish Council for Independent Research (DM, DFF–7015–00050). The remainder of the funding was provided by the Department of Business and Management and the Business and Social Science Faculty at the University of Southern Denmark. Farid Anvari was supported by funding from the European Union's Horizon 2020 research and innovation programme under the Marie Sklodowska-Curie grant agreement No 883785. The funders have/had no role in conceptualization, study design, data collection, analysis, decision to publish, or preparation of the manuscript.

## Author contributions

F.A., S.B., and D.M. equally contributed to the conceptualization of the study, and P.A. provided feedback later to improve conceptualization. F.A., S.B., D.M., and P.A. contributed to the study design. F.A. wrote the first draft. F.A., S.B., D.M., P.A., and V.R.F. all contributed to editing and revising the draft. F.A., S.B., D.M., P.A., and V.R.F. all contributed in creating the analysis plan. D.M. and V.R.F. developed the analysis code with feedback and adjustments provided by F.A., D.M. and S.B. acquired the funding for the project.

## Competing interests

The authors declare no competing interests.
