## [Peer Review File · Nature Communications]

nature portfolio

Peer Review File

Testing the convergent validity, domain generality, and temporal stability of selected measures of people's tendency to exploreEditorial Note: This manuscript was submitted to Nature Communications as a Registered Report. The rebuttals for pre-study and post-study are indicated within this peer review file.

Review Study Plan (Stage 1)

Reviewer #1 (Remarks to the Author):

The authors propose a correlational study to investigate human exploration behavior. More specifically, they aim to compare exploration behaviour between different experimental tasks and questionnaires to study whether exploration behaviour in different situations is supported by a domain-general tendency. In addition, the authors propose to study whether people's tendency to explore is stable across time.

The manuscript is well written, presents an important research questions, a clear rationale, methods section and analysis pipeline. I really like the proposed study, but have two main comments/concerns (for more detail. see comments below). First, the main aim of the proposed study is the investigating of the (convergent) validity of different measures of human exploration behaviour. This is fine, but the authors should also introduce and, at least briefly, discuss the theoretical accounts of exploration-exploitation trade-offs and behaviour. The second point is related to the domain-general hypothesis and the selection/exclusion of exploration tasks. Without a more detailed discussion of what theories/mechanisms underpin the exploration-exploitation trade-offs in the selected/excluded tasks, it is difficult to assess the significance of the proposed study and unclear how results might generalize to other tasks. Finally, I would like to learn more about the reliability with which one can assess people's exploration tendencies in a series of experimental tasks, each of which has to be carried out in just a few minutes.

Detailed comments:

- The authors mention paradigms and tasks used to study exploration-exploitation behavior including tasks "designed to mimic foraging" in the introduction. There are a few studies in which participants' task was to move through space and collect patchily distributed resources (at least in virtual environment e.g. Hills et al., 2013; Wolfe, 2013). Given that foraging theories primarily aim to explain actual foraging behaviour, I believe these studies should be cited.
- The authors do not review foraging theories that aim to explain exploration-exploitation trade-offs. While their study is not aimed at investigating these theories directly, I believe they should be introduced briefly to provide the theoretical context.
- The authors excluded foraging tasks as the exploration-exploitation trade-off in these tasks is inextricably linked with the effect of diminishing returns. I do not quite understand why this would disqualify foraging tasks from the current study, particularly because theories of exploration-exploitation trade-offs have originally been developed to explain foraging behaviour. The authors should provide a more in-depth explanation for why they believe that domain-general tendencies for exploration may not be predictive for foraging tasks.
- Related to my last comment, I wonder about the domain-general hypotheses, and how the selection of the experimental tasks affects how strong the test of domain-generality is. I believe the authors should make explicit what the exact properties of everyday exploration-exploitation trade-off situations they are interested in (or which aspects their study design is able to address) and what aspects/mechanism are beyond the scope of the proposed study. For this discussion, a more detailed explanation of the underpinning theoretical framework would be helpful.
- In the last comment I mentioned that a more detailed explanation of the theoretical underpinning of exploration-exploitation behavior would be useful. This would also help with understanding how the different exploration measures for the alien game fit with the overall aim of the study.
- The authors estimate that it will take participants approximately one hour to complete the study. In this hour, they will need to complete five different exploration-exploitation tasks, fill out questionnaires, provide informed consent, etc. This leaves little time for each of the task. However, to obtain reliable estimates of participants' exploration tendencies, participants will not only have to understand the tasks well, but will also have to have sufficient experience with the tasks to understand costs and potential benefits for different options in each task well. I understand that the authors have piloted their approach and that neither tasks results in floor- or ceiling

performance. However, my question is whether (and why) the authors convinced that they will be able to measure peoples' exploration tendencies reliably using such short tasks?

Reviewer #2 (Remarks to the Author):

A large number of experimental paradigms in the behavioral sciences require participants to engage in both exploration and exploitation (e.g., the multi-armed bandit task). These tasks are often assumed to measure some general tendency of an individual to explore their environment that is stable across time. Surprisingly, few studies have examined whether this is true, i.e., whether the construct of "exploration" is in fact a stable, domain-general trait, and whether there is convergent validity among the variety of tasks used in the behavioral science that purportedly measure this trait. The authors propose to test these claims with a large sample on multiple exploration paradigms, and analyzing the data using pairwise correlations and equivalence testing, factor analysis, and test/re-test reliability at two spaced time points.

Overall, I found the manuscript well-written and well-structured. The authors' hypotheses have important implications for the field, and are relevant given the increasing interest in exploration/search tasks in the past decade. I also think the study is perfectly suited for this two-stage review process: while I have personal doubts about whether the tasks do engage some domain-general and stable exploration trait, the results will be informative regardless of the outcome. (If it turns out that the null hypotheses cannot be rejected, it would certainly be harder to publish the results using a traditional review process that occurs after data collection and analyses). The proposed procedures and analyses are clear enough to replicate the study. The analysis techniques are appropriate, and I commend the authors for commitment to avoid overfitting the data (e.g., pre-hoc Bonferroni adjustments in H1, cross-validation to use exploratory and confirmatory factor analysis in H2). I recommend accepting the manuscript for Stage 1, and look forward to seeing the results. Below, I provide several comments that the authors may want to consider. I believe all of these comments to be relatively minor. I hope the authors give them serious consideration, but leave it to the other reviewers and editor as to whether any of them should be required.

1. The manuscript (particularly the Introduction) makes frequent reference to a number of exploration tasks (e.g., the eight listed on lines 58-62). The five tasks used by the authors are described late in the paper, and tasks not used by the authors are described very little. Yet a number of comparisons are made, and much of the discussion in the Introduction requires an understanding of how these tasks work and the differences between them. I suggest the authors create a Table, referenced in the Intro, which has the task name (in one column) and a brief, 2-3 sentence description of the task from the participant's perspective (in another column). The five tasks used in the current study should be clearly identified (e.g., listed above the unused tasks). This would help make the manuscript more self-contained, so that the reader does not need to consult other work in order to understand the claims.

2. The exploration measures used by the authors are all data-based (as opposed to model-based). However most (if not all) of these tasks have been modeled computationally, in some cases with explicit parameters that represent a latent exploration trait (e.g., the softmax temperature parameter in a multi-armed bandit task). In contrast, the data-based measures are noisy measures of exploration because they are influenced by other factors, such as the participant's goals, the task's optimal strategy, the participant's subjective utility, or differences in observed data across participants leading up to a decision point (due to chance). Examining model-based parameters seems like an interesting future direction, but a large endeavor that can be justified as outside the scope of the current manuscript. However I believe it would be helpful to include a short discussion on the topic and acknowledgement that the proposed exploration measures reflect more than just exploration.

3. The procedure indicates that the self-report scales will be completed immediately after the behavioral tasks. A concern is that participants will answer questions on these scales by thinking

about their performance on the previous tasks. ("Am I uncomfortable making decisions before knowing all of my options? Well my behavior on the previous task suggests that I am..."). In other words, participants may be describing their behavior on the task rather than revealing some trait that actually predicts behavior. Intuitively, I feel like the converse is less likely (I don't expect participants to modify their behavior in the behavioral tasks to match their answers to the scales). Perhaps it would make more sense for participants to complete the scales first, or to counterbalance the order?

4. (line 242) The Methods section begins with a discussion of the Alien task, which is not mentioned in the Introduction. Considering its importance in the pilot data and its use in the experiment, it should be discussed in the Introduction as well – otherwise the transition seems a little odd (you discussed 8+ exploration tasks... and then choose a completely different one to use for this study?)

5. Similar to point #1, I found the detailed discussion of some tasks in the Methods to be confusing because a number of comparisons are made before the tasks are actually described. One possible re-organization takes the discussion of 'excluded tasks' or 'task selection criteria' (lines 318 - 388) and places it in its own section after the description of the tasks used (i.e., moving it to around line 495). Similarly, the section on self-report scales begins obliquely by referring to "the first version of the scale" (What scale? Does it have a name?), followed by an actual description of the scales (lines 514 - 536). It might aid the reader if the scale descriptions were provided at the beginning of that section.

6. The payoff structure in each task is stationary, but are participants aware of this? It may not be obvious to naive participants, who could adjust their behavior throughout the experiment (e.g., one might believe that the first arm in a bandit task has the highest expected value, but after a sequence of bad outcomes they might believe that the experimenter "switched" the arms on them). Normatively, a participant should explore more when the payoff structure is dynamic compared to when it is stationary. In some tasks, such as the Iowa Gambling Task, the instructions intentionally emphasize that the payoff is stationary (i.e. "there are two good decks and two bad decks..."). It may be worth probing this belief in the instructional check questions, to ensure participants understand.

7. I found the Alien game to be a bit confusing because of its multi-attribute nature. The first measure of exploration for the task is how far apart two consecutive sequences are (Hamming distance). Which is a reasonable measure if you believe that each binary choice in the sequence is independent, and that payoffs are based on features and not relations (e.g., it could be that alien enjoys particular bigrams, so each is not independent). As with point #6, I think this aspect could be probed in the instructional check questions.

Typos / grammatical

- (abstract) "Upto" -> "Up to"

- (line 620) "We'll" -> "We will"

- (line 142) Do you mean "conservative" or "liberal"? Setting a low cut-off value ($r = .2$) means that you are more likely to obtain false positives than if you were to set a high cut-off value. A "conservative threshold", in my opinion, is one that minimizes the number of false positives by setting a high bar. Perhaps I'm in the minority in my interpretation, but I found the word choice to be at odds.

Reviewer #3 (Remarks to the Author):

In this registered report, the authors ask whether different experimental measures of exploration (Q1) capture similar (individual-specific) qualities, i.e. testing convergent validity, (Q2) can be attributed to a common task-independent but the individual-specific quality of 'tendency to explore', i.e., testing domain generality, and (Q3) are stable over time, i.e. testing temporal stability.

To do so, the authors plan to complete 5 carefully chosen behavioral tasks and 4 self-reports with a massive number of participants -- while a number of participants will redo the task after one month. The plan is to extract 11 measures for exploration (7 based on behavioral tasks and 4 based on self-reports): then, (Q1) the convergent validity is going to be tested by studying the cross-correlation between different exploration measures, (Q2) the domain generality by conducting common factor analysis on these 11 measures, and (Q3) the temporal stability of each measure by studying its self-correlation between the 1st and the 2nd round of the experiment.

I believe that the answers (either negative or positive) to these 3 questions (Q1-3) are crucial for re-evaluation of the common methodologies and assumptions in studying human exploratory behavior and will help both experimental and computational scientists in different fields (including psychology and neuroscience) in designing, analyzing, and evaluating new experiments, theories, and computational models. Moreover, collecting and publishing such a rich dataset is going to be an important contribution to the field.

The author's choice of experiments and their analysis plan for answering these questions sound relatively promising, but I have some major comments regarding the generality of the study, the interpretability of possible outcomes of the analysis related to Q1 and Q2, the choice of exploration measures, and the details of statistical analyses. I will explain each point in detail below.

Overall, I enjoyed reading the report very much, and I am looking forward to reading its revised version.

1. The generality of the study:

Through Abstract and Introduction, there is an implicit claim that the authors want to study the general notion of the "tendency to explore". However, in Methods, it becomes clear (c.f. lines 318-337) that, based on the choice of behavioral tasks, they study the "tendency to explore" as in the "trade-off between exploration and exploitation" (also see * and ** below). However, exploration can be studied even in the absence of reward (e.g. [1]) or when there is no trade-off (e.g. [2]). I think this limitation should be explicitly mentioned when the questions (Q1-3) are asked in the paper. I believe that even with this limitation, the study is quite general and the questions are very important, but making the assumptions and limitations explicit puts the study in the right context and helps to avoid over-interpretation of the results.

* In contrast to the behavioral tasks and the self-reports 2 to 4, the self-report 1 (i.e. "Exploration scale") is more related to exploration as in curiosity-driven behavior (c.f. [3]) than to the exploration and exploitation trade-off -- at least that is what I understand from the questions quoted in supplementary materials, e.g. the statement "I view challenging situations as an opportunity to grow and learn". I think this should be taken into account when interpreting the final results.

** : In the alien game (AG) and the sampling paradigm (SP), the trade-off between exploration and exploitation is NOT in terms of monetary payoff: in fact, in these two tasks, the more participants explore, the more they will be paid without losing anything other than their time and energy -- in contrast to the other tasks. I think this should also be taken into account when interpreting the final results.

2. The interpretability of possible outcomes of the analysis related to Q1 and Q2

If all 11 exploration measures turn out to be significantly and highly correlated with each other and if it turns out that there exists a model with a single common factor that can explain data best, then the interpretation is straightforward. However, I wonder what would be the authors' plan if it turns out otherwise?

In particular, I can imagine observing a few clusters of positively correlated exploration measures in response to Q1 and similarly finding more than one common factor in response to Q2. I agree that such an observation by itself is interesting and valuable, but I believe that a more careful choice of exploration measures will make interpreting such results easier.

For example, it has been argued that exploration can be decomposed into two components of "random" exploration (RE) and "directed" exploration (DE) [4] and that the DE can have different motives [1,2,5]. It is not trivial whether the tendency for RE is correlated with the tendency for DE. If it is not the case, then it is easy to imagine a model with more than one common factor to have higher explanatory power than any model with a single common factor. If exploratory measures are interpretable in the sense that they are closer to RE or DE, then it may be easier to interpret the common factors (see a particular suggestion of mine in point 3).

Therefore, I suggest adding explicit reasoning for why each measure of exploration is added and what it is supposed to measure -- similar to but more detailed than the comparison between the two measures of the Alien game in lines 449-453. Then, even if all 11 measures turn out to be correlated, the result would be more interesting as it would count as evidence for the similarity of conceptually different measures.

3. The choice of exploration measures in behavioral tasks

While I think the choice of behavioral tasks is very clever (as represented well in Table 2), I think the choice of exploration measures can benefit from some theory-driven arguments. I make this more clear below.

3.1. First, It should be clearly mentioned which measures are designed by the authors and which are the ones taken from the previous studies -- for example, it is not clear whether the hamming distance in the Alien game (AG-Hamming) has been used before or not. Second, I suggest (i) keeping the previously used measures as they had been defined in the paper potentially only for control) and (ii) modifying the ones that were designed by the authors and adding new measures with more theory-driven arguments. Another table can be added representing this information. This is in particular important for answering Q2.

I give some suggestions below and point to the problems that I think should be addressed -- though please feel free to modify my suggestions and re-interpret my comments.

3.2. Multi-armed bandit (MAB):

If the goal is to measure "tendency to explore" as in the "trade-off between exploration and exploitation", then one should always think also about exploitation when measuring exploration. This raises the equivalent but easier question that "what does it mean to exploit in each behavioral task?". For the MAB, exploiting is to choose what has turned out to be best so far. Therefore, exploration is to choose something other than the so-far best option.

If the outcome is too stochastic, then the so-far best option may change frequently. This implies that an agent that always exploits would switch its actions frequently. The behavior of such an agent is considered by the author's first measure (MAB-Switches) as very exploratory, while the agent (by design) does not explore. So, I would claim that the 1st measure combines the effect of the outcome stochasticity with the exploration. I suggest keeping this measure as a control (since it has been used often before), but the results related to it should not be over-interpreted -- particularly when answering Q2.

The 2nd measure (MAB-Greedy) is, however, well designed. In particular, I think it is useful to note that MAB-Greedy is the maximum likelihood estimate of epsilon for an epsilon-greedy algorithm fitted to data. However, the problem with this measure is that it does not dissociate between the random exploration (RE) and the directed exploration (DE) -- i.e. it interprets both types of exploration as random.

I have a model-based suggestion to fix this issue. Similar to Eq. 1 in supplementary materials (also see * below), let's assume $\pi_{i,t}$ is a subject's estimate of the average reward of arm i at time t . Let us assume that the arm i has been chosen $N_{i,t}$ times until time t . Then, following common practice in neuroscience [2,6] and reinforcement learning theory [7], one can consider (similar to Eq. 1) the following action policy

$$p_{i,t} \propto \exp[\tau \cdot (\pi_{i,t} + \beta / \sqrt{N_{i,t}})],$$

where both τ in $[0, \infty)$ and β in $[0, \infty)$ are free parameters. The parameter τ (as explained by the authors) controls the randomness of the policy, whereas the parameter β controls a deterministic preference for the less chosen arms. These two parameters can be estimated (using maximum likelihood, like MAB-Greedy) for each participant -- unfortunately, there is no analytic formula for the estimate, but even a simple greed search can do the job for such a simple model. Then, the estimate of τ can be seen as a measure for RE and the estimate of β as a measure for DE. This will help the interpretation of the results of the analysis for Q1 and Q2.

*: The simulation part in the supplementary material is strange. Based on the model assumption, the only source of exploration is τ . Why are we interested in the correlation between MAB-Switches and MAB-Greedy for a fixed τ ? If MAB-Switches and MAB-Greedy have lots of variabilities for a given τ , then it means that they are not basically capturing the exploration. It would be more interesting to look at the correlations between MAB-Switches and MAB-Greedy and τ -- to see how informative they are about τ (i.e. the only source of exploration in the model)

3.3. Alien game (AG):

The two measures (AG-Hamming and AG-Active Search) are interesting. However, they are not necessarily candidates for two "different conceptualizations of exploration" (line 449). The goal of a participant is to find the painting with the highest price, so which strategy he/she chooses to explore depends on his/her assumptions about the task structure: if symbols are assumed to have semi-independent effects (e.g. aliens always like when the 1st symbol is chosen independently of the rest), then AG-Hamming captures exploration better; but if each combination (painting) is assumed to be completely distinct from the rest, then the 2nd measure one makes more sense. Given the current instruction, I assume different people may make different assumptions. Is it possible to control this? If so, we will most likely observe a reduction in the noise level.

If AG-Hamming is assumed to be a good measure of exploration, then I assume that the rate of decrease in Hamming distance (during one block) should also be an interesting measure -- i.e. how fast they converge to a solution.

3.4. Optional stopping with recall (OS):

As noted also by the authors (lines 306-307), the measure has a clear drawback of not including the current best option and the cost of opening new boxes. There are multiple ways that they can be included.

3.5. Sampling paradigm (SP)

The goal of an agent is to find the best option. So, the measure of exploration should also include how different the two options are -- if the difference is too big, then with a smaller number of sampling one can find the better option and get confident about it.

3.5. Observe or bet (OB)

Same as SP: the difference between estimated probabilities should be taken into account.

4. The plausibility of answering Q2

Common factor analyses have explicit and clear assumptions on the functional form of the dependence of the observable variables on the common factors. However, as discussed in point 3, there may be many non-linearities involved in the transformation of a hypothetical tendency to explore into any of the 11 observable exploration measures. Therefore, it is important to note that not being able to find any common factor does NOT imply that "The behavioral measures are capturing task-specific behaviors" (c.f. Table 1, Q2, column 5). It should rather be interpreted as "We could not find a task-independent variable that ...". This answer would still be interesting, but it would be more accurate.

5. Statistical analyses

- I suggest (in particular to answer Q1) using Bayesian methods for hypothesis testing in addition to the proposed frequentist methods -- e.g. see [8] for an example of Bayesian correlation test. In this sense, you avoid classic problems of frequentist hypothesis testing (such as the tendency to reject null hypotheses and issues for multiple comparisons) and you can also accept the null hypothesis (the lack of correlation) and make more interesting conclusions.

- Having a linear relationship between different exploration measures is a strong assumption (see points 3 and 4). Particularly, this assumption becomes important if we go beyond interpreting the correlation sign and study the effect size (e.g. comparing the correlation with 0.2 for Q1). Therefore, I suggest reporting also the Spearman or the Kendall correlation.

- For correction for multiple comparisons, I am not convinced by how a "family" of hypotheses is defined for Q1. I think the correction should be done according to the total number of hypotheses, i.e. $11 * 10 / 2 = 55$. Also, is there any reason that the authors chose to correct FWER instead of FDR (c.f. [9])?

References:

[1] Kobayashi et al. (2019), DOI: 10.1038/s41562-019-0589-3
[2] Xu et al. (2021), DOI: 10.1371/journal.pcbi.1009070

- [3] Gottlieb and Oudeyer (2018), DOI: 10.1038/s41583-018-0078-0
- [4] Schulz and Gershman (2019), DOI: 10.1016/j.conb.2018.11.003
- [5] Dubey and Griffiths (2020), DOI: 10.1016/j.cobeha.2020.07.008
- [6] Jaegle et al. (2019), DOI: 10.1016/j.conb.2019.08.004
- [7] Strehl and Littman (2008), DOI: 10.1016/j.jcss.2007.08.009
- [8] Rouder and Morey (2012), DOI: 10.1080/00273171.2012.734737
- [9] Efron and Hastie (2016), link: https://web.stanford.edu/~hastie/CASI_files/PDF/casi.pdf

Reviewer #4 (Remarks to the Author):

The current paper proposes a registered report to examine convergent validity between different behavioral and self-report exploration measures. I am reviewing this proposal as an expert in open science practices (including registered reports), along with expertise in personality psychology and measurement. I am not an expert in this particular trait domain, nor am I familiar with most of these tasks and measures.

The authors propose an ambitious design to test an interesting research question. There are many positives about the design, notably a large sample size, together with a thoughtful approach to a variety of exploration tasks and measures. I particularly appreciate that the authors have already shared many research materials and have a plan to share all study data and analysis code on study completion. However, there are some substantial weaknesses in the analysis plan. I outline the major issues below.

1. Most importantly, the study design proposed is of the multi-trait, multi-method (MTMM) variety, yet the authors do not plan to handle the data in the appropriate way to test models from a study with this design. To learn more about this approach, I can recommend the relevant chapter in the Kline structural equation modeling textbook. Many studies have used this approach, but one that might be similar to the study proposed here is Bar-Anan & Vianello (2018, JEP:G). Essentially, rather than testing each correlation one-by-one with an arbitrary 0.20 cutoff, you will instead want to test nested models with a variety of competing structures against each other. In this way, you will be able to test convergent validity alongside questions about the trait-like-ness of these measures in one succinct approach.

2. The authors' plans for assessing and dealing with measurement error are underspecified in the current plan. There is a plan to examine test-retest reliability (good), but there is no mention of any plan to examine internal consistency for either self-report or behavioral measures. Notably, when you want to compare two different types of tests to one another, it is important to consider whether or how measurement error might be different between them. The behavioral tasks that are outlined in the proposal have a variety of different scoring strategies (and numbers of trials) that lend themselves to very different potential for unreliability. Most, but not all, of the behavioral measures involve averaging performance across a number of trials and/or blocks. It would be interesting to know whether scores are consistent (i.e., positively correlated) within these different parts prior to aggregation. You could even model the trials like you might for individual self-report items in the MTMM approach. Even if you stick with your planned scoring strategy rather than modeling individual trials or blocks, you will want to assess consistency prior to aggregation. If the behavioral measures are less reliable than the self-report measures, it will depress correlations among the behavioral measures and between the behavioral measures and the self-report scales.

3. Laudably, the authors intend to examine test-retest reliability of the measures. This is a good idea. However, as with the missing plan for dealing with measurement error, there is a missing plan here for dealing with potential measurement invariance. Again, there are directions in the Kline book for how to approach this. The key issue is that you want to check whether the factor structure for the measurement of each task or scale is consistent over time.

4. The authors propose to use a fixed order, with behavioral tasks before self-report tasks. I'm not sure this is a good idea, given the potential for fatigue over the long protocol. (By the way, can participants really do all five tasks and all the self-report measures in an hour?)

5. As a personality psychologist, I note the absence of other self-report measures, such as "openness to experience" in the big five framework. Typically, openness and extraversion are where we think about exploration fitting in. This is just a note to mention this absence, not a request to include a measure. But the authors might find relevant literature in this domain as well.

6. As a general point, there are many decisions that need to be made to have a well-specified registered report. It seems likely to me that additional issues will come up and need to be resolved after changes are made the analysis plan. That is, I anticipate raising more queries on the next round of review should the authors be given a chance to revise. (As an example, registered reports should have positive controls if possible. This means including checks to ensure that data are of expected quality; e.g., perhaps you can already be reasonably confident that two or more of the self-report measures will correlate strongly with each other. If such a correlation were not present it would indicate some severe issue in the data.)

Smaller points:

7. The authors write "we want to use data from participants motivated to complete the study." Why is this the case? I didn't understand the logic behind this exclusion.

8. I have not seen EFA interpreted using model fit statistics. Typically, people use parallel analysis (good) or some other rule like Kaiser's criterion (less good). This might not be of consequence if you change your modeling strategy to the MTMM technique.

9. The paper is not written to be understandable by an outsider. For instance, "multi-armed bandit" is mentioned several times without explanation in the introduction.

I sign all of my reviews,
Katherine S. Corker

Dear Reviewers,

We would like to thank all of you for your constructive feedback. We believe that the manuscript has been substantially improved thanks to your comments. We hope that we've understood and addressed your comments appropriately and we would be happy to address any further comments you may have. Below we respond to each of your points.

Kind regards,
Authorship team.

Reviewer #1

The authors propose a correlational study to investigate human exploration behavior. More specifically, they aim to compare exploration behaviour between different experimental tasks and questionnaires to study whether exploration behaviour in different situations is supported by a domain-general tendency. In addition, the authors propose to study whether people's tendency to explore is stable across time.

The manuscript is well written, presents an important research questions, a clear rationale, methods section and analysis pipeline. I really like the proposed study, but have two main comments/concerns (for more detail. see comments below). First, the main aim of the proposed study is the investigating of the (convergent) validity of different measures of human exploration behaviour. This is fine, but the authors should also introduce and, at least briefly, discuss the theoretical accounts of exploration-exploitation trade-offs and behaviour. The second point is related to the domain-general hypothesis and the selection/exclusion of exploration tasks. Without a more detailed discussion of what theories/mechanisms underpin the exploration-exploitation trade-offs in the selected/excluded tasks, it is difficult to assess the significance of the proposed study and unclear how results might generalize to other tasks. Finally, I would like to learn more about the reliability with which one can assess people's exploration tendencies in a series of experimental tasks, each of which has to be carried out in just a few minutes.

*** *Response.* Regarding your first main point, as per our responses to your more detailed points below, we've now added some (brief) theoretical context for exploration-exploitation behaviours. To address your second main point, we have provided a more thorough discussion for our decisions regarding which tasks we included, and which we excluded. In so doing, we have also made clear that our conclusions may be limited, but that our investigation is a substantial first step towards understanding the generalizability of exploratory behaviour across tasks. Regarding your final point, as per our more detailed comments below, our analyses of the pilot data show that the measures of exploration are quite reliable. Moreover, as per our responses to Reviewer 4, our analysis plan for the main study now includes checks on the reliability of the different measures and steps we will take to address any potential issues of reliability that may arise.

Detailed comments: [note that we, the authors, numbered each point from the reviewer for ease of referencing]:

Point 1

- The authors mention paradigms and tasks used to study exploration-exploitation behavior including tasks “designed to mimic foraging” in the introduction. There are a few studies in which participants’ task was to move through space and collect patchily distributed resources (at least in virtual environment e.g. Hills et al., 2013; Wolfe, 2013). Given that foraging theories primarily aim to explain actual foraging behaviour, I believe these studies should be cited.

*** *Response.* We had already cited papers that used the foraging task you mention (i.e., Hills et al., 2008, 2010). We’ve now also added the two citations you’ve noted.

Point 2

- The authors do not review foraging theories that aim to explain exploration-exploitation trade-offs. While their study is not aimed at investigating these theories directly, I believe they should be introduced briefly to provide the theoretical context.

*** *Response.* We wanted to be brief in the introduction. Nonetheless, you’ve raised an important point. Therefore, we’ve now added the following theoretical context to the manuscript (pp. 5-6):

“For example, researchers have hypothesized a domain-general cognitive search process that calibrates people’s tendencies towards exploration and exploitation (Hills & Dukas, 2012; Hills et al., 2008, 2010, Hills et al., 2015; Wilke et al., 2009). A logical consequence of this hypothesis is that there is a domain-general tendency to explore, such that how and how much people explore and exploit will be similar across different contexts.”

Point 3

- The authors excluded foraging tasks as the exploration-exploitation trade-off in these tasks is inextricably linked with the effect of diminishing returns. I do not quite understand why this would disqualify foraging tasks from the current study, particularly because theories of exploration-exploitation trade-offs have originally been developed to explain foraging behaviour. The authors should provide a more in-depth explanation for why they believe that domain-general tendencies for exploration may not be predictive for foraging tasks.

*** *Response.* We do not claim that domain-general tendencies for exploration will not be predictive of foraging tasks. We’ve added the following to the manuscript to clarify the issue (p. 21):

“Perhaps more importantly, Von Helversen et al. (2018) used confirmatory factor analysis to examine the correlation between exploratory behaviour in a foraging task (the number of new patches visited by participants) and in a 5-armed bandit task (the switch rate), finding a negative correlation after removal of outliers and a nonsignificant correlation otherwise. This finding suggests, as mentioned by Von Helversen et al., that structural differences between tasks may hide any domain general tendencies towards exploration. We chose to use the multi-armed bandit task because several of the other tasks that we also include are derivatives of the multi-armed bandit task (i.e., the sampling paradigm and observe or bet task), and because we are more familiar with these tasks, relative to foraging paradigms. Domain general tendencies should be more likely to emerge when comparing tasks that are more similar to one another, such as those

mentioned. Therefore, although we use some of the theoretical reasoning developed to explain foraging behaviour, we've made design choices for the proposed study that we think are defensible and which offer a compelling overall research design."

We also refer you to our response to the next point, where we further explain our reasoning.

Point 4

- Related to my last comment, I wonder about the domain-general hypotheses, and how the selection of the experimental tasks affects how strong the test of domain-general is. I believe the authors should make explicit what the exact properties of everyday exploration-exploitation trade-off situations they are interested in (or which aspects their study design is able to address) and what aspects/mechanism are beyond the scope of the proposed study. For this discussion, a more detailed explanation of the underpinning theoretical framework would be helpful.

*** *Response.* Following from our response to your previous point, we further added to the manuscript (p. 22):

"We selected tasks that we thought might be more likely to be related to one another due to the similarities they have. This means that our test for domain-general is somewhat more limited than it could otherwise be. Nonetheless, if there is a domain-general tendency to explore, and the measures typically used in various literatures capture that tendency, then we should observe the data patterns we've predicted among the measures that we've selected. If these data patterns aren't observed among our relatively homogeneous tasks, then we can be less confident in the ability of these tasks in capturing a domain-general tendency (or at least the summary measures typically used from them, such as the switch-rate in the bandit task). There may still be a domain-general tendency that is captured by foraging tasks and other exploration tasks, but that would be an empirical question for future research to examine. On the other hand, if we do observe the predicted data patterns, then we can be more confident in the idea that behavioural tasks and the summary measures used from them can capture a domain-general tendency to explore. Future research could then examine how far that general tendency stretches (e.g., whether it extends to foraging tasks). Moreover, researchers could examine whether the domain-general tendency extends to cognitive foraging tasks (e.g., information search, memory search, and/or anagram search). In addition, given that there is a diverse range of foraging tasks, it would be an interesting question for future research to examine the convergent validity of these relative to one another."

Point 5

- In the last comment I mentioned that a more detailed explanation of the theoretical underpinning of exploration-exploitation behavior would be useful. This would also help with understanding how the different exploration measures for the alien game fit with the overall aim of the study.

*** *Response.*

The purpose of including the different measures of exploration from the alien game is to examine whether the two measures vary in their relationships with measures from other tasks. If the two measures have different relationships with the measures from other tasks, then this could speak to

the different conceptualizations of exploration, and help to explain why some measures do or don't correlate with each other.

For instance, active search measures how much people will continue search for the best alternative, reflecting whether people make a change to the status quo on each trial. This measure has a conceptual correspondence to the self-report measures of maximization which are essentially designed to capture the idea that some people have a tendency to conduct an exhaustive search for the best alternative. The measure also corresponds to the optional stopping task, which examines when people decide to settle with what has already been discovered. Given that we already include the Alien Game in the study, we decided that it makes sense to also include the other measure of exploration used in the literature from this task, the Hamming distance, which captures how far people explore.

The Hamming distance measures how many decision attributes/features are changed from a trial to another (and it's thus bounded between 0 and 10, the total number of decision attributes that jointly create a configuration), so that small values of this measure indicate search in a neighborhood of the current configuration, whereas larger values indicate searching in far away portions of the configuration space. Therefore, this measure captures different qualities of exploratory/search behavior, according to the distinction between *local* and *global* search made in experiments that use the Alien Game as well as agent-based simulation modeling that investigates organizational learning and decision-making (e.g. Denrell et al. 2004; Ghemawat and Levinthal 2008). Conceptually similar measures are also used, for instance, in citizen science projects that investigate remote optimization (Reck et al. 2018).

We decided to leave this explanation out of the introduction which we wanted to keep brief and to the point. These theoretical discussions will be added to the general discussion section if and when they become relevant (e.g., if we observe one but not the other measure from the alien game correlating with measures from other tasks or the self-reports).

Point 6

- The authors estimate that it will take participants approximately one hour to complete the study. In this hour, they will need to complete five different exploration-exploitation tasks, fill out questionnaires, provide informed consent, etc. This leaves little time for each of the task. However, to obtain reliable estimates of participants' exploration tendencies, participants will not only have to understand the tasks well, but will also have to have sufficient experience with the tasks to understand costs and potential benefits for different options in each task well. I understand that the authors have piloted their approach and that neither tasks results in floor- or ceiling performance. However, my question is whether (and why) the authors convinced that they will be able to measure peoples' exploration tendencies reliably using such short tasks?

*** *Response.* We understand your concerns about each measure's reliability. We have reason to believe that the measures are reliable based on our pilot data. We used our pilot data to report reliability analyses, referred to briefly in the manuscript and reported in full in the Supplemental Information (under the heading, "Results from Pilot Data"). In these analyses, we correlated exploratory behaviour for each measure across the different blocks of the same task. For example, in the multi-armed bandit task (MAB) we correlated the switch rate in the first two blocks with the switch rate in the second two blocks, and then we correlated the switch rate in the odd numbered

blocks with the switch rate in the even numbered blocks. The results of these analyses showed that almost all of the measures had good reliability. Namely, the correlations were: switch rate and complement of the best reply rate in MAB (.77-.91); optional stopping task (.61 and .67); sampling paradigm (.50-.65); observe or bet task (.60); hamming distance in alien game (.55-.75); and active search in alien game (.39-.59). The latter measure showed lower reliability for one of the analyses but this was likely due to the alien game having fewer participants ($n = 20$) as a result of a software error that resulted in dropping some data.

Moreover, in our response to Reviewer 4 (point 2), we now also report the Cronbach's alphas for each measure, which captures the internal consistency (another measure of reliability) of the measure. The Cronbach's alpha for each measure shows satisfactory reliability estimates for each behavioural measure. We have also included a plan for dealing with potential problems with reliability of a measure (see our response to Reviewer 4 point 2).

Reviewer #2

A large number of experimental paradigms in the behavioral sciences require participants to engage in both exploration and exploitation (e.g., the multi-armed bandit task). These tasks are often assumed to measure some general tendency of an individual to explore their environment that is stable across time. Surprisingly, few studies have examined whether this is true, i.e., whether the construct of "exploration" is in fact a stable, domain-general trait, and whether there is convergent validity among the variety of tasks used in the behavioral science that purportedly measure this trait. The authors propose to test these claims with a large sample on multiple exploration paradigms, and analyzing the data using pairwise correlations and equivalence testing, factor analysis, and test/re-test reliability at two spaced time points.

Overall, I found the manuscript well-written and well-structured. The authors' hypotheses have important implications for the field, and are relevant given the increasing interest in exploration/search tasks in the past decade. I also think the study is perfectly suited for this two-stage review process: while I have personal doubts about whether the tasks do engage some domain-general and stable exploration trait, the results will be informative regardless of the outcome. (If it turns out that the null hypotheses cannot be rejected, it would certainly be harder to publish the results using a traditional review process that occurs after data collection and analyses). The proposed procedures and analyses are clear enough to replicate the study. The analysis techniques are appropriate, and I commend the authors for commitment to avoid overfitting the data (e.g., pre-hoc Bonferonni adjustments in H1, cross-validation to use exploratory and confirmatory factor analysis in H2). I recommend accepting the manuscript for Stage 1, and look forward to seeing the results. Below, I provide several comments that the authors may want to consider. I believe all of these comments to be relatively minor. I hope the authors give them serious consideration, but leave it to the other reviewers and editor as to whether any of them should be required.

Point 1

1. The manuscript (particularly the Introduction) makes frequent reference to a number of exploration tasks (e.g., the eight listed on lines 58-62). The five tasks used by the authors are described late in the paper, and tasks not used by the authors are described very little. Yet a

number of comparisons are made, and much of the discussion in the Introduction requires an understanding of how these tasks work and the differences between them. I suggest the authors create a Table, referenced in the Intro, which has the task name (in one column) and a brief, 2-3 sentence description of the task from the participant's perspective (in another column). The five tasks used in the current study should be clearly identified (e.g., listed above the unused tasks). This would help make the manuscript more self-contained, so that the reader does not need to consult other work in order to understand the claims.

*** *Response.* This is a good suggestion and we've now included a reference in the introduction to a Table that contains a brief description of each task. We didn't include the grid search task or the chain task because we described these briefly in the text already. For ease of reference, we reproduce the table (Table 1) here:

Task	Description	In this study
Multi-armed bandit	Participants choose between a given amount of options on each trial. Each option has a predefined payoff distribution. Participants see how many points they've won after each choice. How often participants change their choice across options from one trial to the next reflects how much they explore.	Yes
Alien game	Participants toggle on and off buttons (or decision attributes) that have to be configured jointly. Participants choose a combination of on and off buttons, and receive a reward associated with that button combination. How much they change the configuration of on and off buttons from one trial to the next reflects how much they explore.	Yes
Optional stopping with recall	Participants see a grid of options each hiding a predetermined stationary reward. Clicking on an option reveals the reward behind that option but comes at a cost. The costs accrue, so that revealing more options is more costly than revealing fewer options. Participants can choose when to stop revealing options, at which point they receive a reward equal to the highest paying option revealed so far minus the total accrued cost. The number of options a person reveals reflects how much they explore.	Yes
Sampling paradigm	Participants see two or more options each with some probability of paying out a reward. Participants can click the options without cost to learn about the options' payoff distribution. Once satisfied, participants choose which option they want to be paid out from. The number of times participants sample options to learn about their value reflects how much they explore.	Yes

Observe or bet	Participants see two lights on the screen and only one light turns on in any given trial with some predetermined probability. On each trial, participants can choose to observe which light turns on so that they can learn about the probabilities, or they can choose to bet whether one or the other light turns on. Observing means they see which light turns on but they earn no points. Betting means they don't see which light turns on but they get a point if they bet correctly and lose a point otherwise (they only know how many points they've won or lost at the end of the block of trials). The number of trials in which participants choose to observe reflects how much they explore.	Yes
Spatial foraging	Participants see a landscape with many options/locations on their screen and their search icon is placed in a starting location in the landscape. The landscape contains rewards at unknown locations. Participants must physically move the search icon through the landscape in search of the hidden rewards. The proportion of total surface area that is visited by participants reflects how much they explore (though there are also other measures of exploration and different types of foraging tasks.)	No
Secretary task	Participants are presented with a random sequence of options, one at a time, from a pool of options (the options are described as applicants for a secretary job). On each trial, participants choose to either reject the option they're presented with or to accept it. The pool of options is ranked from the worst to the best. The task is to accept the best option but participants can only accept the current option and can't go back to an option they've already rejected. The number of options participants search through reflects how much they explore.	No
Card search task	Participants are presented with a deck of cards face down (on the computer screen). Each card is associated with a predetermined value. Participants have a specified number of turns (e.g., 20) in which they can choose to either turn a new card over to see the points (i.e., explore) or to choose to earn the value of a card that they have already turned over (i.e., exploit). They can exploit the same card over multiple turns. The number of cards a participant turns over is the measure of how much they've explored.	No

Point 2

2. The exploration measures used by the authors are all data-based (as opposed to model-based). However most (if not all) of these tasks have been modeled computationally, in some cases with explicit parameters that represent a latent exploration trait (e.g., the softmax temperature parameter in a multi-armed bandit task). In contrast, the data-based measures are noisy measures of exploration because they are influenced by other factors, such as the participant's goals, the

task's optimal strategy, the participant's subjective utility, or differences in observed data across participants leading up to a decision point (due to chance). Examining model-based parameters seems like an interesting future direction, but a large endeavor that can be justified as outside the scope of the current manuscript. However I believe it would be helpful to include a short discussion on the topic and acknowledgement that the proposed exploration measures reflect more than just exploration.

*** *Response.* Thank you for pointing this out. You're right that our study clearly resembles a data-based approach for both the exploratory tasks and the self-report scales. A model-based approach would certainly also be very interesting and should be conducted in a future study. Indeed, our intention was to include discussion of this at the second stage submission, in the discussion section of the paper. We now discuss this in the introduction (pp. 25-26):

“Our measures of exploration, particularly for the behavioural tasks, are based on one approach used in past work and relies on summarizing people's responses across trials and blocks. Although the tasks in the present study are designed (e.g., the appearance of each task, number of trials and blocks, and scaling of payoffs) to reduce noise, the various summary measures of exploration may still, to some degree, deviate from each other because they can be influenced by factors other than a general tendency to explore. For example, behaviour in a task may be impacted by the task's optimal strategy (if there is one), the participants' personal goals, chance variation between participants in the observed payoffs leading to a decision point, and the sequence of the blocks of the task presented to each person. Importantly, behaviour in the present study's tasks have been, and can be, computationally modelled with parameters that represent a latent tendency to explore. And such measures can be designed to account for the influence of the task specific factors on behaviour. Examining the convergent validity of the measures from computational models is beyond the scope of the present study, but is definitely worthy of future research”.

Point 3

3. The procedure indicates that the self-report scales will be completed immediately after the behavioral tasks. A concern is that participants will answer questions on these scales by thinking about their performance on the previous tasks. (“Am I uncomfortable making decisions before knowing all of my options? Well my behavior on the previous task suggests that I am...”). In other words, participants may be describing their behavior on the task rather than revealing some trait that actually predicts behavior. Intuitively, I feel like the converse is less likely (I don't expect participants to modify their behavior in the behavioral tasks to match their answers to the scales). Perhaps it would make more sense for participants to complete the scales first, or to counterbalance the order?

*** *Response.* Following your suggestion, we thought through this issue more carefully and decided to go with the option of counterbalancing the order of presenting the self-reports or behavioural tasks first. Therefore, now, participants will be randomly allocated to either respond to the self-reports first or to complete the behavioural tasks first. The self-reports will all be presented together and the behavioural tasks will be presented together. The relevant section of the Design in the manuscript has been updated (p. 15).

Point 4

4. (line 242) The Methods section begins with a discussion of the Alien task, which is not mentioned in the Introduction. Considering its importance in the pilot data and its use in the experiment, it should be discussed in the Introduction as well – otherwise the transition seems a little odd (you discussed 8+ exploration tasks... and then choose a completely different one to use for this study?)

*** *Response.* We've now referred to the alien game by name in the relevant section of the introduction (see p. 4) and, as per our response to your point #1, we also refer to the Table that gives brief descriptions for every task we've mentioned in the introduction, including the alien game.

Point 5

5. Similar to point #1, I found the detailed discussion of some tasks in the Methods to be confusing because a number of comparisons are made before the tasks are actually described. One possible re-organization takes the discussion of 'excluded tasks' or 'task selection criteria' (lines 318 - 388) and places it in its own section after the description of the tasks used (i.e., moving it to around line 495). Similarly, the section on self-report scales begins obliquely by referring to "the first version of the scale" (What scale? Does it have a name?), followed by an actual description of the scales (lines 514 - 536). It might aid the reader if the scale descriptions were provided at the beginning of that section.

*** *Response.* We agree with you and have restructured this part of the Method section as suggested.

Point 6

6. The payoff structure in each task is stationary, but are participants aware of this? It may not be obvious to naive participants, who could adjust their behavior throughout the experiment (e.g., one might believe that the first arm in a bandit task has the highest expected value, but after a sequence of bad outcomes they might believe that the experimenter "switched" the arms on them). Normatively, a participant should explore more when the payoff structure is dynamic compared to when it is stationary. In some tasks, such as the Iowa Gambling Task, the instructions intentionally emphasize that the payoff is stationary (i.e. "there are two good decks and two bad decks..."). It may be worth probing this belief in the instructional check questions, to ensure participants understand.

*** *Response.* The payoff structures in all tasks are stationary within each block, but may change from one block to the next. For some tasks the instructions make this clear: in the alien game participants are told that "if you submit the same picture in the same block, you'll be paid the same price. [...] The value of the pictures will change from one block to the next."; in the observe or bet participants are told "...this probability will remain the same throughout each block of trials but will change from one block to the next.". In the optional stopping task participants choose to accept their currently highest paying box or to keep searching, implying that the rewards within the block are stationary. We've now added that "The costs for opening boxes, and the rewards from each

box, changes from one block to the next.”. In the sampling paradigm they can sample each option as much as they want before choosing which to be paid out from, which implies that the rewards are stationary within the block. Nonetheless, we’ve now also added “The average payout from each button remains the same within each block, but may change from one block to the next.”. In the multi-armed bandit, we didn’t include such a statement about stationary rewards. We now tell participants that “The average payout from each button remains the same within each block but may change from one block to the next.”.

We discussed whether it is worth probing participants’ beliefs and decided against this because including another comprehension check question would add to the burden on participants. In addition, given the large number of tasks and instructions we decided that the issue of beliefs may not be too problematic. That is, regardless of what a participant believes, if there is a trait-like tendency to explore then the higher that participant is on the trait the more they’ll explore in the task and vice versa.

Point 7

7. I found the Alien game to be a bit confusing because of its multi-attribute nature. The first measure of exploration for the task is how far apart two consecutive sequences are (Hamming distance). Which is a reasonable measure if you believe that each binary choice in the sequence is independent, and that payoffs are based on features and not relations (e.g., it could be that alien enjoys particular bigrams, so each is not independent). As with point #6, I think this aspect could be probed in the instructional check questions.

**** Response.* In the Alien Game that we use the binary decision attributes in the configuration are not independent. The interdependence between these choices is defined by the parameter K which is static and set to three for all blocks of the alien game. $K=3$ results in a low-medium complex landscape in which three unknown decision attributes are interdependent with each other. Note that choices concerning all ten binary decision attributes have to be made to obtain a payoff (and complete a round); hence, the Hamming distance is a meaningful measure for all levels of interdependence. The measure is also not only used in Alien Game experiments but also in agent-based simulation modeling that investigates organizational learning and decision-making (e.g. Denrell et al. 2004; Ghemawat and Levinthal 2008). We use the game the same way it has been used in this literature and we therefore opt to keep it as is. Importantly, as per our response to Reviewer 3 (point 3.3), we added instructions telling participants that the attributes are interdependent.

As per our response to Reviewer 1 (point 5), the active search measure is the main measure for the alien game that we were interested in but because the literature often uses the Hamming distance we decided to also incorporate this measure into the analyses.

Note that in all blocks we used landscapes in which a given combination of attributes is always associated with the same payoff. Hence, we have stationary payoffs as described in point #6.

Typos / grammatical

- (abstract) “Upto” -> “Up to”

- (line 620) “We’ll” -> “We will”

- (line 142) Do you mean “conservative” or “liberal”? Setting a low cut-off value ($r = .2$) means that you are more likely to obtain false positives than if you were to set a high cut-off value. A “conservative threshold”, in my opinion, is one that minimizes the number of false positives by setting a high bar. Perhaps I’m in the minority in my interpretation, but I found the word choice to be at odds.

*** *Response.* Thanks for these points. We fixed the typo in the abstract and removed “conservative” in reference to the cut-off (the cut-off doesn’t need a descriptor so clarity is served by just removing that adjective).

Reviewer #3

In this registered report, the authors ask whether different experimental measures of exploration (Q1) capture similar (individual-specific) qualities, i.e. testing convergent validity, (Q2) can be attributed to a common task-independent but the individual-specific quality of 'tendency to explore', i.e., testing domain generality, and (Q3) are stable over time, i.e. testing temporal stability.

To do so, the authors plan to complete 5 carefully chosen behavioral tasks and 4 self-reports with a massive number of participants -- while a number of participants will redo the task after one month. The plan is to extract 11 measures for exploration (7 based on behavioral tasks and 4 based on self-reports): then, (Q1) the convergent validity is going to be tested by studying the cross-correlation between different exploration measures, (Q2) the domain generality by conducting common factor analysis on these 11 measures, and (Q3) the temporal stability of each measure by studying its self-correlation between the 1st and the 2nd round of the experiment.

I believe that the answers (either negative or positive) to these 3 questions (Q1-3) are crucial for re-evaluation of the common methodologies and assumptions in studying human exploratory behavior and will help both experimental and computational scientists in different fields (including psychology and neuroscience) in designing, analyzing, and evaluating new experiments, theories, and computational models. Moreover, collecting and publishing such a rich dataset is going to be an important contribution to the field.

The author's choice of experiments and their analysis plan for answering these questions sound relatively promising, but I have some major comments regarding the generality of the study, the

interpretability of possible outcomes of the analysis related to Q1 and Q2, the choice of exploration measures, and the details of statistical analyses. I will explain each point in detail below.

Overall, I enjoyed reading the report very much, and I am looking forward to reading its revised version.

Point 1

1. The generality of the study:

Through Abstract and Introduction, there is an implicit claim that the authors want to study the general notion of the "tendency to explore". However, in Methods, it becomes clear (c.f. lines 318-337) that, based on the choice of behavioral tasks, they study the "tendency to explore" as in the "trade-off between exploration and exploitation" (also see * and ** below). However, exploration can be studied even in the absence of reward (e.g. [1]) or when there is no trade-off (e.g. [2]). I think this limitation should be explicitly mentioned when the questions (Q1-3) are asked in the paper. I believe that even with this limitation, the study is quite general and the questions are very important, but making the assumptions and limitations explicit puts the study in the right context and helps to avoid over-interpretation of the results.

*** *Response.* You're correct. Our focus is mostly on exploration-exploitation trade-off situations. We've now made it clearer that, in addressing the 3 questions, our study is focused on the context of exploration-exploitation trade-offs:

"We designed a study where more than 500 participants will complete five behavioural tasks and four self-report scales developed to measure exploratory behaviour, most of which examine exploratory behaviour in the context of exploration-exploitation trade-offs." (abstract).

"Many, though not all, exploratory behaviours involve a trade-off between exploration and exploitation." (p. 3).

"We propose a study that addresses these questions in the context of exploration-exploitation trade-offs." (p. 3).

* In contrast to the behavioral tasks and the self-reports 2 to 4, the self-report 1 (i.e. "Exploration scale") is more related to exploration as in curiosity-driven behavior (c.f. [3]) than to the exploration and exploitation trade-off -- at least that is what I understand from the questions quoted in supplementary materials, e.g. the statement "I view challenging situations as an opportunity to grow and learn". I think this should be taken into account when interpreting the final results.

*** *Response.* You're correct in your interpretation of the scale. In fact, the exploration scale you refer to is from the Curiosity and Exploration Inventory. We will interpret the results of this scale with this point in mind. Of course, if the scale captures individual differences in curiosity-driven exploration, then we should observe that people who score higher on this scale also search more in at least some of the tasks, such as in the sampling paradigm, presumably because they're curious about the payoffs from the options.

** : In the alien game (AG) and the sampling paradigm (SP), the trade-off between exploration and exploitation is NOT in terms of monetary payoff: in fact, in these two tasks, the more participants explore, the more they will be paid without losing anything other than their time and energy -- in contrast to the other tasks. I think this should also be taken into account when interpreting the final results.

*** *Response*. This is a very good point that we hadn't thought about. Though in the alien game there is a monetary opportunity cost produced due to the explore-exploit trade-off. This is because the points that a participant earns in each round/trial accrues towards their total points which then determines the bonus they earn. Therefore, if people find a relatively high paying sequence, especially in an early round, and then decide to explore other sequences means that they would be forgoing the payout from that high-paying one in the hopes of finding a higher-paying sequence. But this would be at the risk of finding a lower-paying sequence. Pay-off schemes similar to that of the alien game have recently been used in other exploration-exploitation tasks designed by Song, Bnaya and Ma (2019) and Sand et al. (2020). Nonetheless, your point stands for the sampling paradigm. Based on this, we've now added another category to Table 4 that outlines the structural similarities/differences between the different tasks in our study. Specifically, we've added a column to indicate whether each task has a trade-off between exploration and exploitation in terms of monetary rewards (see also text in manuscript on pp. 24-25). And we will take this into account when interpreting the results.

Point 2

2. The interpretability of possible outcomes of the analysis related to Q1 and Q2

If all 11 exploration measures turn out to be significantly and highly correlated with each other and if it turns out that there exists a model with a single common factor that can explain data best, then the interpretation is straightforward. However, I wonder what would be the authors' plan if it turns out otherwise?

In particular, I can imagine observing a few clusters of positively correlated exploration measures in response to Q1 and similarly finding more than one common factor in response to Q2. I agree that such an observation by itself is interesting and valuable, but I believe that a more careful choice of exploration measures will make interpreting such results easier.

For example, it has been argued that exploration can be decomposed into two components of "random" exploration (RE) and "directed" exploration (DE) [4] and that the DE can have different motives [1,2,5]. It is not trivial whether the tendency for RE is correlated with the tendency for DE. If it is not the case, then it is easy to imagine a model with more than one common factor to have higher explanatory power than any model with a single common factor. If exploratory measures are interpretable in the sense that they are closer to RE or DE, then it may be easier to interpret the common factors (see a particular suggestion of mine in point 3).

Therefore, I suggest adding explicit reasoning for why each measure of exploration is added and what it is supposed to measure -- similar to but more detailed than the comparison between the two measures of the Alien game in lines 449-453. Then, even if all 11 measures turn out to be correlated, the result would be more interesting as it would count as evidence for the similarity of conceptually different measures.

*** Response. Our reasoning for choosing several of the measures really amounts to what has been used in the literature and what are comparable tasks and measures. For example, the number of samples in the sampling paradigm and the number of observe trials in the observe-or-bet task are standard measures of exploration in these tasks, as used by researchers and reported in the literature. Similarly, the switch-rate in the multi-armed bandit task, the Hamming distance and active search in the alien game, and the number of options searched in the optional stopping task are what have been used in the literature. Although the inverse of the best-reply rate isn't typically used in past research as a measure of exploration, the best-reply rate itself is very often used as a measure of exploitation. Given that exploitation is presumably on the opposite end of the spectrum as exploration, we argue that the inverse of the best-reply rate should therefore be capturing exploration.

We agree that a likely outcome is that we find clusters of correlations and multiple factor models that fit the data better than a one-factor model. To interpret different clusters, we will use Table 4 which describes the structural similarities and differences between the tasks and measures to provide theoretical explanations for any clustering that we observe. This approach already potentially helps with the interpretation of models in which more than one factor fit the data and where various measures are clustered closer together than others. In other words, our analytic plan, combined with the table of similarities/differences between tasks and measures, already achieves what you suggest in terms of interpretability of various outcomes. For the specifics, see the "Updated Analysis Plan for Hypothesis 2" at the end of our response to Reviewer 4.

Your suggestion to consider whether each of the measures we have is more reflective of random exploration or directed exploration is interesting. However, in our design the distinction between random and directed exploration is not as sharp. The tasks that could have directed exploration are the alien game and the multi-armed bandit. In the alien game people could use multi-attribute information or information about the uncertainty of the options to direct their exploration. The hamming distance could also express directedness in search. In the multi-armed bandit, by contrast, people can use only uncertainty to guide exploration, and it would be challenging to devise a simple measure that would capture uncertainty and "directed exploration" in this case, unless we engage in cognitive modeling, which we chose not to pursue in this study.

Point 3

3. The choice of exploration measures in behavioral tasks

While I think the choice of behavioral tasks is very clever (as represented well in Table 2), I think the choice of exploration measures can benefit from some theory-driven arguments.

I make this more clear below.

Point 3.1

3.1. First, It should be clearly mentioned which measures are designed by the authors and which are the ones taken from the previous studies -- for example, it is not clear whether the hamming distance in the Alien game (AG-Hamming) has been used before or not. Second, I suggest (i) keeping the previously used measures as they had been defined in the paper potentially only for control) and (ii) modifying the ones that were designed by the authors and adding new measures

with more theory-driven arguments. Another table can be added representing this information. This is in particular important for answering Q2.

*** *Response.* We appreciate your well-thought-out suggestions and thorough critique of the measures. Regarding your point about modifying measures and adding new measures with more theory-driven arguments, we think that this goes quite beyond the scope of what we have intended for our study. Our intention is to examine the convergent validity of different measures used by researchers to examine exploratory behaviour and whether those measures capture a domain-general tendency. Therefore, we've focused on measures that have been used. We respond to your further points below.

Regarding which measures were designed by the authors for this specific study, and which were taken from the previous studies, we note that all of the measures we propose to use for the present study have been used and reported in the literature, except the single general exploration question that we adapted from the general risk preference item (see p. 27 of manuscript regarding this item).

Indeed, the switch-rate from the multi-armed bandit, the number of samples in the sampling paradigm, the number of observe trials in the observe or bet task, are all measures and tasks exactly as used in the literature by researchers (not developed by the authors of the present paper). The citations for the measures used in each of these tasks are already in the manuscript.

The alien game was developed by one of the authors and was used in a previously published paper. It resembles a basic behavioural task with NK landscapes that has been used by other researchers to examine exploratory behaviour often by using the Hamming Distance or related measures (e.g., Tracy et al., 2017; Vuculescu, 2017; Giannoccaro et al., 2020). In these task variations, instead of an alien buying pictures, the cover framing is different such that, for example, participants are managers of a firm trying to create new products to make profits. Active search in the alien game was developed by one of the authors in a previously published paper but it simply mirrors, or is similar to, measures used in other games (such as number of options searched in the secretary game). The measure used in the optional stopping task has also been used in the literature (Bhatia et al., 2021; this is a citation that we had forgotten to add previously) and is identical to the measure used in the common secretary task but with recall. The manuscript now makes clear whether and in which citations each of these measures from the alien game and the optional stopping task have been used in the literature.

Regarding answering question 2 in our paper (i.e., “Does exploratory behaviour generalize across different tasks and situations such that different behavioural measures capture a domain-general tendency?”; p. 3), there are three potential framings of this question. The first question is whether there is convergent validity of exploration measures, as they are typically used in the literature across disciplines. If established, such convergent validity would imply the presence of a general underlying tendency to explore. The second question is whether there is a general tendency to explore that *can* be captured by the tasks and measures typically used in the literature across disciplines. The third question is whether there is a general tendency to explore that can be captured by *any* tasks and measures.

The distinction between these three framings of the question is important. The third question is much larger than what we intend on examining. There may be a general tendency to explore and measures may need to be specifically designed and thoroughly tested for capturing this tendency, validated in multi-phase studies across different samples of participants. But that is beyond the

scope of our paper. The second question is also slightly larger than the one we had intended on answering. That is, there may be a general tendency to explore, and the tasks used in the literature may be useful in capturing this tendency, but the task structures and measures may first need to be tweaked and specifically designed to capture the tendency. This too is beyond the scope of our paper. The first question is the one we set out to answer. That is, we want to see whether there is a general tendency to explore that is captured by measures typically used in the literature as measures of exploration. Importantly, and as you've pointed out, the second question may potentially be addressed using our data. However, we think this would be suitable for future work either by using our data (assuming there are enough trials and blocks to retrieve the model parameters) or by designing a study better suited for answering that question; see also Reviewer 2 point 2 who essentially agrees that this would be out of the scope of our paper.

I give some suggestions below and point to the problems that I think should be addressed -- though please feel free to modify my suggestions and re-interpret my comments.

Point 3.2

3.2. Multi-armed bandit (MAB):

If the goal is to measure "tendency to explore" as in the "trade-off between exploration and exploitation", then one should always think also about exploitation when measuring exploration. This raises the equivalent but easier question that "what does it mean to exploit in each behavioral task?". For the MAB, exploiting is to choose what has turned out to be best so far. Therefore, exploration is to choose something other than the so-far best option.

If the outcome is too stochastic, then the so-far best option may change frequently. This implies that an agent that always exploits would switch its actions frequently. The behavior of such an agent is considered by the author's first measure (MAB-Switches) as very exploratory, while the agent (by design) does not explore. So, I would claim that the 1st measure combines the effect of the outcome stochasticity with the exploration. I suggest keeping this measure as a control (since it has been used often before), but the results related to it should not be over-interpreted -- particularly when answering Q2.

The 2nd measure (MAB-Greedy) is, however, well designed. In particular, I think it is useful to note that MAB-Greedy is the maximum likelihood estimate of epsilon for an epsilon-greedy algorithm fitted to data. However, the problem with this measure is that it does not dissociate between the random exploration (RE) and the directed exploration (DE) -- i.e. it interprets both types of exploration as random.

I have a model-based suggestion to fix this issue. Similar to Eq. 1 in supplementary materials (also see * below), let's assume $\pi_{i,t}$ is a subject's estimate of the average reward of arm i at time t . Let us assume that the arm i has been chosen $N_{i,t}$ times until time t . Then, following common practice in neuroscience [2,6] and reinforcement learning theory [7], one can consider (similar to Eq. 1) the following action policy

$$p_{i,t} \propto \exp\left[\tau \cdot \left(\pi_{i,t} + \beta / \sqrt{N_{i,t}}\right)\right],$$

where both τ in $[0, \infty)$ and β in $[0, \infty)$ are free parameters. The parameter τ (as explained by the authors) controls the randomness of the policy, whereas the parameter β

controls a deterministic preference for the less chosen arms. These two parameters can be estimated (using maximum likelihood, like MAB-Greedy) for each participant -- unfortunately, there is no analytic formula for the estimate, but even a simple greed search can do the job for such a simple model. Then, the estimate of tau can be seen as a measure for RE and the estimate of beta as a measure for DE. This will help the interpretation of the results of the analysis for Q1 and Q2.

*: The simulation part in the supplementary material is strange. Based on the model assumption, the only source of exploration is tau. Why are we interested in the correlation between MAB-Switches and MAB-Greedy for a fixed tau? If MAB-Switches and MAB-Greedy have lots of variabilities for a given tau, then it means that they are not basically capturing the exploration. It would be more interesting to look at the correlations between MAB-Switches and MAB-Greedy and tau -- to see how informative they are about tau (i.e. the only source of exploration in the model)

*** *Response.* This is a very thought-provoking comment and thank you for the time and effort it took to provide such detailed points. We agree with you that the switch-rate is important to keep in the study due to its use in past research as a measure of exploration. Indeed, this was our reason for using it. We also agree that the inverse of the best-reply rate, or the MAB-Greedy, is better designed to capture exploratory behaviour, as distinct from exploitative behaviour. This is indeed why we thought to use it, but also because the best-reply rate is used in the literature as a measure of exploitation, presumably the opposite of exploration. We also take your point that this measure doesn't distinguish between random and directed exploration.

We further agree that model-based measures may better capture tendencies for exploration, not just for the bandit task, but also for some of the other tasks that have models from the literature with parameters that reflect exploratory tendencies. However, applying the model-based approach goes beyond the scope of the current study (see also Reviewer 2's point 2, who agrees that the model-based approach would be quite a monumental task in itself). We now write in the manuscript that model-based approaches are a possibility that future work should pursue (see our response to Reviewer 2's point 2).

We analyzed correlations between the MAB-Switches and MAB-Greedy measures for a given tau to assess their consistency relative to one another. That is, we wanted to know for a given level of exploration (tau), how much agreement there would be between the switch-rate and the inverse of the best-reply rate. Hence, in our simulations we were not assessing the extent to which the two measures are reflective of an underlying tendency to explore, but, rather, the consistency between the two measures given the same underlying tendency to explore. We make this clearer in the Supplemental where we added (p. 15):

"In essence, we wanted to know for a given level of τ , how much agreement there would be between the switch-rate and the inverse of the best-reply rate."

Point 3.3

3.3. Alien game (AG):

The two measures (AG-Hamming and AG-Active Search) are interesting. However, they are not necessarily candidates for two "different conceptualizations of exploration" (line 449). The goal of a participant is to find the painting with the highest price, so which strategy he/she chooses to

explore depends on his/her assumptions about the task structure: if symbols are assumed to have semi-independent effects (e.g. aliens always like when the 1st symbol is chosen independently of the rest), then AG-Hamming captures exploration better; but if each combination (painting) is assumed to be completely distinct from the rest, then the 2nd measure one makes more sense. Given the current instruction, I assume different people may make different assumptions. Is it possible to control this? If so, we will most likely observe a reduction in the noise level.

If AG-Hamming is assumed to be a good measure of exploration, then I assume that the rate of decrease in Hamming distance (during one block) should also be an interesting measure -- i.e. how fast they converge to a solution.

*** *Response*. It is correct that the instructions do not make clear that the decision attributes are interdependent to some degree. While the original instructions of the game make this clear, we ended up shortening the instructions of all games to reduce the time needed for the completion of the experiment. In the case of the Alien Game this shortening meant that we deleted this highly relevant aspect, which also Reviewer 2 pointed out. We have now clarified this in the instructions making the interdependence of decision variables as well as the need to make 10 deliberate decisions in every trial clearer. Participants are now told that:

“...you don’t know what the alien likes and how the interconnectedness between the different symbols influences the payoff.” and

“... you must make a decision for every symbol in every trial as the alien considers the entire combination of selected and not-selected symbols.”

One would indeed expect that the rate of decrease in Hamming distance should be an interesting measure. However, many prior experimental studies using the Alien Game have surprisingly not detected such a decrease while subjects were engaging in active search (e.g. Vuculescu 2017; Billinger et al. 2014, 2021; Giannoccaro et al., 2020). Your suggestion is certainly an interesting hypothesis and given the findings from prior experiments future research should investigate conditions under which the Hamming distance decreases within a block. For the present study we decided not to include an additional measure.

Point 3.4

3.4. Optional stopping with recall (OS):

As noted also by the authors (lines 306-307), the measure has a clear drawback of not including the current best option and the cost of opening new boxes. There are multiple ways that they can be included.

*** *Response*. Indeed the reviewer is correct that the stochasticity in the task (and the different rewards experienced by our participants) could affect the quality of the measures. To counteract this issue we opted for a larger number of blocks for the task (8 in total, for 4 different cost levels). We did two additional analyses to make sure that the measure is reliable. First, we ran a simulation in which we assumed that agents varied in their stopping thresholds (which in turn is determined by the cost of search, also see Bhatia et al. 2021) and assessed the reliability of the measure we propose to use (i.e., number of boxes opened). These simulations showed that the measure is

highly reliable. Additionally, we looked at Cronbach's alpha (as for the other measures), finding that the number of opened boxes in the optional stopping task was particularly reliable (see response to Reviewer 4 point 2). This suggests that our design sufficiently addresses the potential drawbacks.

An alternative way to measure exploration in this task would be to use average stopping thresholds used by the participants by averaging the values of the best options opened before stopping. We would expect that people with a higher tendency to explore would use a higher stopping threshold (and therefore they would open more boxes on average). If indeed this is the case we would expect the measures to overlap to a great extent. That said, we believe that the number of opened boxes is more direct and easy to communicate and therefore we decided to keep on using that.

Point 3.5

3.5. Sampling paradigm (SP)

The goal of an agent is to find the best option. So, the measure of exploration should also include how different the two options are -- if the difference is too big, then with a smaller number of sampling one can find the better option and get confident about it.

**** Response.* For this task we also use a measure that is often used in the literature as a measure of exploration. It goes beyond the aims of our paper to change the measure. Moreover, even given your point, people who presumably have a stronger tendency towards exploration should sample more relative to people with weaker tendency, regardless of differences between the options. Given the correlations for the reliability analyses (i.e., between the different blocks of the task) and the variability and range of scores in the pilot study, we believe that there is sufficient inter-individual variability. In other words, the differences between the options should not hide individual differences in the tendency to explore in the task.

Point 3.6 [note that the reviewer numbered this point as 3.5]

3.6. Observe or bet (OB)

Same as SP: the difference between estimated probabilities should be taken into account.

**** Response.* Our response to this point is the same as for the sampling paradigm. Indeed, the number of observe trials is the typical measure of exploration in this task.

Point 4

4. The plausibility of answering Q2

Common factor analyses have explicit and clear assumptions on the functional form of the dependence of the observable variables on the common factors. However, as discussed in point 3, there may be many non-linearities involved in the transformation of a hypothetical tendency to explore into any of the 11 observable exploration measures. Therefore, it is important to note that not being able to find any common factor does NOT imply that "The behavioral measures are capturing task-specific behaviors" (c.f. Table 1, Q2, column 5). It should rather be interpreted as "We could not find a task-independent variable that ...". This answer would still be interesting, but it would be more accurate.

**** Response.* You're correct. We will adjust our interpretation accordingly. Namely, we note in Table 2b that if no two behavioural measures and no combination of behavioural and self-report measures fit onto the same common factor, then we can only conclude that "we did not find evidence for a latent variable that captures a domain general behavioural tendency across tasks". See also p. 40 of manuscript.

Point 5

5. Statistical analyses

- I suggest (in particular to answer Q1) using Bayesian methods for hypothesis testing in addition to the proposed frequentist methods -- e.g. see [8] for an example of Bayesian correlation test. In this sense, you avoid classic problems of frequentist hypothesis testing (such as the tendency to reject null hypotheses and issues for multiple comparisons) and you can also accept the null hypothesis (the lack of correlation) and make more interesting conclusions.

**** Response.* With frequentist methods, though we can't accept the null, we can test for equivalence with a given range of effect sizes, which we've planned on doing and which often provides similar answers as to when using Bayes factors (Lakens et al., 2020). Nonetheless, following your suggestion, we think it's a good idea to complement the correlation tests with default Bayesian test for correlation using the BayesFactor package in R with JZS default priors (Wetzels & Wagenmakers, 2012). However, we believe this fits best in exploratory analyses for the Stage 2 submission.

- Having a linear relationship between different exploration measures is a strong assumption (see points 3 and 4). Particularly, this assumption becomes important if we go beyond interpreting the correlation sign and study the effect size (e.g. comparing the correlation with 0.2 for Q1). Therefore, I suggest reporting also the Spearman or the Kendall correlation.

**** Response.* You're right in so far as the assumption of linearity may not hold. Therefore, in exploratory analyses, we will also report the Spearman correlation for each pair of measures and interpret the results accordingly.

- For correction for multiple comparisons, I am not convinced by how a "family" of hypotheses is defined for Q1. I think the correction should be done according to the total number of hypotheses, i.e. $11 * 10 / 2 = 55$. Also, is there any reason that the authors chose to correct FWER instead of FDR (c.f. [9])?

*** *Response*. There are many reasonable methods to correct for multiple comparisons. And we see the benefits of what you suggest, namely using Bonferroni corrections treating every single test as part of the same family of tests. Therefore, we've changed the correction for multiple comparisons. The corrected alpha for each pair of correlations is now $(.05/55)$.0009. For 95% power to detect a correlation of at least .2, with a corrected alpha of .0009, one-sided correlation tests require a sample of 556. The relevant sections of the manuscript, such as the analysis plan for hypothesis 1, have been changed accordingly, including the target sample size.

References:

- [1] Kobayashi et al. (2019), DOI: 10.1038/s41562-019-0589-3
- [2] Xu et al. (2021), DOI: 10.1371/journal.pcbi.1009070
- [3] Gottlieb and Oudeyer (2018), DOI: 10.1038/s41583-018-0078-0
- [4] Schulz and Gershman (2019), DOI: 10.1016/j.conb.2018.11.003
- [5] Dubey and Griffiths (2020), DOI: 10.1016/j.cobeha.2020.07.008
- [6] Jaegle et al. (2019), DOI: 10.1016/j.conb.2019.08.004
- [7] Strehl and Littman (2008), DOI: 10.1016/j.jcss.2007.08.009
- [8] Rouder and Morey (2012), DOI: 10.1080/00273171.2012.734737
- [9] Efron and Hastie (2016), link: https://web.stanford.edu/~hastie/CASI_files/PDF/casi.pdf

Reviewer # 4

The current paper proposes a registered report to examine convergent validity between different behavioral and self-report exploration measures. I am reviewing this proposal as an expert in open science practices (including registered reports), along with expertise in personality psychology and measurement. I am not an expert in this particular trait domain, nor am I familiar with most of these tasks and measures.

The authors propose an ambitious design to test an interesting research question. There are many positives about the design, notably a large sample size, together with a thoughtful approach to a variety of exploration tasks and measures. I particularly appreciate that the authors have already shared many research materials and have a plan to share all study data and analysis code on study completion. However, there are some substantial weaknesses in the analysis plan. I outline the major issues below.

Point 1.

1. Most importantly, the study design proposed is of the multi-trait, multi-method (MTMM) variety, yet the authors do not plan to handle the data in the appropriate way to test models from a study with this design. To learn more about this approach, I can recommend the relevant chapter in the Kline structural equation modeling textbook. Many studies have used this approach, but one that might be similar to the study proposed here is Bar-Anan & Vianello (2018, JEP:G). Essentially, rather than testing each correlation one-by-one with an arbitrary 0.20 cutoff, you will instead want to test nested models with a variety of competing structures against each other. In this way, you will be able to test convergent validity alongside questions about the trait-like-ness of these measures in one succinct approach.

Given the points you've raised (including below), we've now reconsidered the analysis plan. However, the MTMM method is not applicable to our case. Although our design is multi-method it is not multi-trait and therefore the MTMM method cannot be applied as such. Therefore, we retain the pairwise correlational analyses for tests of convergent validity between each pair of measures. However, we've updated our modeling approach for the exploratory and confirmatory factor analyses. To do this, we included a new collaborator on the project who has expertise in psychometrics and measurement theory. The updated analysis plan is fully detailed in the manuscript. The updated factor analyses for Hypothesis 2 are on pp. 35-41. For your convenience, we've reproduced the factor analyses plan below, at the end of our response to your final comment under the heading "Updated Analysis Plan for Hypothesis 2".

Point 2

2. The authors' plans for assessing and dealing with measurement error are underspecified in the current plan. There is a plan to examine test-retest reliability (good), but there is no mention of any plan to examine internal consistency for either self-report or behavioral measures. Notably, when you want to compare two different types of tests to one another, it is important to consider whether or how measurement error might be different between them. The behavioral tasks that are outlined in the proposal have a variety of different scoring strategies (and numbers of trials) that lend themselves to very different potential for unreliability. Most, but not all, of the behavioral measures involve averaging performance across a number of trials and/or blocks. It would be interesting to know whether scores are consistent (i.e., positively correlated) within these different parts prior to aggregation. You could even model the trials like you might for individual self-report items in the MTMM approach. Even if you stick with your planned scoring strategy rather than modeling individual trials or blocks, you will want to assess consistency prior to aggregation. If the behavioral measures are less reliable than the self-report measures, it will depress correlations among the behavioral measures and between the behavioral measures and the self-report scales.

We agree with your points about checking the internal reliability of the tasks in the main study. We already considered this for the pilot data but hadn't incorporated such checks for the main analyses. Here is what we reported in the manuscript for the pilot data (p. 13):

"Moreover, we examined the internal consistency of each measure of exploration in the behavioural tasks. We did this by correlating the measure of exploration in each task across different blocks. For example, we calculated the switch rate for the bandit task using only the odd numbered blocks (i.e., blocks 1 and 3) and correlated this with the switch rate in the even

numbered blocks (i.e., blocks 2 and 4). Apart from one combination of blocks for active search in the alien game, for which the small sample size produced large uncertainty around the estimates, all measures showed good internal consistency (see Supplemental Information).”.

As we noted in our response to Reviewer 1 point 6, the correlations were: switch rate and complement of the best reply rate in MAB (.77-.91); optional stopping task (.61 and .67); sampling paradigm (.50-.65); observe or bet task (.60); hamming distance in alien game (.55-.75); and active search in alien game (.39-.59). The latter measure showed lower reliability for one of the analyses but this was likely due to the alien game having fewer participants ($n = 20$) as a result of a software error that resulted in dropping some data.

We now realize that perhaps a better test of internal consistency is to calculate Cronbach’s alpha for each task with each block treated as an observation. For the main study, we therefore propose to calculate Cronbach’s alpha for each measure as an indicator of internal consistency. It should be kept in mind that because the behavioural tasks involve probabilistic payoffs, the internal consistency is unlikely to be as high as in self-reports. More importantly, though, self-reports will tend to have higher internal consistency because they are often contaminated with common methods variance (e.g., consistent responding, social desirability, etc.) which behavioural tasks are not prone to, at least not to the same extent. Hence, the internal consistency of the behavioural tasks will naturally be lower than those typically observed and desired for self-reports. This will indeed depress the correlations between the different behavioural measures and between the behavioural and self report measures. Nonetheless, our study design (e.g., sample size) is such that the tests have high statistical power to detect substantially smaller effect sizes than are typically expected in convergent validity tests. Moreover, there is no concrete rule about satisfactory Cronbach’s alpha cutoffs. All cutoffs are generally rules of thumb, selected based on somewhat subjective standards, and are suggested to be used as “guides”.

Given all of the above, we’ve added the following text to the manuscript in the newly added “Positive controls and reliability analyses” section in the “Analysis Plan”, under the heading “Internal consistency” (p. 30-32):

“We will examine the internal consistency of each behavioural measure to determine whether it would be problematic to aggregate the measure across the different blocks of the task. This is because low internal consistency will deflate the correlation between a measure and all other measures. To assess internal consistency, we will calculate Cronbach’s alpha using each block of a task as an observation point (like an item on a self-report scale). Kline (p. 92)⁹⁶ states that Cronbach’s alpha values of about .70 may be considered as adequate but that these classifications are merely “some guidelines”. Importantly, such guidelines are typically derived from researchers’ experience with self-reports, and self-reports are prone to many types of common methods variance (e.g., consistent responding, social desirability, acquiescence responding, etc)⁹⁷ that spuriously inflate inter-item correlations and thus Cronbach’s alpha⁹⁸. In contrast, behavioural tasks have not been shown to be consistently prone to these types of method variance, and though they may be prone to other types of common method variance, we think it unlikely that the correlations between blocks of a task would be as spuriously inflated as the correlations between self-report items. Moreover, it may be that some of the behavioural tasks will have lower internal consistency due to stochasticity (e.g., luck of the draw), though we have made design choices to mitigate this issue. In sum, we will use a Cronbach’s alpha cutoff of .55 as an indicator of satisfactory internal consistency for the behavioural measures. If a behavioural measure shows good internal consistency by this standard, we will aggregate across the blocks of the task and test convergent validity with the correlational analyses as originally planned.”

“However, if a behavioural measure shows poor internal consistency, such as a Cronbach’s alpha below .55, then we will use a formative modeling approach (see Diamantopoulos et al., 2008; Diamantopoulos & Siguaw, 2006) to do an additional test of its convergent validity. In formative models, it’s assumed that the observed variables are the causes of the construct of interest, rather than the construct being the cause of the variables. As such, in formative models, there is no requirement for the observed variables that make up a construct to be correlated with one another (i.e., no requirement for internal consistency). To test convergent validity and domain generality using formative models, we will examine the correlations between the latent variables in a structural equation model where the behavioural measure(s) that have poor internal consistency are modeled as formative latent factors (i.e., the scores on the blocks of that measure cause the latent factor) and the other measures are modeled as reflective latent factors (i.e., the latent factor causes the observed scores on the blocks of that measure). Specifically, we will create a composite score for the behavioural task with poor reliability using principal components analysis (PCA) and test how this composite score correlates with the other measures. In PCA, the component loadings for each block of the task will be used as weights which are then used to calculate the composite score across blocks. Thus, the composite score will be a weighted sum score.”

“Importantly, we will still examine the Pearson correlations after aggregating scores across blocks. This is because the measures we have proposed, including their aggregation across trials and blocks, are as they are used in the literature. It is therefore important to examine whether the measures, as used in the literature, have convergent validity with one another.”

“For the self-reports we will use a Cronbach’s alpha of .70 as the threshold for satisfactory internal consistency. If a self-report measure shows unsatisfactory internal consistency, then we will (i) remove items until the measure satisfies our threshold before creating a composite score to test for convergent validity and (ii) remove these same items from the factor analyses tests of domain-generality. If even after removing the worst items reliability does not achieve the .70 threshold, we will use PCA to calculate the composite scores with all the items and use the component’s loadings to evaluate the importance of each item to the composite. Importantly, we will still examine the correlation of the full scale unweighted composite scores to examine whether the measure, as originally validated and currently used in the literature, has convergent validity; and we will do similarly for examining domain-generality.”

As a minor note, we calculated the Cronbach’s alpha for each measure in the pilot data and added the following to the main paper in the “Pilot Data and Simulations” section (p. 13):

“A better measure of internal consistency is the Cronbach’s alpha. We therefore calculated the Cronbach’s alpha for each measure of the behavioural tasks, using each block as an observation. All measures showed satisfactory internal consistency as measured by Cronbach’s alpha, though some had wide (95%) confidence intervals due to the small sample size: switch rate = .91 [.80, .95]; best-reply-complement = .56 [.14, .75]; Hamming = .87 [.60, .95]; active search = .67 [.25, .85]; observe or bet = .75 [.27, .92]; optional stopping = .80 [.48, .90]; samples = .74 [.56, .83]. Importantly, as detailed in the “Internal consistency” section of the “Analysis Plan”, we expect behavioural tasks to show lower internal consistency than self-reports, particularly because of the stochasticity underlying some tasks, and so shouldn’t be held to the same standards on this measure of reliability. Therefore, we believe that even the best-reply complement shows promise for satisfactory internal consistency.”

Point 3

3. Laudably, the authors intend to examine test-retest reliability of the measures. This is a good idea. However, as with the missing plan for dealing with measurement error, there is a missing plan here for dealing with potential measurement invariance. Again, there are directions in the Kline book for how to approach this. The key issue is that you want to check whether the factor structure for the measurement of each task or scale is consistent over time.

Following your suggestion, we've added the following to the "Positive controls and reliability analyses" section of the "Analysis Plan", under the heading "Measurement invariance" (p. 32-33):

"Because we will have measurements at two different points in time for a large group of our participants (see Hypothesis 3), we can test for measurement invariance in regards to time. There are two ways to achieve that: either testing each task individually for measurement invariance, or by testing the structural equation model derived from the factor analyses. We opted for the second approach because this it provide important information about the measurement model selected from the factor analyses."

"We will test the model selected from the factor analyses for Hypothesis 2 for different levels of invariance to time using nested likelihood ratio tests of the configural, weak factorial, strong factorial, and strict variance models. We will set the level of significance at 0.02, as this approach involves sequential testing. The 0.02 level was derived from a Bonferroni correction of the usual 0.05 level of significance. If at any level we cannot establish invariance for the entire construct, we will use the bias-corrected bootstrap confidence intervals approach to identify and remove any problematic tasks and try to establish partial measurement invariance for the remaining tasks at the same level. The bias-corrected bootstrap confidence intervals approach to measurement invariance is centered on evaluating the confidence intervals for all the differences between parameters in the model. If a confidence interval does not include zero, it can be inferred that the specific parameter is not invariant."

"For example, if we can establish weak factorial invariance for the entire model, but not strong factorial invariance, we may remove any tasks that do not satisfy strong factorial invariance in order to establish partial strong invariance for the model in the remaining tasks. If we cannot establish even partial measurement invariance for the full measurement model, then we will assume that the model is unlikely to be temporally stable. We will, nonetheless, examine the test-retest reliability of each measure using the pairwise correlations for Hypothesis 3."

Point 4

4. The authors propose to use a fixed order, with behavioral tasks before self-report tasks. I'm not sure this is a good idea, given the potential for fatigue over the long protocol. (By the way, can participants really do all five tasks and all the self-report measures in an hour?)

We've now decided to randomize the order, between participants, of whether they first complete the behavioral tasks or the self-report scales (see our response to Reviewer 2 point 3).

Participants in the pilot study took an average of 63 minutes to complete everything. The fastest time was 21 minutes and the slowest was 135 minutes. We note that the study involved a few self-report measures that we haven't included in the registered report (these are *some* facets from the

BFI-2). Moreover, the behavioral tasks are diverse and they have been designed to resemble small games so that they are engaging, rather than tedious and boring. In fact, 15 of the 38 participants from the pilot studies explicitly stated that the study was fun and/or enjoyable, and this is not counting those who said that it was interesting.

Based on the experience of the authors in using Prolific, the addition of the remainder of the BFI-2 (i.e., another 40 self report items) should add only about another 7 minutes to the average time participants take to complete the study.

Point 5

5. As a personality psychologist, I note the absence of other self-report measures, such as “openness to experience” in the big five framework. Typically, openness and extraversion are where we think about exploration fitting in. This is just a note to mention this absence, not a request to include a measure. But the authors might find relevant literature in this domain as well.

Thanks for bringing this to our attention. We had already included a few facets from the BFI-2 for exploratory purposes. Your comment has made us reconsider to include the entire BFI-2, because we had only the energy level facet from extraversion and the intellectual curiosity and creative imagination from open-mindedness (plus two facets from conscientiousness). This addition is possible because people typically don't take long to complete these self-report measures and our study, with the times we listed in our response to your previous comment, already included participants responding to 5 facets from the BFI-2 (20 items). We now realize that including the full BFI-2 (60 items) will be valuable not only for our own exploratory analyses but also for other researchers who may reuse the data in future. Nonetheless, we will use the personality variables in exploratory analyses.

Point 6

6. As a general point, there are many decisions that need to be made to have a well-specified registered report. It seems likely to me that additional issues will come up and need to be resolved after changes are made the analysis plan. That is, I anticipate raising more queries on the next round of review should the authors be given a chance to revise. (As an example, registered reports should have positive controls if possible. This means including checks to ensure that data are of expected quality; e.g., perhaps you can already be reasonably confident that two or more of the self-report measures will correlate strongly with each other. If such a correlation were not present it would indicate some severe issue in the data.)

We will make any changes that we believe will improve the quality of the study. As one type of quality check, we make sure that participants pass the comprehension checks for the behavioural tasks' instructions before they can continue with the study. This is to ensure that they read and understand the tasks' instructions. Additionally, the example you've provided is a good one to anticipate. Hence, in the newly added “Positive controls and reliability analyses” section in the “Analysis Plan”, we now state that (p. 30):

“The maximization tendency scale and the alternative search subscale from the maximization inventory should be positively correlated. Indeed, this is shown in past research ($r = .38$; Turner et al., 2012) and in our pilot data ($r = .8$). Therefore, as a positive control in our main analyses, we

expect that the maximization tendency scale and the alternative search scale will be positively correlated by at least $r = .3$. If the correlation between these two scales is too low then the data on the self-reports will be suspect. In that case we will conduct the correlational and factor analyses with and without the self-reports as a type of sensitivity check and draw conclusions accordingly.”

Point 7

Smaller points:

7. The authors write “we want to use data from participants motivated to complete the study.” Why is this the case? I didn’t understand the logic behind this exclusion.

To clarify the reason for the exclusion rule, we’ve included the following in the manuscript where this rule is presented (p. 29):

“Specifically, participants only receive performance-based bonuses if they complete the entire study. And monetary incentives are known to create differences in effort and to change behaviour in some judgment and decision tasks (Camerer & Hogarth, 1999; Voslinsky & Azar, 2021). This is particularly important for tasks that participants can choose to end without putting any effort in exploring the different options (i.e., sampling paradigm, optional stopping task). Participants who drop out may, furthermore, have been less motivated to do well in the tasks they completed, particularly if they drop out intentionally due to having other demands on their time during the online experiment. Therefore, the difference in receiving vs not receiving incentives, in addition to likely differences in the motivation of participants to complete vs not complete the study, may create noise in the data and thus reduce statistical power to detect covariation between the different measures of exploration.”

Point 8

8. I have not seen EFA interpreted using model fit statistics. Typically, people use parallel analysis (good) or some other rule like Kaiser’s criterion (less good). This might not be of consequence if you change your modeling strategy to the MTMM technique.

Thank you for bringing our attention to this. As per our response to your point 1, we have fully updated the analysis plan for Hypothesis 2, which includes the EFA. The analysis plan is fully detailed below, at the end of your review, under the heading “Updated Analysis Plan for Hypothesis 2”.

Point 9

9. The paper is not written to be understandable by an outsider. For instance, “multi-armed bandit” is mentioned several times without explanation in the introduction.

Reviewer 2 raised a similar concern in their point 1. We’ve therefore now added a (new) Table 1 that briefly explains the different behavioural tasks we reference in the introduction section (see response to Reviewer 2 point 1). The Table also mentions whether the task is included in the present study.

I sign all of my reviews,

Katherine S. Corker

Updated Analysis Plan for Hypothesis 2

This section reproduces text from the manuscript (pp. 35-41).

Hypothesis 2. Our analyses for testing Hypothesis 2 includes three baseline confirmatory models and three exploratory data-driven models. For these analyses, we will randomly divide the data into two roughly equal parts (approx. $N = 350$ each). With one half of the data, we will first derive three exploratory data-driven models. With the other half of the data, we will then compare all six models (the three baseline confirmatory models and the three data-driven models) against each other with confirmatory factor analysis (CFA). For these analyses, we will use the aggregated scores of each measure as described in the “Design” section (e.g., the switch-rate aggregated across the different blocks of the multi-armed bandit task, and the mean ratings on the items of the exploration scale).

The baseline models represent three basic theoretical assumptions and are depicted in Figures 1-3. In the first baseline model, Baseline1, the explained variance is accounted for by only one common underlying factor (i.e., the tendency to explore). In Baseline2, the explained variance is accounted for by only two specific oblique factors reflecting the methodologies of measurement (i.e., behavioral and self-report measures). In Baseline3, the explained variance is accounted for by two orthogonal factors reflecting common method variance for the behavioral tasks and the self-report measures, and one common underlying factor representing the tendency to explore. In the first exploratory model, Exploratory1, we will use both exploratory graph analysis and parallel analysis combined with exploratory factor analysis to derive a simple structure model. In Exploratory2, the explained variance is accounted for one common underlying factor representing the tendency to explore in addition to the factors derived for Exploratory1. In Exploratory3, we will select between bifactor exploratory models with a single bifactor according to the interpretability of the largest loadings. Now we describe the models in more detail, including how the data-driven models will be derived, and then we detail how we will do the model comparisons in the CFA.

Baseline1. The first baseline model assumes a general factor representing the tendency to explore, thus excluding any methodological effect of the elicitation methods of explorative behavior. Figure 1 visualizes Baseline1.

Figure 1. Note. E: general factor. SwR: multi-armed bandit switch-rate. BRC: multi-armed bandit best-reply complement. AcS: alien game active search. HaD: alien game Hamming distance. OSW: optional stopping with recall. Smp: sampling paradigm. OoB: observe or bet. ExS: exploration scale. MTS: maximization tendency scale. AIS: alternative search. GEE: general explore-exploit question.

Baseline2. The second baseline model, visualized in Figure 2, assumes the existence of two specific factors, one associated with the self-report measures and the other associated with the behavioral measures. Specifically, the explained variance is accounted for by only two specific oblique factors reflecting the methodologies of measurement (i.e., behavioral and self-report measures). This model thus assumes that the tendency to explore is expressed differently in the two broad categories of measures, best captured by two separate factors instead of one. The two factors are allowed (and expected) to be correlated with each other.

Figure 2. Note. BT: behavioral-tasks specific factor. SR: self-reports specific factor. SwR: multi-armed bandit switch-rate. BRC: multi-armed bandit best-reply complement. AcS: alien game active search. HaD: alien game Hamming distance. OSW: optional stopping with recall. Smp: sampling

paradigm. OoB: observe or bet. ExS: exploration scale. MTS: maximization tendency scale. AIS: alternative search. GEE: general explore-exploit question.

Baseline3. The third baseline model, visualized in Figure 3, relies on the assumption that the different measures are affected by a single common factor reflecting the general tendency to explore, but part of the observed variances are also explained by common method variance. Specifically, the explained variance is accounted for by two orthogonal factors reflecting common method variance for the behavioral tasks and the self-report measures, and one common underlying factor representing the tendency to explore.

Figure 3. Note. BT: behavioral-tasks common method factor. SR: self-report measures' common method factor. E: general factor. SwR: multi-armed bandit switch-rate. BRC: multi-armed bandit best-reply complement. AcS: alien game active search. HaD: alien game Hamming distance. OSW: optional stopping with recall. Smp: sampling paradigm. OoB: observe or bet. ExS: exploration scale. MTS: maximization tendency scale. AIS: alternative search. GEE: general explore-exploit question.

Exploratory1. The first exploratory model will be derived using two methods for assessing the dimensionality of an instrument (i.e., the inferred number of latent common causes): (i) exploratory graph analysis (Golino & Epskamp, 2017) and (ii) parallel analysis combined with exploratory factor analysis. Exploratory graph analysis will tell us the number of factors that can be extracted and also the measures that pertain to each factor. In contrast, parallel analysis only tells us the number of factors that can be extracted, but not which measures pertain to which factors. Therefore, in a second step after the parallel analysis, we will use exploratory factor analysis with the number of factors predetermined by the parallel analysis. In the exploratory factor analysis, we will consider a measure to be derived from a factor if its absolute loading on the factor is at least 0.30—this threshold is within the bounds of what has been recommended (Howard, 2016) and

done (Frey et al., 2017) in the literature. The model thus derived from the combination of parallel analysis and exploratory factor analysis will be compared with the model derived from the exploratory graph analysis. In this comparison, the most interpretable between these two models will be selected as Exploratory1. Interpretability here refers to what makes sense from a theoretical perspective. Essentially, we will choose the model with the most parsimonious theoretical justification. The theoretical justification will be based on, for example, the structural similarities and differences between the tasks noted in Table 4. The selected model will be Exploratory1 and the factors will also be used as one component of the Exploratory2 model.

Exploratory2. The second exploratory model assumes that there is a single general factor that captures people's tendency to explore, as with Baseline1. However, we will also keep the factors identified in Exploratory1. Exploratory2 will therefore be a bifactor model with a general tendency to explore that loads to all measures but that also includes the factors identified in Exploratory1 loading to their respective measures.

Exploratory3. The third exploratory model will be derived using exploratory bifactor analysis (Jennrich & Bentler, 2011, 2012). Because there is no dimensionality approach for bifactor models, we will select the "best" model from the exploratory bifactor analysis based on its interpretability regarding the loadings of the tasks to each extracted factor. As described for Exploratory1, our model selection based on interpretability depends on what makes the most sense from a parsimonious theoretical perspective. The selected model will be Exploratory3.

Confirmatory factor analysis: Model comparisons. By now, we will have used a random half of our data to derive the above exploratory models. We will then use the remaining random half of our data to fit the six models described in the preceding paragraphs (i.e., Baseline1, Baseline2, Baseline3, Exploratory1, Exploratory2, and Exploratory3) and compare them using CFA. For the EFA-derived models (i.e., Exploratory1-3), absolute loadings below 0.30 will be fixed as 0. If the data from the tasks are normally distributed, parameter estimation will be done with robust Maximum Likelihood estimation (MLR). If the normality assumption does not hold, parameter estimation will be done with the Weighted Least Square Mean and Variance Adjusted Estimators (WLSMV; e.g., Suh, 2015).

The models will be compared with regard to fit indices commonly used in the context of model comparison in factor analysis: CFI; TLI; and RMSEA. The best fitting model will therefore be the one with CFI and TLI closest to 1 and RMSEA closest to 0. If two or more models provide a very similar fit to the data, the model we select will be the one that makes the most sense from a parsimonious theoretical perspective. For instance, Baseline3 may have better fit indices than Baseline1, even though Baseline1 shows good enough fit indices. In this case, the latter will be the most parsimonious from a theoretical perspective. That is, it would be reasonable to assume that the unidimensional model (i.e., Baseline1) is a better description of the data. The selected model will also be evaluated for measurement invariance using the analysis plan described earlier.

If the best fitting models in the CFA are those in which no two behavioural measures, or no combination of behavioural and self-report measures, fit well onto the same factor, or if we fail to identify any such models that fit the data well, then we will conclude that we did not find evidence for a latent variable that captures a domain general behavioural tendency across tasks. There will be evidence for a domain-general tendency to explore, *E*, to the extent that multiple behavioural measures (or a combination of behavioural and self-report measures) fit well onto a common factor together. The strongest evidence for *E*, and the validity of the measures in capturing it, will be if a single factor can be extracted that explains variance in all of the 11 measures of exploration.

References

- Bhatia, S., He, L., Zhao, W. J. & Analytis, P. P. (2021). Cognitive models of optimal sequential search with recall. *Cognition* **210**, 104595.
- Billinger, S., Srikanth, K., Stieglitz, N. & Schumacher, T. R. (2021). Exploration and exploitation in complex search tasks: How feedback influences whether and where human agents search. *Strategic Management Journal*, *42*, 361–385.
- Billinger, S., Stieglitz, N. & Schumacher, T. R. (2014). Search on rugged landscapes: An experimental study. *Organization Science*, *25*, 93–108.
- Camerer, C. F. & Hogarth, R. M. (1999). The Effects of Financial Incentives in Experiments: A Review and Capital-Labor-Production Framework. *Journal of Risk and Uncertainty*, *19*, 7–42.
- Denrell, J., Fang, C. and Levinthal, D.A., 2004. From T-mazes to labyrinths: Learning from model-based feedback. *Management Science*, *50*(10), pp.1366-1378.
- Diamantopoulos, A., Riefler, P. & Roth, K. P. (2008). Advancing formative measurement models. *Journal of Business Research*, *61*, 1203–1218.
- Diamantopoulos, A. & Siguaw, J. A. (2006). Formative Versus Reflective Indicators in Organizational Measure Development: A Comparison and Empirical Illustration. *British Journal of Management*, *17*, 263–282.
- Frey, R., Pedroni, A., Mata, R., Rieskamp, J., & Hertwig, R. (2017). Risk preference shares the psychometric structure of major psychological traits. *Science Advances*, *3*(10), e1701381.
- Ghemawat, P. and Levinthal, D., 2008. Choice interactions and business strategy. *Management Science*, *54*(9), pp.1638-1651.
- Giannoccaro, I., Galesic, M., Massari, G. F., Barkoczi, D. & Carbone, G. (2020). Search behavior of individuals working in teams: A behavioral study on complex landscapes. *Journal of Business Research* *118*, 507–516.
- Golino, H. F., & Epskamp, S. (2017). Exploratory graph analysis: A new approach for estimating the number of dimensions in psychological research. *PLoS ONE*, *12*(6), e0174035.
- Gorrell, G., Ford, N., Madden, A., Holdridge, P. & Eaglestone, B. (2011). Countering method bias in questionnaire-based user studies. *Journal of Documentation* *67*, 507–524.
- Heck, R., Vuculescu, O., Sørensen, J.J., Zoller, J., Andreasen, M.G., Bason, M.G., Ejlertsen, P., Elfasson, O., Haikka, P., Laustsen, J.S. and Nielsen, L.L., (2018). Remote optimization of an ultracold atoms experiment by experts and citizen scientists. *Proceedings of the National Academy of Sciences*, *115*(48), pp.E11231-E11237.
- Hills, T. T. & Dukas, R. (2012). The evolution of cognitive search. in *Cognitive search: Evolution, algorithms, and the brain* 11–24 .
- Hills, T. T., Todd, P. M. & Goldstone, R. L. (2008). Search in external and internal spaces: Evidence for generalized cognitive search processes. *Psychol. Sci.* *19*, 802–808.
- Hills, T. T., Todd, P. M. & Goldstone, R. L. (2010). The central executive as a search process: Priming exploration and exploitation across domains. *J. Exp. Psychol. Gen.* *139*, 590–609.
- Hills, T. T., Todd, P. M., Lazer, D., Redish, A. D. & Couzin, I. D. (2015). Exploration versus exploitation in space, mind, and society. *Trends in Cognitive Science*, *19*, 46–54.

- Howard, M. C. (2016). A review of exploratory factor analysis decisions and overview of current practices: What we are doing and how can we improve?. *International Journal of Human-Computer Interaction*, 32(1), 51-62.
- Jennrich, R. I., & Bentler, P. M. (2011). Exploratory bi-factor analysis. *Psychometrika*, 76(4), 537-549.
- Jennrich, R. I., & Bentler, P. M. (2012). Exploratory bi-factor analysis: The oblique case. *Psychometrika*, 77(3), 442-454.
- Kline, R. B. *Principles and practice of structural equation modeling*. (The Guilford Press, 2016).
- Lakens, D., McLatchie, N., Isager, P. M., Scheel, A. M., & Dienes, Z. (2020). Improving inferences about null effects with Bayes factors and equivalence tests. *The Journals of Gerontology: Series B*, 75(1), 45-57.
- Podsakoff, P. M., MacKenzie, S. B., Lee, J.-Y. & Podsakoff, N. P. (2003). Common method biases in behavioral research: A critical review of the literature and recommended remedies. *Journal of Applied Psychology* 88, 879–903.
- Sang, K., Todd, P. M., Goldstone, R. L., & Hills, T. T. (2020). Simple Threshold Rules Solve Explore/Exploit Trade-offs in a Resource Accumulation Search Task. *Cognitive science*, 44(2), e12817.
- Song, Mingyu, Zahy Bnaya, and Wei Ji Ma. (2019). "Sources of suboptimality in a minimalistic explore–exploit task." *Nature Human Behaviour*, 3.4, 361-368.
- Suh, Y. (2015). The performance of maximum likelihood and weighted least square mean and variance adjusted estimators in testing differential item functioning with nonnormal trait distributions. *Structural Equation Modeling: A Multidisciplinary Journal*, 22(4), 568-580.
- Turner, B. M., Rim, H. B., Betz, N. E. & Nygren, T. E. (2012). The Maximization Inventory. *Judgment and Decision Making*, 7, 48–60.
- von Helversen, B., Mata, R., Samanez-Larkin, G. R. & Wilke, A. (2018). Foraging, exploration, or search? On the (lack of) convergent validity between three behavioral paradigms. *Evolutionary Behavioral Science*, 12, 152–162.
- Voslinsky, A. & Azar, O. H. (2021). Incentives in experimental economics. *Journal of Behavioral and Experimental Economics*, 93, 101706.
- Vuculescu, O. (2017). Searching far away from the lamp-post: An agent-based model. *Strategic Organization*, 15, 242–263.
- Wetzels, R., & Wagenmakers, E. J. (2012). A default Bayesian hypothesis test for correlations and partial correlations. *Psychonomic bulletin & review*, 19(6), 1057-1064.
- Wilke, A., Hutchinson, J. M. C., Todd, P. M. & Czienskowski, U. (2009). Fishing for the right words: Decision rules for human foraging behavior in internal search tasks. *Cognitive Science* 33, 497–529.

Reviewers' Comments:

Reviewer #1 (Remarks to the Author):

The authors have very carefully addressed my comments (and the comments of the other reviewers). I do not have any further comments or suggestions and I recommend accepting the manuscript for Stage 1.

Reviewer #2 (Remarks to the Author):

I would like to thank the authors for the detailed replies to the comments from myself and other reviewers. The authors have satisfactorily considered my concerns, and I endorse acceptance for stage 1.

Reviewer #3 (Remarks to the Author):

I thank the authors for their hard work and believe that the revised report has significantly improved.
The authors have either successfully addressed my points or convinced me that what I have asked for was beyond the scope of their work – which is now more clearly discussed in the text.

In general, I enjoyed reading the manuscript, and I am looking forward to seeing the final results of the analyses.

I only have two minor comments that you can find below; I do not demand seeing a further revised report before going to the 2nd stage.

Minor comment 1:

The new analysis plan for testing hypothesis 2 is promising.
I understand that the plan follows the common practice in frequentist statistics, but, as someone with a background in machine learning, I was wondering why you do not use the 1st half of the data also for fitting the parameters of the 6 models (in addition to deriving the 3 exploratory models)?

Then you can use the 2nd half only for 'testing' and comparing the models.
I would be more convinced by the results of such analyses.

Minor comment 2:

I think the newly added section "Selection of behavioural tasks" (lines 441 to 569) and particularly the paragraph on the usage of computational methods for measuring exploration (lines 565 to 569) are well and fairly written and put the manuscript in the right context. In the current version, I could not find any reference to studies that used computational models of exploration to explain experimental data. Because the part has also a purpose of reviewing the literature, I think it would increase the strength and quality of the manuscript if the authors point readers to some relevant examples in the final version of the manuscript, e.g.,

- WK Zajkowski, M Kossut, RC Wilson (2017), eLife 6:e27430. DOI: 10.7554/eLife.27430

- HA Xu, A Modirshanechi, MP Lehmann, W Gerstner, and MH Herzog (2021), PLOS Computational Biology 17(6): e1009070. DOI: 10.1371/journal.pcbi.1009070

- M Dubois, J Habicht, J Michely, R Moran, RJ Dolan, TU Hauser (2021), eLife 10:e59907. DOI: 10.7554/eLife.59907

Reviewer #4 (Remarks to the Author):

The revisions to this registered report proposal have greatly improved the study plan. I address my previous points one by one, in the same order (I'm reviewer 4).

1. As you note, your exploration may reveal more than one substantive (non-methodological) factor, in which case, you would be in the MTMM situation. Regardless, the new approach seems much improved from the previous iteration. Comparing the single factor, to a two factor methodological model, to a bifactor model (two factors for method and one for exploration) seems likely to be a fruitful way to approach the problem.

2. The updated plan for assessing measurement reliability is improved. My personal preference is that rather than using cutoffs for reliability to make post hoc modifications to measures (self-report measures) or to change the measurement strategy (behavioral measures) is to instead simply assess reliability as the measures are typically used in the literature, and if the values are low, interpret subsequent correlations accordingly. For instance, if you found that behavioral measures tended to have dramatically lower reliability than self-report measures, then (even though this is somewhat expected and predictable), you know that this puts an upper limit on correlations with these measures. It suggests limitations to the value of measures with low reliability. So these would be caveats to discuss and interpret, rather than indications that modifications are necessary. These reliabilities are estimates, which should be fairly precise owing to your fairly large planned sample, but they are still estimates nonetheless.

3. I am glad that the authors plan to test measurement invariance. I agree that the choice to test their SEM model for invariance will be best. I would not use a statistical significance cutoff, and would instead consider using alternative fit indices (Chung & Rensvold, 2002) or effect sizes (Nye & Drasgow, 2011).

4. This point is resolved.

5. Thanks for the additional information about personality measures. I would like to see all collected measures disclosed, perhaps in an appendix. The other measures are outlined in full in the supplemental materials, but I do not see any information about the BFI in there (apologies if I missed it).

6. I do not know if the procedure for Nature Communications requires it for registered reports, but I recommend preregistering the study (once the final version has been agreed on here in the review process), prior to the onset of data collection. By this I mean that the authors should complete one of the registration forms on OSF (here are several options: <https://osf.io/zab38/wiki/home/>). The reason to register the study is as follows: (1) it makes the study discoverable in the registry later on, (2) you can include the stage 1 protocol as an attachment, preserving it for future readers, (3) you will fill out all of the fields in the preregistration form, making certain that they are all addressed. You should be able to use prose from this proposal to complete the majority of the fields, but it's a useful exercise nonetheless to make sure that your responses are as complete and well-defined as you might hope. If this journal has a different procedure, obviously you would follow that, but if the procedure is not clear, this would be my recommendation.

7-9. These points are addressed.

Thank you for taking the time to address my earlier critiques, and I look forward to learning the outcomes of the study.

I sign all of my reviews,
Katherine S. Corker

Reviewers 1 & 2 recommend acceptance without further revisions

Response. Great to hear.

Reviewer 3

Minor Comment 1

The new analysis plan for testing hypothesis 2 is promising. I understand that the plan follows the common practice in frequentist statistics, but, as someone with a background in machine learning, I was wondering why you do not use the 1st half of the data also for fitting the parameters of the 6 models (in addition to deriving the 3 exploratory models)?

Then you can use the 2nd half only for ‘testing’ and comparing the models. I would be more convinced by the results of such analyses.

Response. We agree with the reviewer that it makes sense to use the 1st half of the data also for fitting the parameters of the confirmatory models. However, this means that, in the first half of the data that are used to derive the exploratory data-driven models, the confirmatory models will likely not have as good fit statistics as the exploratory models. This is likely to happen because the exploratory models are selected due to the fact that they are the best models that can be extracted from that sample. For this reason, the exploratory models should fit the first half of the data better than they fit the second half of the data used for testing. On the other hand, the confirmatory models should fit both halves of the full dataset equally well. Taken together, whereas the exploratory models should show poorer fit in the second test half of the data as compared to the first half, the confirmatory models should show little change in fit (this latter would suggest that there is little to no systematic change across the two halves of the data). Therefore, we will fit the confirmatory models on both the first half and the second half of the data, but the final decision regarding selection of the best fitting model will be based only on the results found in the second half of the data.

We have updated the analysis plan in the text:

“With one half of the data, we will first derive three exploratory data-driven models **and do a preliminary check of how well the confirmatory models fit the data.**” (p. 34).

“By now, we will have used a random half of our data to derive the above exploratory models **and preliminarily examined the confirmatory models.**” (p. 39).

Minor Comment 2

I think the newly added section “Selection of behavioural tasks” (lines 441 to 569) and particularly the paragraph on the usage of computational methods for measuring exploration (lines 565 to 569) are well and fairly written and put the manuscript in the right context. In the current version, I could not find any reference to studies that used computational models of exploration to explain experimental data. Because the part has also a purpose of reviewing the literature, I think it would increase the strength and quality of the manuscript if the authors point readers to some relevant examples in the final version of the manuscript, e.g.,

- WK Zajkowski, M Kossut, RC Wilson (2017), eLife 6:e27430. DOI: 10.7554/eLife.27430

- HA Xu, A Modirshanechi, MP Lehmann, W Gerstner, and MH Herzog (2021), PLOS Computational Biology 17(6): e1009070. DOI: 10.1371/journal.pcbi.1009070

- M Dubois, J Habicht, J Michely, R Moran, RJ Dolan, TU Hauser (2021), eLife 10:e59907. DOI: 10.7554/eLife.59907

Response. This is a good suggestion, as readers may like to be pointed to relevant literature. We have added these citations to the manuscript, specifically following the sentence that reads, “Importantly, behaviour in the present study’s tasks have been, and can be, computationally modelled with parameters that represent a latent tendency to explore (e.g., ^{93–95}).” (p. 26).

Reviewer 4

Point 1

As you note, your exploration may reveal more than one substantive (non-methodological) factor, in which case, you would be in the MTMM situation. Regardless, the new approach seems much improved from the previous iteration. Comparing the single factor, to a two factor methodological model, to a bifactor model (two factors for method and one for exploration) seems likely to be a fruitful way to approach the problem.

Response. Thank you for the suggestions and we’re glad you agree that our analysis plan is now much improved.

Point 2

The updated plan for assessing measurement reliability is improved. My personal preference is that rather than using cutoffs for reliability to make post hoc modifications to

measures (self-report measures) or to change the measurement strategy (behavioral measures) is to instead simply assess reliability as the measures are typically used in the literature, and if the values are low, interpret subsequent correlations accordingly. For instance, if you found that behavioral measures tended to have dramatically lower reliability than self-report measures, then (even though this is somewhat expected and predictable), you know that this puts an upper limit on correlations with these measures. It suggests limitations to the value of measures with low reliability. So these would be caveats to discuss and interpret, rather than indications that modifications are necessary. These reliabilities are estimates, which should be fairly precise owing to your fairly large planned sample, but they are still estimates nonetheless.

Response. This is a very good suggestion. Therefore, we have updated our analysis plan so that we will use the results from the reliability analyses to determine our conclusions. The relevant section of the analysis plan for internal consistency now says (p. 31):

“Moreover, the measures we have proposed, including the aggregation of the behavioural measures across trials and blocks of a task, are as they are used in the literature. It is important to examine whether the measures, as used in the literature, have convergent validity with one another. Taken together, therefore, instead of using cut-offs to modify the measures or analysis plan, we will temper our conclusions to take account of any results that show a measure to have dramatically low internal consistency relative to the other measures. Specifically, very poor internal consistency of a measure will put an upper bound on how strongly the measure can correlate with other measures. We will further conclude that a measure with poor internal consistency should be treated with caution as it may be indicative of a poor measure.”

Point 3

I am glad that the authors plan to test measurement invariance. I agree that the choice to test their SEM model for invariance will be best. I would not use a statistical significance cutoff, and would instead consider using alternative fit indices (Chueng & Rensvold, 2002) or effect sizes (Nye & Drasgow, 2011).

Response. Following the reviewer’s suggestion, we changed the measurement invariance analysis as follows (p. 32):

“We will test the model selected from the factor analyses for Hypothesis 2 for different levels of invariance to time using the differences between fit indices of the configural, weak factorial, strong factorial, and strict invariance models. Specifically, the fit indices we will use are ΔCFI , $\Delta \Gamma$, and $\Delta \text{McDonald's NCI}$ (Chueng & Rensvold, 2002). As decision criteria, a ΔCFI larger than .01, a $\Delta \Gamma$ larger than .001, and $\Delta \text{McDonald's NCI}$ larger than .02 indicate that the null hypothesis of invariance should be rejected and that the

measurement model is thus not invariant across time. If at any level we cannot establish invariance for the entire construct, we will use the bias-corrected bootstrap confidence intervals approach to identify and remove any problematic tasks and try to establish partial measurement invariance for the remaining tasks at the same level. The bias-corrected bootstrap confidence intervals approach to measurement invariance is centered on evaluating the confidence intervals for all the differences between parameters in the model. If a confidence interval does not include zero, it can be inferred that the specific parameter is not invariant.”

Point 4

This point is resolved.

Response. Great!

Point 5

Thanks for the additional information about personality measures. I would like to see all collected measures disclosed, perhaps in an appendix. The other measures are outlined in full in the supplemental materials, but I do not see any information about the BFI in there (apologies if I missed it).

Response. We have now reported the BFI-2 items that we include for exploratory purposes in the Supplemental.

Point 6

I do not know if the procedure for Nature Communications requires it for registered reports, but I recommend preregistering the study (once the final version has been agreed on here in the review process), prior to the onset of data collection. By this I mean that the authors should complete one of the registration forms on OSF (here are several options: <https://osf.io/zab38/wiki/home/>). The reason to register the study is as follows: (1) it makes the study discoverable in the registry later on, (2) you can include the stage 1 protocol as an attachment, preserving it for future readers, (3) you will fill out all of the fields in the preregistration form, making certain that they are all addressed. You should be able to use prose from this proposal to complete the majority of the fields, but it's a useful exercise nonetheless to make sure that your responses are as complete and well-defined as you might hope. If this journal has a different procedure, obviously you would follow that, but if the procedure is not clear, this would be my recommendation.

Response. We'll preregister the Stage 1 manuscript as a whole document on the OSF.

Points 7-9

These points are addressed.

Response. Great!

Review Study Plan (Stage 2)

REVIEWER COMMENTS

Reviewer #1 (Remarks to the Author):

Please see my comments related to the questions for the stage 2 review below.

- Whether the data are able to test the authors' proposed hypotheses by passing the approved outcome-neutral criteria (such as absence of floor and ceiling effects or success of positive controls)

Comment: Yes, the data seems to pass the approved outcome-neutral criteria. The only measure that may potentially be problematic is the Observe of Bet measure. However, the Observe of Bet measure was considered independent of all other measures in the analyses, and is therefore unlikely to substantially affect the interpretation of the presented analyses.

- Whether the Introduction, rationale and stated hypotheses are the same as the approved Stage 1 submission (I have appended a copy of the in-principle accepted Stage 1 manuscript and Supplementary Information)

Comment: As far as I can tell the introduction, rationale, and stated hypotheses are identical to the Stage 1 manuscript, apart from minor changes such as verb tenses. However, I leave it up to the publisher to compare the stage 1 and stage 2 version in detail.

- Whether the authors adhered to the registered experimental procedures

Comment: Yes, as far as I can tell, the authors adhered to the registered experimental procedures.

- Whether any unregistered exploratory analyses added by the authors are justified, methodologically sound, and informative

Comment: The authors provided additional unregistered analyses (see Results re Hypothesis 2, for example), which were clearly labelled as unregistered analyses, were well justified and informative. As far as I can tell, these additional analyses were also methodologically sound.

- Whether the authors' conclusions are justified given the data

Comment: I believe the conclusions are justified given the data and the analysis. Having said this, the majority of the discussion section seems to provide ideas/discussions about why behavioural (lab-based) measures would not capture an existing domain-general tendency to explore in real life. While this is an important discussion, I would liked to read more about the alternative explanation, i.e. that there is no such thing as a domain-general tendency to explore in real life, and that the implications of this explanation are for the field and theories of exploitation and exploration.

Reviewer #2 (Remarks to the Author):

The Stage 2 registered report adheres to the authors' experimental and analysis plans described in Stage 1. The focus of the manuscript is on the planned analyses, with few unplanned analyses (and these unplanned analyses are clearly marked). The results and discussion are clearly written. I recommend that the manuscript be accepted as-is. I also commend the authors for conducting a data-intensive project. Collecting data from five separate behavioral tasks (at two time points) is tantamount to running five experiments (in effort). It is always great to see comprehensive

research like this when the authors' time resources could have been devoted to multiple, smaller projects that might be less impactful.

With regards to the results, it is dismaying to see that there is virtually no evidence for convergent validity among the tasks. Firstly, because psychologists often use the same language and refer to the same cognitive constructs in discussing these tasks. Most often, research that measures "exploration" chooses to only one behavioral task to measure. This work casts doubt on existing research that has sought to discover associations between exploratory behavior and other constructs. Secondly, because the goal of any psychological task is to measure behavior that is expected to generalize (e.g., to real-world behavior). If the exploratory behavior among experimental tasks is virtually uncorrelated, it casts doubt on the Discussion section of many papers that wax philosophical on how their results generalize to real-world behavior. Unfortunately, this is not unique to exploratory behavior - the authors point out a similar lack of convergent validity in other domains such as risk preference. This is surely problematic for a field that is focused on modeling "experimental tasks," while naturalistic studies are few and far between. I hope the field takes notice.

Reviewer #3 (Remarks to the Author):

The authors did a fantastic job of delivering what they had promised in the first stage of the submission. The results and methodology for testing Hypotheses 1 and 3 are in general sound and closely followed by what was agreed on in the first stage. The methodology for testing Hypothesis 2 is also closely followed by the preregistered report, but I have some concerns regarding the reported results and their interpretation for Hypothesis 2. I discuss this further below.

Overall, I believe that addressing my comments should be straightforward, and I believe that the manuscript is going to be an important seminal work for our community.

* Main comment: Results and their interpretation for Hypothesis 2:

1. I am a bit confused by the results in Table 10. First, it seems like the differences between the training and testing performance are not calculated correctly (e.g., if the other values in the table are correct, then the difference in CFI for Exploratory 1 must be $0.988 - 0.901 = 0.087$, but the reported value is -0.005). This makes me wonder whether there are other errors in the reported values and how much I can trust the results. In addition, I am puzzled by why **all** models perform better on the testing set than the training set for **all** performance measures. This is odd; if this is indeed the case and the reported values are correct, do the authors have any intuition/insight on why this is the case? I believe this should be commented on in the text.

2. Let us assume for now that the main values in Table 10 are correct. These results imply that both models Baseline 3 and Exploratory 1 are relatively good fits to the data -- while Exploratory 1 is slightly better than Baseline 3. Although I agree with the general conclusion of the authors that there seems to be little evidence for a general tendency to explore given all results for Hypotheses 1 and 2, I do not think the authors' argument against Baseline 3 is strong/rigorous enough. First, the significance of the difference between the goodness of fit of Baseline 3 and Exploratory 1 in Table 10 is not clear to me; in particular, the authors themselves mention that "Regarding the best fitting models, Baseline3 and Exploratory1 are similar, though Exploratory1 has better-fit statistics in the test sample. Ordinarily, therefore, we could select both Baseline3 and Exploratory1" [page 61]. The authors' main argument against Baseline 3 is based on the loadings in Tables 7 and 11, where they claim that Baseline 3 does not make sense from a "theoretical perspective". This is a bit circular: The argument is based on the assumption that all 11 (model-free) measures considered here must be positively correlated with a general tendency to explore, while at the same time, an important question is whether the (model-free) measures that are frequently used in the field are appropriate measures of exploration -- due to all potential flaws that the authors discuss in the method section and we touched upon in the reviews of stage 1. In other words, one

could in principle argue that Baseline 3 does actually make sense from a theoretical perspective and additionally implies that the commonly used (model-free) measures of exploration are not necessarily pure measures of exploration -- which explains why a weighted (with mixed signs) combination of them appear to be linked to a general tendency to explore. Overall, I do not think the authors are making a strong and rigorous case by the argument of "not making sense from a theoretical perspective"; I would suggest toning down the argument and focusing on the goodness of fit -- unless I have missed something important in their line of reasoning.

* Other comments

1. The presentation of results is not consistent throughout the manuscript, is in some situations incomplete, and overall needs a significant amount of polishing. I mention a couple of examples here:

1.1. The names of different measurements are not consistent throughout the manuscript (e.g., Figure 4 versus Tables 5 and 6). This makes following the arguments particularly difficult when multiple figures and tables are involved (e.g., I had to keep reminding myself that "General Explore-Exploit Question" is the same as "single_item_exploration").

1.2. Figure 5: There is no explanation about the links between nodes and what they imply (both their existence and their thickness). It remains unclear why there is no link between hamming distance in the alien game and bandit exploit compliment or why self-reports are not in a fully connected cluster instead of a line (as expected from the results of Hypothesis 1).

1.3. Figure 6: It is not mentioned what the dash lines are.

1.4. Figure 4: This is minor, but I think some color codes for the labels (or different thicknesses of table lines) can be helpful to make the difference between self-reports and behavioral tasks more apparent.

2. I liked the discussion section, the speculation about the potential explanation of the results, and the potential future directions. I would like to add some remarks and point authors to some related references.

2.1. As mentioned by the authors, the results for hypotheses 1 and 2 are in line with recent findings in several other domains in the behavioral sciences for the lack of convergent validity. It would be interesting to mention that this is not limited to behavioral measurements and that there have been recent studies showing little evidence for the convergence validity of even physiological measurements (e.g., <https://doi.org/10.1093/cercor/bhac309> and <https://doi.org/10.1016/j.isci.2023.106017>).

2.2. The authors point out that studying real-world exploratory behaviors is necessary for making a strong conclusion about the existence of a general tendency to explore. While I fully agree with them, I would like to point out that the middle ground is to go beyond simple experimental tasks on exploration-exploitation trade-off and consider, for example, (i) non-instrumental exploration tasks (<https://doi.org/10.1038/s41583-018-0078-0>), (ii) tasks for exploration in complex environments with sparse rewards (<https://doi.org/10.1016/j.tins.2023.10.002>), and (iii) exploration in computer games (<https://doi.org/10.31234/osf.io/hbsvj> and <https://doi.org/10.1038/s41562-023-01661-2>).

2.3. I really liked the idea of the authors that the tendency to explore may be domain-specific. I think one way to quantify and test this idea in future studies is based on models that modulate the value of information and exploration dependent on the context (e.g., ideas similar to <https://doi.org/10.1037/rev0000349> and <https://doi.org/10.1101/2022.07.05.498835>).

Reviewer #4 (Remarks to the Author):

I reviewed this Stage 2 Registered Report from my perspective as a Stage 1 reviewer. It appears that the planned research was executed as agreed. It also appears that the authors made a good faith effort to disclose any deviations from the Stage 1 protocol. The final product aligns with research in other areas showing little convergent validity between behavioral tasks designed to measure a purportedly similar construct. I really enjoyed reading the discussion section, which seemed to be a thoughtful and careful interpretation of the results. I'm excited for this study to be out – it should be a quite a nice model for other projects with other traits.

One small request – can you please cite the Stage 1 registration, data, materials, and code in the paper itself (with the DOIs and all)? That way, future readers will always have it (even if publishers change the way they report disclosures like “data availability” and so on.

I sign all of my reviews,
Katherine S. Corker

Reviewer #1 (Remarks to the Author):

Please see my comments related to the questions for the stage 2 review below.

- Whether the data are able to test the authors' proposed hypotheses by passing the approved outcome-neutral criteria (such as absence of floor and ceiling effects or success of positive controls)

Comment: Yes, the data seems to pass the approved outcome-neutral criteria. The only measure that may potentially be problematic is the Observe of Bet measure. However, the Observe of Bet measure was considered independent of all other measures in the analyses, and is therefore unlikely to substantially affect the interpretation of the presented analyses.

- Whether the Introduction, rationale and stated hypotheses are the same as the approved Stage 1 submission (I have appended a copy of the in-principle accepted Stage 1 manuscript and Supplementary Information)

Comment: As far as I can tell the introduction, rationale, and stated hypotheses are identical to the Stage 1 manuscript, apart from minor changes such as verb tenses. However, I leave it up to the publisher to compare the stage 1 and stage 2 version in detail.

- Whether the authors adhered to the registered experimental procedures

Comment: Yes, as far as I can tell, the authors adhered to the registered experimental procedures.

- Whether any unregistered exploratory analyses added by the authors are justified, methodologically sound, and informative

Comment: The authors provided additional unregistered analyses (see Results re Hypothesis 2, for example), which were clearly labelled as unregistered analyses, were well justified and informative. As far as I can tell, these additional analyses were also methodologically sound.

- Whether the authors' conclusions are justified given the data

Comment: I believe the conclusions are justified given the data and the analysis. Having said this, the majority of the discussion section seems to provide ideas/discussions about why behavioural (lab-based) measures would not capture an existing domain-general tendency to explore in real life. While this is an important discussion, I would like to read more about the alternative explanation, i.e. that there is no such thing as a domain-general tendency to explore in real life, and that the implications of this explanation are for the field and theories of exploitation and exploration.

Response. We thank the reviewer for their effort, time, and kind words. We have added some additional discussion about the implications of there being no domain-general tendency (pp. 72-73):

“In case people’s exploratory preferences are context-dependent or domain-specific, an important task for future research and theorizing would be to define the boundaries of the domains. For example, is it a single domain of exploratory behaviour when people try out new foods, regardless of whether it is at new restaurants or new recipes to cook at home, or are these domains separate from one another? The boundaries of each domain should thus be outlined and conclusions from any given study should be clear about the contexts to which the results can be generalized. Theories and the parameters thereof would have predictive power only within a domain’s boundaries and new empirical studies would be needed for different domains. Furthermore, the parameters of cognitive models used to explain people’s behaviour in the exploration-exploitation dilemma would need to be retrieved in each specific context separately (see ⁵⁹).”

Reviewer #2 (Remarks to the Author):

The Stage 2 registered report adheres to the authors' experimental and analysis plans described in Stage 1. The focus of the manuscript is on the planned analyses, with few unplanned analyses (and these unplanned analyses are clearly marked). The results and discussion are clearly written. I recommend that the manuscript be accepted as-is. I also commend the authors for conducting a data-intensive project. Collecting data from five

separate behavioral tasks (at two time points) is tantamount to running five experiments (in effort). It is always great to see comprehensive research like this when the authors' time resources could have been devoted to multiple, smaller projects that might be less impactful.

Response. We thank the reviewer for their kind words and appreciation for our project.

With regards to the results, it is dismaying to see that there is virtually no evidence for convergent validity among the tasks. Firstly, because psychologists often use the same language and refer to the same cognitive constructs in discussing these tasks. Most often, research that measures "exploration" chooses to only one behavioral task to measure. This work casts doubt on existing research that has sought to discover associations between exploratory behavior and other constructs. Secondly, because the goal of any psychological task is to measure behavior that is expected to generalize (e.g., to real-world behavior). If the exploratory behavior among experimental tasks is virtually uncorrelated, it casts doubt on the Discussion section of many papers that wax philosophical on how their results generalize to real-world behavior. Unfortunately, this is not unique to exploratory behavior - the authors point out a similar lack of convergent validity in other domains such as risk preference. This is surely problematic for a field that is focused on modeling "experimental tasks," while naturalistic studies are few and far between. I hope the field takes notice.

Response. At first we were also dismayed. But after thinking deeper about where this leaves us as a field and writing about the potential avenues for future research we started to get excited about the potential of doing research to answer the questions that arose from our work. We also hope the field takes notice and shares our excitement.

Reviewer #3 (Remarks to the Author)

The authors did a fantastic job of delivering what they had promised in the first stage of the submission. The results and methodology for testing Hypotheses 1 and 3 are in general sound and closely followed by what was agreed on in the first stage. The methodology for testing Hypothesis 2 is also closely followed by the preregistered report, but I have some concerns regarding the reported results and their interpretation for Hypothesis 2. I discuss this further below.

Overall, I believe that addressing my comments should be straightforward, and I believe that the manuscript is going to be an important seminal work for our community.

Response. We thank the reviewer for their work and kind words, and also for their comments helping us to improve our manuscript. We address the Reviewer's concerns below.

* Main comment: Results and their interpretation for Hypothesis 2:

1. I am a bit confused by the results in Table 10. First, it seems like the differences between the training and testing performance are not calculated correctly (e.g., if the other values in the table are correct, then the difference in CFI for Exploratory 1 must be $0.988 - 0.901 = 0.087$, but the reported value is -0.005). This makes me wonder whether there are other errors in the reported values and how much I can trust the results. In addition, I am puzzled by why **all** models perform better on the testing set than the training set for **all** performance measures. This is odd; if this is indeed the case and the reported values are correct, do the authors have any intuition/insight on why this is the case? I believe this should be commented on in the text.

Response. Regarding the Reviewer's first point about the incorrect numbers in Table 10, we found that there are indeed incorrect numbers regarding the differences between the training and testing performance. However, the problems were due only to copying/pasting, not wrong calculations in our analyses. We redid all of our analyses to guarantee that the main results were correct.

Regarding the observation that all of the fit statistics improved in going from the training sample to the test samples for all models, we provide new supplementary analyses to show that the models performed better on the testing sample than on the training sample simply due to chance. We added these analyses to the Supplemental (under the heading "Main Study: Sensitivity Analysis of Differences Between Training and Testing Samples" on pp. 15-18 of Supplemental). In sum, to examine whether the observation that all models had better fit indices in the test sample was due to chance, we conducted two sets of non-preregistered analyses reported in the Supplemental (see section "Main Study: Sensitivity Analysis of Differences Between Training and Testing Samples"). We note that the CFI, TLI, and RMSEA are sensitive to the values of the correlations between the variables, so that an improvement in one will be reflected by improvements also in the others. In other words, when one fit statistic improves the others are also very likely to improve. Importantly, our Supplementary analyses showed that the differences in the indices between the testing and training samples occurred by chance (i.e., with 1,000 random splits we found that the fit statistics were in most cases marginally better in the training samples, and that confidence intervals for the fit statistics overlap). Regarding this, we added some text the main paper and the Supplemental.

In the main text, we now state (p. 61):

"We note here that all models had better fit statistics in the test sample as compared to the training sample; but non-preregistered analyses showed that this was due to chance (see Supplemental, section titled "Main Study: Sensitivity Analysis of Differences Between Training and Testing Samples")."

In the Supplemental we say (pp. 15-19):

"The results shown in Table 10 of the main text indicate that the models provided a better fit for the testing sample. Because CFI, TLI, and RMSEA are sensitive to the values of the correlations between the variables, an improvement in one will be reflected by improvements also in the others. And it is quite likely that the random split, by pure chance, made the correlations in the testing sample somewhat larger than the correlations in the training sample. We did two analyses to assess the stability of our results. In short, our analyses showed that the differences in the indices between the testing and training samples occurred by chance (i.e., with 1,000 random splits we found that the fit statistics were in most cases marginally better in the training samples, and that confidence intervals for the fit statistics overlap)."

“The first analysis consisted in comparing the correlations of the measures with one another in the training sample against the same correlations in the test sample. We did this using z tests. This analysis is done by, first, transforming correlation coefficient values, or r values, into z scores with Fisher’s transformation.”

We presented these results in Table S4 (which we don’t reproduce here due to how large it is).

“It is possible to see that only four correlation differences were statistically significant when comparing the training and the testing sample at the threshold of .05 for the p-value. Three of these values were negative, indicating larger values in the testing sample. However, with a Bonferroni correction of the threshold (i.e., .05 / 55) for the p-value, no difference was significant. Therefore, it is reasonable to infer that the differences between fit in the training and testing samples observed in Table 10 are due to chance.”

“The second analysis consisted in generating 1,000 random training and testing splits of the sample and calculating the mean and 95% confidence interval of the fit statistics of the models. The results of this analysis are shown in Table S5. Before discussing the results, we note that the correlations between the measures in the training sample are never going to be exactly the same as the correlations in the test sample. So it will *always* be the case that the correlations are either slightly higher or slightly lower in one sample than the other when the sample is split into two equal halves, assuming that the split into subsamples was random so that the relative correlations between the measures remains largely unchanged as compared to the correlations in the full sample (in this case, relative correlations being close to zero). Therefore, given the dependency of the fit statistics on the correlations between the measures, we would expect the fit statistics for all models to show a similar pattern when going from training to test samples. From Table S5, it is possible to see that, the fit statistics are very slightly better in the training sample than the test sample, but the confidence intervals almost always (10 out of 12 comparisons) overlap with the point estimates and *all* confidence intervals for all the fit statistics between the training and the testing samples overlap with one another. The fit statistics between the training and test samples can therefore not be statistically distinguished. Moreover, it is possible to see that the confidence intervals for all the fit statistics were better (i.e., larger CFI and TLI, and smaller RMSEA) for the Exploratory1 model, as compared to the Baseline3 model.”

We reproduce Table S5 below.

Table S5

Descriptive statistics of the 1,000 random splits of the sample

Model	Sample	Fit Statistic	Mean	SE	Lower Bound 95% CI	Upper Bound 95% CI
Baseline1	Training	CFI	0.4369	0.0020	0.4349	0.4389
	Testing		0.4367	0.0019	0.4348	0.4386
	Training	TLI	0.2961	0.0025	0.2936	0.2986
	Testing		0.2959	0.0024	0.2935	0.2983
	Training	RMSEA	0.1103	0.0002	0.1101	0.1105
	Testing		0.1106	0.0002	0.1104	0.1108
Baseline2	Training	CFI	0.7814	0.0013	0.7801	0.7828
	Testing		0.7806	0.0013	0.7793	0.7819
	Training	TLI	0.7205	0.0017	0.7188	0.7221

	Testing		0.7193	0.0017	0.7177	0.7210
	Training	RMSEA	0.0693	0.0002	0.0691	0.0695
	Testing		0.0695	0.0002	0.0693	0.0697
Baseline3	Training	CFI	0.9049	0.0014	0.9036	0.9063
	Testing		0.9040	0.0015	0.9026	0.9055
	Training	TLI	0.8416	0.0023	0.8393	0.8439
	Testing		0.8400	0.0024	0.8376	0.8425
	Training	RMSEA	0.0516	0.0004	0.0513	0.0520
	Testing		0.0519	0.0004	0.0515	0.0523
Exploratory1	Training	CFI	0.9537	0.0009	0.9528	0.9547
	Testing		0.9528	0.0009	0.9519	0.9537
	Training	TLI	0.9402	0.0013	0.9389	0.9414
	Testing		0.9389	0.0012	0.9376	0.9401
	Training	RMSEA	0.0302	0.0004	0.0298	0.0306
	Testing		0.0307	0.0004	0.0303	0.0311

Note. SE: standard error. CI: confidence interval.

2. Let us assume for now that the main values in Table 10 are correct. These results imply that both models Baseline 3 and Exploratory 1 are relatively good fits to the data -- while Exploratory 1 is slightly better than Baseline 3. Although I agree with the general conclusion of the authors that there seems to be little evidence for a general tendency to explore given all results for Hypotheses 1 and 2, I do not think the authors' argument against Baseline 3 is strong/rigorous enough. First, the significance of the difference between the goodness of fit of Baseline 3 and Exploratory 1 in Table 10 is not clear to me; in particular, the authors themselves mention that "Regarding the best fitting models, Baseline3 and Exploratory1 are similar, though Exploratory1 has better-fit statistics in the test sample. Ordinarily, therefore, we could select both Baseline3 and Exploratory1" [page 61]. The authors' main argument against Baseline 3 is based on the loadings in Tables 7 and 11, where they claim that Baseline 3 does not make sense from a "theoretical perspective". This is a bit circular: The argument is based on the assumption that all 11 (model-free) measures considered here must be positively correlated with a general tendency to explore, while at the same time, an important question is whether the (model-free) measures that are frequently used in the field are appropriate measures of exploration -- due to all potential flaws that the authors discuss in the method section and we touched upon in the reviews of stage 1. In other words, one could in principle argue that Baseline 3 does actually make sense from a theoretical perspective and additionally implies that the commonly used (model-free) measures of exploration are not necessarily pure measures of exploration -- which explains why a weighted (with mixed signs) combination of them appear to be linked to a general tendency to explore. Overall, I do not think the authors are making a strong and rigorous case by the argument of "not making sense from a theoretical perspective"; I would suggest toning down the argument and focusing on the goodness of fit -- unless I have missed something important in their line of reasoning.

Response. This point is well taken! We believe that we were not clear in our reasoning. Therefore, we have toned down our argument from a theoretical perspective and have clarified some things in the manuscript regarding Baseline3 (pp. 62):

“However, Exploratory1 makes more sense from a theoretical perspective as compared to Baseline3. In Baseline3, for the training sample, 5 of the 11 measures had negative loadings on the exploration factor, and only the two bandit measures had loadings above .300. For the test sample, as shown in Table 11, 2 of the 11 measures had negative loadings on the exploration factor, and only the two bandit measures and the hamming distance measure of the alien task had loadings above .300. Therefore, although Baseline3 had somewhat satisfactory fit indices, suggesting that the measures capture some common latent cause, we believe that the latent cause is uninterpretable in relation to a general tendency to explore.

The basic assumption in latent variable modelling (or psychometrics in general) is that correlations indicate that observed variables can be explained by the same latent common cause. In our case, higher values on the observed variables indicate that people explore more. Therefore, under the critical assumption that the latent variable is the general tendency to explore, we would expect the observed variables to be positively correlated with one another and to thus have positive loadings on the latent variable. Negative loadings are contradictory to the critical assumption. For example, the sampling paradigm has a negative loading on the latent variable in Baseline3. This means that in this task people who took more samples before choosing an option (i.e., those who could be said to have “explored more”) would have a lower general tendency to explore, which is at odds with the definition of the measure itself. From this perspective, therefore, the obtained negative loadings mean that the latent variable in Baseline3 makes little sense with respect to an assumed general tendency to explore.”

We also based our decision of the best fitting model on the factor loadings and interpretability because fit indices can be biased towards some models (e.g., <https://psycnet.apa.org/doi/10.1037/abn0000434>; <https://doi.org/10.3758/s13428-018-1055-2>; <https://doi.org/10.1080/00273171.2020.1779642>). In our understanding, using fit indices alone can lead to choosing models that overfit the data and are not interpretable.

Nonetheless, as suggested by the Reviewer, and as indicated in our response to the Reviewer’s previous point, we also refer more strongly to the better fit statistics of Exploratory1 relative to Baseline3 especially in the light of the new non-preregistered analyses (pp. 62-63):

“Regarding Exploratory1, only 1 measure loads slightly less than .300 on its respective factor and there are no measures with negative loadings. Moreover, Exploratory1’s structure better reflects the correlation patterns in Table 6 as compared to Baseline3. Importantly, in non-preregistered analyses, we did 1,000 random splits into training and test samples and calculated the mean for each of the fit statistics as well as their 95% confidence intervals. These are reported in the Supplemental (Table S5). Exploratory1 had significantly better fit statistics than Baseline3 (i.e., higher CFI and TLI, and lower RMSEA) as indicated by the fact that the confidence intervals did not overlap with the point estimates. Thus, Exploratory1 is the best fitting model based on model comparisons as well as on theoretical grounds.”

* Other comments

1. The presentation of results is not consistent throughout the manuscript, is in some situations incomplete, and overall needs a significant amount of polishing. I mention a couple of examples here:

1.1. The names of different measurements are not consistent throughout the manuscript (e.g., Figure 4 versus Tables 5 and 6). This makes following the arguments particularly difficult when multiple figures and tables are involved (e.g., I had to keep reminding myself that "General Explore-Exploit Question" is the same as "single_item_exploration").

Response. We have changed this to “single-item exploration question” throughout the entire manuscript, including in the methods section. Nothing else in the methods section was changed.

1.2. Figure 5: There is no explanation about the links between nodes and what they imply (both their existence and their thickness). It remains unclear why there is no link between hamming distance in the alien game and bandit exploit compliment or why self-reports are not in a fully connected cluster instead of a line (as expected from the results of Hypothesis 1).

Response. We added the following note to Figure 5 (p. 57).

Note. Nodes of the same colour form a common factor. The edges connecting the nodes represent the regularised partial correlations. Green edges are positive partial correlations and red edges are negative partial correlations. The strength of the partial correlation is represented by the thickness of the edges, with thicker edges representing stronger partial correlations. These partial correlations are used within EGA to estimate the clusters of variables and are not used inferentially in any other means in this study.

1.3. Figure 6: It is not mentioned what the dash lines are.

Response. We added the following complement to the note of Figure 6 (p. 59).

Note. SR: self-report measures' common factor. ExS: exploration scale. MTS: maximization tendency scale. AIS: alternative search. GEE: single-item exploration question. BnD: bandit task common factor. SwR: multi-armed bandit switch-rate. BRC: multi-armed bandit best-reply complement. Aln: alien game common factor. AcS: alien game active search. HaD: alien game Hamming distance. OSW: optional stopping with recall. Smp: sampling paradigm. OoB: observe or bet. The dashed arrows represent items with loadings fixed to 1.

1.4. Figure 4: This is minor, but I think some color codes for the labels (or different thicknesses of table lines) can be helpful to make the difference between self-reports and behavioral tasks more apparent.

Response. Given that this is only a minor point, and is regarding preferences rather than something substantive, we have chosen to keep the figure as is. This is because it should be clear that the cluster of correlated measures are the self-reports, as indicated in the main text. However, we noticed that the Figure's notes did not show the variable names and what they are. We have therefore added notes to the Figure 4 to indicate the variable names: (p. 50):

("bandit_switch = switch rate in multi-armed bandit task. bandit_exploit_complement = the complement of the best reply rate in multi-armed bandit task. alien_hamming = Hamming distance in alien game. alien_active_search = active search measure in alien game. optional_stopping = exploration in the optional stopping task. sampling = average number of samples per block in the sampling paradigm. observe_bet = average number of observe trials in the observe or bet task. alternative_search = mean ratings on the items of the alternative search scale. maximization = mean ratings on the items of the maximization tendency scale. exploration = mean ratings on the items of the exploration scale. single_item_exploration = ratings on the single-item measure of the general tendency to explore-exploit.").

It should therefore now be clear which are the self-reports.

2. I liked the discussion section, the speculation about the potential explanation of the results, and the potential future directions. I would like to add some remarks and point authors to some related references.

2.1. As mentioned by the authors, the results for hypotheses 1 and 2 are in line with recent findings in several other domains in the behavioral sciences for the lack of convergent validity. It would be interesting to mention that this is not limited to behavioral measurements and that there have been recent studies showing little evidence for the convergence validity of even physiological measurements (e.g., <https://doi.org/10.1093/cercor/bhac309> and <https://doi.org/10.1016/j.isci.2023.106017>).

Response. We have added the following to the relevant section of the manuscript (pp. 68-69): "Similar issues have also been found for measures of physical properties, such as there being no correlations between different analysis methods of electroencephalogram (EEG) data obtained from the same group of participants.^{116,117}".

2.2. The authors point out that studying real-world exploratory behaviors is necessary for making a strong conclusion about the existence of a general tendency to explore. While I fully agree with them, I would like to point out that the middle ground is to go beyond simple experimental tasks on exploration-exploitation trade-off and consider, for example, (i) non-instrumental exploration tasks (<https://doi.org/10.1038/s41583-018-0078-0>), (ii) tasks for exploration in complex environments with sparse rewards (<https://doi.org/10.1016/j.tins.2023.10.002>), and (iii) exploration in computer games (<https://doi.org/10.31234/osf.io/hbsvj> and <https://doi.org/10.1038/s41562-023-01661-2>).

Response. After having looked through the reviewer's suggested references, we have added the following to the manuscript (p. 75):

"In addition, researchers could examine exploratory behaviour in environments that are more inherently motivating such as in videogames^{131,132} and virtual reality environments¹³³ to see whether behaviour in such contexts correlates with behaviour in the other contexts mentioned already. A major challenge down this path will be in quantifying exploration in real-world contexts, and connecting these to already established measures of exploratory behaviour in the lab or with self-reports."

2.3. I really liked the idea of the authors that the tendency to explore may be domain-specific. I think one way to quantify and test this idea in future studies is based on models that modulate the value of information and exploration dependent on the context (e.g., ideas similar to <https://doi.org/10.1037/rev0000349> and <https://doi.org/10.1101/2022.07.05.498835>).

Response. With respect, we have looked at the references provided by the Reviewer and cannot see how to use them for the ideas in our manuscript. Nonetheless, we believe that the Reviewer’s general point about quantifying and examining domain-specific exploration is addressed by our response to Reviewer 1. Specifically, we have added some ideas to the discussion section about domain-specific exploration and the implications of this for researchers, and how researchers may examine such domain-specific tendencies.

Reviewer #4 (Remarks to the Author):

I reviewed this Stage 2 Registered Report from my perspective as a Stage 1 reviewer. It appears that the planned research was executed as agreed. It also appears that the authors made a good faith effort to disclose any deviations from the Stage 1 protocol. The final product aligns with research in other areas showing little convergent validity between behavioral tasks designed to measure a purportedly similar construct. I really enjoyed reading the discussion section, which seemed to be a thoughtful and careful interpretation of the results. I’m excited for this study to be out – it should be a quite a nice model for other projects with other traits.

One small request – can you please cite the Stage 1 registration, data, materials, and code in the paper itself (with the DOIs and all)? That way, future readers will always have it (even if publishers change the way they report disclosures like “data availability” and so on.

I sign all of my reviews,
Katherine S. Corker

Response. We thank the Reviewer for their kind words. We have added the DOI registration link for the Stage 1 manuscript to the paper in the Protocol Registration section (p. 77):

“We preregistered the accepted Stage 1 manuscript on the OSF prior to any data collection. The registration can be found here: <https://doi.org/10.17605/OSF.IO/64QJU>”.

We have also added a DOI link for the data, materials, and code in the Code Availability section (p. 80):

“The data for both the pilot and the main study, the analyses codes for the pilot and main studies, and the power analysis code for the equivalence testing, can be found on the Open Science Framework via this link to facilitate blind peer review: <https://doi.org/10.17605/OSF.IO/F62MY>”.

REVIEWERS' COMMENTS

Reviewer #3 (Remarks to the Author):

I thank the authors for addressing all my comments carefully. I believe that the manuscript is now ready for publication and will be a great resource for future studies in the field.

On last minor comment (typo): To have consistency across models in Table 10, the values in the last row (i.e., the "Difference" row for "Exploratory1") must have negative signs (so that Difference is always equal to Test - Training).

Reviewer 3

I thank the authors for addressing all my comments carefully. I believe that the manuscript is now ready for publication and will be a great resource for future studies in the field.

On last minor comment (typo): To have consistency across models in Table 10, the values in the last row (i.e., the "Difference" row for "Exploratory1") must have negative signs (so that Difference is always equal to Test - Training).

Response. We have now made this change.
--